# Non-stationary Bandit Convex Optimization: A Comprehensive Study

Xiaoqi Liu[*]    Dorian Baudry[*†]    Julian Zimmert[‡]    Patrick Rebeschini[*]    Arya Akhavan[*§]

## Abstract

Bandit Convex Optimization is a fundamental class of sequential decision-making problems, where the learner selects actions from a continuous domain and observes a loss (but not its gradient) at only one point per round. We study this problem in non-stationary environments, and aim to minimize the regret under three standard measures of non-stationarity: the number of switches $S$ in the comparator sequence, the total variation $\Delta$ of the loss functions, and the path-length $P$ of the comparator sequence. We propose a polynomial-time algorithm, Tilted Exponentially Weighted Average with Sleeping Experts (TEWA-SE), which adapts the sleeping experts framework from online convex optimization to the bandit setting. For strongly convex losses, we prove that TEWA-SE is minimax-optimal with respect to known $S$ and $\Delta$ by establishing matching upper and lower bounds. By equipping TEWA-SE with the Bandit-over-Bandit framework, we extend our analysis to environments with unknown non-stationarity measures. For general convex losses, we introduce a second algorithm, clipped Exploration by Optimization (cExO), based on exponential weights over a discretized action space. While not polynomial-time computable, this method achieves minimax-optimal regret with respect to known $S$ and $\Delta$, and improves on the best existing bounds with respect to $P$.

## 1 Introduction

Many real-world decision-making problems, such as resource allocation, experimental design, or hyperparameter tuning require repeatedly selecting an action from a continuous space under uncertainty and limited feedback. These settings are naturally modeled as Bandit Convex Optimization (see [1] for an introduction), in which an adversary fixes a sequence of $T$ loss functions $f_1, f_2, \ldots, f_T : \mathbb{R}^d \to \mathbb{R}$ beforehand, and a learner sequentially interacts with the adversary for $T$ rounds. At each round $t$, the learner selects an action $\boldsymbol{z}_t$ from a continuous arm set $\Theta \subseteq \mathbb{R}^d$, assumed to be convex and compact. The learner then incurs a loss $f_t(\boldsymbol{z}_t)$ and observes a noisy feedback:

$$y_t = f_t(\boldsymbol{z}_t) + \xi_t \,, \tag{1}$$

where $\xi_t$ is a sub-Gaussian noise variable (Definition 1). The goal is to minimize the learner's *regret* with respect to (w.r.t.) a performance benchmark. In the online learning literature [2, 3], the benchmark is typically the best *static* action in hindsight, with cumulative loss $\min_{\boldsymbol{z} \in \Theta} \sum_{t=1}^{T} f_t(\boldsymbol{z})$.

However, non-stationarity arises in many applications where different actions may work well during different time intervals. Hence, a line of works [4–8] propose to compare the learner's actions against

---

[*]University of Oxford. Correspondence to: {`shirley.liu, arya.akhavan`}`@stats.ox.ac.uk`.

[†]Univ. Grenoble Alpes, Inria, CNRS, Grenoble INP, LIG, 38000 Grenoble, France.

[‡]Google Research.

[§]École Polytechnique de Paris, IP Paris.

39th Conference on Neural Information Processing Systems (NeurIPS 2025).

a sequence of comparators $\boldsymbol{u}_1, \ldots, \boldsymbol{u}_T \in \Theta$, leading to a regret defined as

$$R(T, \boldsymbol{u}_{1:T}) \coloneqq \sum_{t=1}^{T} \mathbf{E}\left[f_t(\boldsymbol{z}_t) - f_t(\boldsymbol{u}_t)\right], \tag{2}$$

where $\boldsymbol{u}_{1:T}$ denotes $(\boldsymbol{u}_t)_{t=1}^{T}$, and the expectation is taken w.r.t. the randomness in the learner's actions $\boldsymbol{z}_t$'s and the randomness of the noise variables $\xi_t$'s, similarly to the standard notion of pseudo-regret in the bandit literature, see e.g., [9, Section 4.8]. Choosing the regret-maximizing comparator in (2) gives rise to the notion of *dynamic regret*, defined as

$$R^{\mathsf{dyn}}(T) \coloneqq \max_{\boldsymbol{u}_{1:T} \in \Theta^T} R(T, \boldsymbol{u}_{1:T}). \tag{3}$$

While addressing non-stationarity through dynamic regret has been extensively studied in multi-armed bandits (e.g., [10–12]), it remains relatively underexplored in continuum bandits [7, 13, 14]. This work aims to bridge this gap by proposing algorithms for Bandit Convex Optimization that achieve sublinear dynamic regret. Such a rate is generally unattainable without imposing structural constraints on the environment, i.e., the comparator sequence and the loss function sequence [6]. For the comparator sequence, two commonly studied constraints are the number of switches [15] and the path-length [4], defined respectively as

$$S(\boldsymbol{u}_{1:T}) \coloneqq 1 + \sum_{t=2}^{T} \mathbf{1}\left(\boldsymbol{u}_t \neq \boldsymbol{u}_{t-1}\right) \leq S, \qquad P(\boldsymbol{u}_{1:T}) \coloneqq \sum_{t=2}^{T} \|\boldsymbol{u}_t - \boldsymbol{u}_{t-1}\| \leq P. \tag{4}$$

For the loss function sequence, a popular constraint is the total variation [7], defined as

$$\Delta(f_{1:T}) \coloneqq \sum_{t=2}^{T} \max_{\boldsymbol{z} \in \Theta} |f_t(\boldsymbol{z}) - f_{t-1}(\boldsymbol{z})| \leq \Delta. \tag{5}$$

The constraints that the upper bounds $S$, $P$ and $\Delta$ respectively impose on the comparators or on the loss functions lead to different notions of regret. We call the regret for environments constrained by $S$ the *switching regret*, which we define as

$$R^{\mathsf{swi}}(T, S) \coloneqq \max_{\boldsymbol{u}_{1:T} : S(\boldsymbol{u}_{1:T}) \leq S} R(T, \boldsymbol{u}_{1:T}). \tag{6}$$

Similarly, we call the regret for environments constrained by $P$ the *path-length regret*, denoted by $R^{\mathsf{path}}(T, P)$. We also use $R^{\mathsf{dyn}}(T, \Delta)$ and $R^{\mathsf{dyn}}(T, \Delta, S)$ to denote the dynamic regret, where the arguments after $T$ specify environment constraints. See (8) for rigorous definitions.

We detail in Section 1.3 conversion results between the different regret definitions that we introduced: a sublinear switching regret $R^{\mathsf{swi}}(T, S)$ implies sublinear $R^{\mathsf{dyn}}(T, \Delta)$, $R^{\mathsf{dyn}}(T, \Delta, S)$ and $R^{\mathsf{path}}(T, P)$, as illustrated in Figure 1 (see also [16, 17]). Furthermore, the upper bounds on the switching regret presented in this work are derived from upper bounds on the *adaptive regret* [18, 19], which is defined using an interval length $\mathsf{B} \in [T]$ as follows,

$$R^{\mathsf{ada}}(\mathsf{B}, T) \coloneqq \max_{\substack{p, q \in [T], \\ 0 < q - p \leq \mathsf{B}}} \max_{\boldsymbol{u} \in \Theta} \sum_{t=p}^{q} \mathbf{E}\left[f_t(\boldsymbol{z}_t) - f_t(\boldsymbol{u})\right]. \tag{7}$$

With an appropriate tuning of $\mathsf{B}$ depending on $S$, an adaptive regret sublinear in $\mathsf{B}$ implies a switching regret sublinear in $T$ through a simple reduction, see e.g., discussions in [19]. We illustrate the relations between these regret notions in Figure 1.

Figure 1: Conversions between regrets: $R_1 \longrightarrow R_2$ means that if regret $R_1$ is sublinear in $T$ (or B), then regret $R_2$ is also sublinear in $T$, see Proposition 1 for precise mathematical statements.

We conclude this section by detailing the main notation and assumptions used throughout the paper.

**Notation** For $k \in \mathbb{N}^+$, we denote by $[k]$ the set of positive integers $\leq k$. We denote by $\| \cdot \|$ the Euclidean norm, $\mathbb{B}^d = \{ \boldsymbol{u} \in \mathbb{R}^d : \|\boldsymbol{u}\| \leq 1 \}$ the unit Euclidean ball, and $\Pi_\Theta(\boldsymbol{x}) = \arg\min_{\boldsymbol{w} \in \Theta} \|\boldsymbol{w} - \boldsymbol{x}\|$ the Euclidean projection of $\boldsymbol{x}$ to $\Theta$. We use $a \vee b \equiv \max(a, b)$ and $a \wedge b \equiv \min(a, b)$. If $A, B$ depend on $T$, we use $A = \mathcal{O}(B)$ (resp. $A = \Omega(B)$) when there exists $c > 0$ s.t. $A \leq cB$ (resp. $\geq$) with $c$ independent of $T, d, S, \Delta$ and $P$. To hide polylogarithmic factors in $T$, we use interchangeably $A = \widetilde{\mathcal{O}}(B)$ and $A \lesssim B$ (resp. $A = \widetilde{\Omega}(B)$ and $A \gtrsim B$), e.g., $A \leq T \log T \implies A = \widetilde{\mathcal{O}}(T)$. Moreover, $A = o(B)$ means $A/B \to 0$ as $T \to \infty$.

**Assumptions** For simplicity, we assume that the time horizon $T$ is known in advance; the case of unknown $T$ can be handled using the standard doubling trick [20]. For some $\sigma > 0$, the noise variables $(\xi_t)_{t=1}^T$ are $\sigma$-sub-Gaussian. For all $t \in [T]$, $\max_{\boldsymbol{x} \in \Theta} |f_t(\boldsymbol{x})| \leq 1$. We consider two cases: (i) general convex losses $f_t$, where we assume Lipschitz continuity with constant $K$, and (ii) the special strongly-convex case, where we assume $\beta$-smoothness. The domain $\Theta$ is assumed to contain a ball of radius $r$ for some constant $r > 0$, and has a bounded diameter $\text{diam}(\Theta) := \sup\{\|\boldsymbol{x} - \boldsymbol{w}\| : \boldsymbol{x}, \boldsymbol{w} \in \Theta\} \leq D$ for some constant $D > 0$.

## 1.1 Main contributions

Existing works on non-stationary Bandit Convex Optimization study different aspects of the problem in isolation: [7, 14] focus on dynamic regret $R^{\text{dyn}}(T, \Delta)$, while [13, 21] address path-length regret $R^{\text{path}}(T, P)$. The present work aims to systematically unify and extend previously scattered results, establishing a complete picture of the state-of-the-art regret bounds w.r.t. all three non-stationarity measures $S, \Delta$ and $P$.

Our first contribution is a polynomial-time algorithm called Tilted Exponentially Weighted Average with Sleeping Experts (TEWA-SE), which we design by adapting a series of works from online convex optimization [22–24] to the bandit setting with zeroth-order feedback. It addresses the absence of gradient information by employing the randomized perturbation technique from [25, 26] to estimate gradients, combined with the design of quadratic surrogate loss functions depending on a uniform upper bound on the norm of the gradient estimates.

Following [22–24], TEWA-SE runs multiple expert algorithms with different learning rates in parallel, and combines them using a tilted exponentially weighted average. This allows TEWA-SE to adapt to the curvature of the loss function $f_t$'s without prior knowledge of parameters such as the strong-convexity parameter. For a given interval length B, an appropriately tuned TEWA-SE simultaneously achieves an adaptive regret of the order $\sqrt{d}\mathsf{B}^{\frac{3}{4}}$ for general convex losses and $d\sqrt{\mathsf{B}}$ for strongly-convex losses (Theorem 1). Consequently, for a known $S$, we prove that an optimal tuning of TEWA-SE yields a switching regret upper bound of order $\sqrt{d}S^{\frac{1}{4}}T^{\frac{3}{4}}$ for general convex losses (Corollary 1). In the same result, we further prove that if the losses are strongly convex, and that $\Delta$ is known and incorporated in the tuning of TEWA-SE, the algorithm simultaneously satisfies a $\min\left\{d\sqrt{ST}, d^{\frac{2}{3}}\Delta^{\frac{1}{3}}T^{\frac{2}{3}}\right\}$ dynamic regret bound. Importantly, TEWA-SE does *not* need to know the actual strong-convexity parameter, inheriting the adaptivity properties of the framework developed in [22–24]. We prove that this dynamic regret upper bound is minimax-optimal in $T, d, S$ and $\Delta$ by establishing a matching lower bound (Theorem 2). Finally, still for strongly-convex losses, we prove that TEWA-SE can also achieve a path-length regret of the order $d^{\frac{2}{3}}P^{\frac{1}{3}}T^{\frac{2}{3}}$ when $P$ is known. We summarize these results in Table 1. To overcome the restriction of knowing $S, \Delta$ and $P$ to optimally tune TEWA-SE, we also analyze a variant equipped with the Bandit-over-Bandit framework [27].

Table 1: Regret bounds we obtain for $R^{\text{swi}}(T, S)$, $R^{\text{dyn}}(T, \Delta)$ and $R^{\text{path}}(T, P)$, respectively, for algorithms tuned with known $S, \Delta$ and $P$ (polylogarithmic factors omitted). Straight underlines indicate minimax-optimal rates. A wavy underline indicates the result is either new to the literature (strongly-convex case) or improves on the best-known $P^{\frac{1}{4}}T^{\frac{3}{4}}$ rate [13] (general convex case).

|  | TEWA-SE (Algorithm 1) | cExO (Algorithm 3) |
|---|---|---|
| Convex | $\sqrt{d}S^{\frac{1}{4}}T^{\frac{3}{4}}$, $d^{\frac{2}{5}}\Delta^{\frac{1}{5}}T^{\frac{4}{5}}$, $d^{\frac{2}{5}}P^{\frac{1}{5}}T^{\frac{4}{5}}$ | $d^{\frac{5}{2}}\sqrt{ST}$, $d^{\frac{5}{3}}\underline{\Delta^{\frac{1}{3}}T^{\frac{2}{3}}}$, $d^{\frac{5}{3}}\underwave{P^{\frac{1}{3}}T^{\frac{2}{3}}}$ |
| Strongly convex | $\underline{d\sqrt{ST}}$, $\underline{d^{\frac{2}{3}}\Delta^{\frac{1}{3}}T^{\frac{2}{3}}}$, $\underwave{d^{\frac{2}{3}}P^{\frac{1}{3}}T^{\frac{2}{3}}}$ | |

For general convex losses with known $S, \Delta$ and $P$, TEWA-SE achieves a suboptimal $T^{\frac{3}{4}}$ rate (Corollary 1), matching the rates in similar analysis for the static regret [25, 26]. Thus, the second contribution of this work is the clipped Exploration by Optimization (cExO) algorithm with improved guarantees for this setting, which uses exponential weights on a discretized action space $\Theta$ with clipping [28]. For a given interval length B, this algorithm with an optimally tuned learning rate w.r.t. B attains an order $d^{\frac{5}{2}}\sqrt{B}$ adaptive regret (Theorem 3). When $S, \Delta$ and $P$ are known beforehand, this algorithm with an optimally tuned learning rate achieves the minimax-optimal dynamic regret w.r.t. $S$ and $\Delta$ simultaneously, and attains a $P^{\frac{1}{3}}T^{\frac{2}{3}}$ path-length regret (Corollary 2), improving on the previous best $P^{\frac{1}{4}}T^{\frac{3}{4}}$ [13]. While this algorithm is not polynomial-time computable and has suboptimal rates w.r.t. the problem dimension $d$, it provides insights that may guide future research toward developing efficient algorithms with optimal guarantees for the convex case.

## 1.2 Related work

The literature on *Bandit Convex Optimization* (BCO) has traditionally focused on minimizing the static regret, see the recent monograph [1] for a comprehensive historical overview. Both convex and strongly convex objective functions have attracted significant attention, beginning with the foundational work of [25] and further developed in subsequent studies such as [29–36]. Minimizing regret in non-stationary environments has only received attention more recently [7, 13, 14, 21], see also [1, Section 2.4] for an overview for this topic. Among these works, [7, 14] study $R^{\mathsf{dyn}}(T, \Delta)$, whereas [13, 21] focus on $R^{\mathsf{path}}(T, P)$. As we explained above (and formalize in Section 1.3), the switching regret $R^{\mathsf{swi}}(T, S)$ can induce guarantees on both $R^{\mathsf{dyn}}(T, \Delta)$ and $R^{\mathsf{path}}(T, P)$, but the reverse does not necessarily hold. Therefore, the results in these works cannot be readily extended to provide regret guarantees w.r.t. all three measures $S, \Delta$ and $P$.

Minimizing regret in environments with non-stationarity measures such as $S, \Delta$ and $P$ have been addressed with greater depth in *Online Convex Optimization* (OCO), where the learner has direct access to gradient information and can query the gradient or function value at multiple points of the loss function per round. The state-of-the-art algorithm with optimal adaptive regret guarantees is MetaGrad with sleeping experts [24], which queries only one gradient per round, and adapts to curvature information of the loss function such as strong-convexity when available. Our polynomial-time algorithm TEWA-SE builds upon [24] and its precursors [22, 23], adapting this approach to BCO by replacing the exact gradient per round with an approximate gradient estimate, and by designing a quadratic surrogate loss. The approach in [24] follows a long line of successive developments in OCO from expert tracking methods [15, 20, 37–41] to the study of adaptive regret [18, 19, 42–46], with recent advances [24, 47–50] reducing the query complexity from $\mathcal{O}(\log T)$ to $\mathcal{O}(1)$ per round, while achieving optimal adaptive regret or dynamic regret. The adaptivity of [24] directly inherits from MetaGrad [22] and its extension [23], which themselves build on earlier adaptive methods [51, 52].

For general convex functions, the approach of substituting a one-point gradient estimate for the exact gradient in each round of an OCO algorithm often yields suboptimal $T^{\frac{3}{4}}$ rates, both in static regret [25, 26] and dynamic regret [13, 21]; see also our Corollary 1. A series of breakthroughs [28, 32, 53–56] indicate that $\sqrt{T}$ rates (up to logarithms) are attainable for static regret, at the cost of a higher dependency on $d$. Our cExO algorithm follows this line of work, using exponential weights on a discretized action space [28]. By playing inside a clipped domain, we transform the algorithm from one with $\sqrt{T}$ static regret into one with $\sqrt{B}$ adaptive regret (modulo logarithms) for intervals of length $\leq$ B, which in turn leads to regret guarantees w.r.t. $S, \Delta$ and $P$.

Finally, we mention that non-stationarity has been widely studied in the *Multi-Armed Bandit* (MAB) literature. A substantial body of work has focused on adapting standard policies—such as UCB [57, 58], EXP3 [59], and Thompson Sampling [60–62]—to perform effectively under non-stationarity. These adaptations often employ mechanisms to discard outdated information, either *actively* (e.g., change-detection methods [12, 63–67]), or *passively* (e.g., discounted rewards [10, 68], sliding windows [69, 70], or scheduled restarts [11]), but are not straightforward to adapt to BCO.

## 1.3 Conversions between different regret definitions

We present the key conversions between different regret notions, illustrated in Figure 1 above. Using the definition of $R^{\text{dyn}}(T)$ in (3), we overload notation slightly to define

$$R^{\text{dyn}}(T, \Delta) := \sup_{f_{1:T}:\Delta(f_{1:T})\leq\Delta} \sum_{t=1}^{T} \mathbf{E}\Big[f_t(\boldsymbol{z}_t) - \min_{\boldsymbol{z}\in\Theta} f_t(\boldsymbol{z})\Big], \tag{8}$$

and $R^{\text{dyn}}(T, \Delta, S)$ additionally constrains $1 + \sum_{t=2}^{T} \min_{(\boldsymbol{z}_t^*, \boldsymbol{z}_{t-1}^*)\in(\mathcal{Z}_t^*, \mathcal{Z}_{t-1}^*)} \mathbf{1}(\boldsymbol{z}_t^* \neq \boldsymbol{z}_{t-1}^*) \leq S$ where $\mathcal{Z}_t^* := \arg\min_{\boldsymbol{z}\in\Theta} f_t(\boldsymbol{z})$ for all $t \in [T]$. In Proposition 1, we show how the adaptive regret $R^{\text{ada}}(\mathsf{B}, T)$ can be used to bound the switching regret $R^{\text{swi}}(T, S)$, which in turn can be used to bound the dynamic regret $R^{\text{dyn}}(T, \Delta, S)$ and path-length regret $R^{\text{path}}(T, P)$. Consequently, $R^{\text{ada}}(\mathsf{B}, T)$ and $R^{\text{swi}}(T, S)$ are the primary objects to analyze.

**Proposition 1.** *Suppose that an algorithm can be calibrated to satisfy $R^{\text{ada}}(\mathsf{B}, T) \leq C\mathsf{B}^\kappa$, for any interval length $\mathsf{B} \in [T]$, for some factor $C > 0$ that is at most polynomial in $d$ and $\log(T)$, and $\kappa \in [0, 1)$.*

*Then, for any $S, S_\Delta, S_P \in [T]$, an appropriate choice of $\mathsf{B}$ yields the following regret guarantees:*

**Switching:** $\mathsf{B} = \lceil\frac{T}{S}\rceil$ *guarantees that* $R^{\text{swi}}(T, S) \leq 2^{1+\kappa}CS^{1-\kappa}T^\kappa$.

**Dynamic:** $\mathsf{B} = \lceil\frac{T}{S}\rceil \vee \lceil\frac{T}{S_\Delta}\rceil$ *yields* $R^{\text{dyn}}(T, \Delta, S) \leq R^{\text{swi}}(T, S) \wedge \left(R^{\text{swi}}(T, S_\Delta) + \Delta\lceil\frac{T}{S_\Delta}\rceil\right)$.

**Path-length:** $\mathsf{B} = \lceil\frac{T}{S_P}\rceil$ *ensures that* $R^{\text{path}}(T, P) \leq R^{\text{swi}}(T, S_P) + \frac{P}{r}\cdot\lceil\frac{T}{S_P}\rceil$.

The proof is provided in Appendix B. We note that the reduction from $R^{\text{path}}(T, P)$ to $R^{\text{swi}}(T, S)$ in Proposition 1 is new and employs simple geometric arguments (see Lemma 2 in Appendix B). This reduction simplifies the analysis of $R^{\text{path}}(T, P)$, though it can yield slightly looser bounds on $R^{\text{path}}(T, P)$ than a direct analysis, as discussed in [17].

## 2 The TEWA-SE algorithm

In this section, we develop a polynomial-time algorithm called Tilted Exponentially Weighted Average with Sleeping Experts (TEWA-SE, Algorithm 1), building on the two-layer structure of previous experts-based algorithms [18, 19, 43]. Each expert in TEWA-SE is uniquely defined by its lifetime and learning rate. We denote the active experts at time $t$ by $E_1, E_2, \ldots, E_{n_t}$, where $E_i$ operates over interval $I_i$ with learning rate $\eta_i$. In each round $t$, the active experts each propose an action, denoted by $\boldsymbol{x}_{t,I_i}^{\eta_i}$, and a meta-algorithm aggregates them into a single meta-action $\boldsymbol{x}_t$ by computing their tilted exponentially weighted average [22, 24], see line 7 in the pseudo-code. Then the algorithm receives a noisy evaluation of $f_t$ at $\boldsymbol{x}_t$ and constructs an approximate *gradient estimate* $\boldsymbol{g}_t \in \mathbb{R}^d$ of $f_t$ at $\boldsymbol{x}_t$. Both $\boldsymbol{x}_t$ and $\boldsymbol{g}_t$ are shared with all experts, who update their actions via online gradient descent on their *surrogate* loss functions defined using $\boldsymbol{x}_t$ and $\boldsymbol{g}_t$.

TEWA-SE employs the Geometric Covering scheme from [19, 24] to schedule experts across different time intervals, and the exponential grid from [22, 24] to assign varied learning rates to the experts. These deterministic schemes ensure that only a *logarithmic* number of experts are active per round, maintaining computational efficiency. Intuitively, the meta-algorithm achieves low adaptive regret on the original loss function because, for each subinterval of times, there exists at least one individual expert with low static regret on their surrogate loss functions on this subinterval. This is guaranteed by the careful design of the exponential grid of learning rates. While full details of TEWA-SE is deferred to Appendix C.1, we highlight below the distinctions between this paper and prior works.

**Construction of one-point gradient estimate** For a fixed parameter $h \in (0, r)$, we define the clipped domain $\tilde{\Theta} = \{\boldsymbol{u} \in \Theta : \boldsymbol{u} + h\mathbb{B}^d \subset \Theta\}$, where $h < r$ ensures $\tilde{\Theta} \neq \emptyset$. In each round $t$, we select a meta-action $\boldsymbol{x}_t \in \tilde{\Theta}$ and query the function at a perturbed point $\boldsymbol{x}_t + h\boldsymbol{\zeta}_t$, receiving noisy feedback $y_t = f_t(\boldsymbol{x}_t + h\boldsymbol{\zeta}_t) + \xi_t$, where $\boldsymbol{\zeta}_t \in \mathbb{R}^d$ is sampled uniformly from the unit sphere $\partial\mathbb{B}^d$. This allows us to construct the gradient estimate $\boldsymbol{g}_t = (d/h)y_t\boldsymbol{\zeta}_t$. As implied by [25, Lemma 1], the

---

**Algorithm 1** Tilted Exponentially Weighted Average with Sleeping Experts (TEWA-SE)

---

**Input:** $d, T, \mathsf{B}, h = \min\left(\sqrt{d}\mathsf{B}^{-\frac{1}{4}}, r\right), \tilde{\Theta} = \{\boldsymbol{u} \in \Theta : \boldsymbol{u} + h\mathbb{B}^d \subset \Theta\}$, $G$ as in (10), expert algorithm $E(I, \eta)$ defined in Algorithm 2, and $(n_t)_{t \in [T]}$ and $(I_i, \eta_i)_{i \in [n_t]}$ $\forall t \in [T]$

1: **for** $t = 1, 2, \ldots, T$ **do**
2:     **for** $E_i \equiv E_i(I_i, \eta_i) \in \{E_1, E_2, \ldots, E_{n_t}\}$ **do**             $\triangleright$ $n_t$ experts active at $t$
3:         Receive action $\boldsymbol{x}_{t,I_i}^{\eta_i}$ from expert $E_i$
4:         **if** $\min\{\tau : \tau \in I_i\} = t$ **then** initialize $L_{t-1,I_i}^{\eta_i} = 0$, clipped domain $\tilde{\Theta}$ and parameter $G$
5:         **end if**
6:     **end for**
7:     Set meta-action as $\boldsymbol{x}_t = \sum_{i=1}^{n_t} \eta_i \exp(-L_{t-1,I_i}^{\eta_i})\boldsymbol{x}_{t,I_i}^{\eta_i} / \sum_{j=1}^{n_t} \eta_j \exp(-L_{t-1,I_j}^{\eta_j})$
8:     Sample $\boldsymbol{\zeta}_t$ uniformly from $\partial\mathbb{B}^d$
9:     Query point $\boldsymbol{z}_t = \boldsymbol{x}_t + h\boldsymbol{\zeta}_t$ to obtain $y_t = f_t(\boldsymbol{z}_t) + \xi_t$
10:     Construct gradient estimate $\boldsymbol{g}_t = (d/h)y_t\boldsymbol{\zeta}_t$
11:     **for** $i = 1, 2, \ldots, n_t$ **do**
12:         Send meta-action $\boldsymbol{x}_t$ and $\boldsymbol{g}_t$ to $E_i$
13:         Increment cumulative loss $L_{t,I_i}^{\eta_i} = L_{t-1,I_i}^{\eta_i} + \ell_t^{\eta_i}(\boldsymbol{x}_{t,I_i}^{\eta_i})$     $\triangleright$ $\ell_t^{\eta_i}(\cdot)$ depends on $\boldsymbol{x}_t$ and $\boldsymbol{g}_t$
14:     **end for**
15: **end for**

---

vector $\boldsymbol{g}_t$ is an unbiased gradient estimate of a spherically smoothed version of $f_t$ at $\boldsymbol{x}_t$, satisfying

$$\mathbf{E}[\boldsymbol{g}_t|\boldsymbol{x}_t] = \nabla\hat{f}_t(\boldsymbol{x}_t), \quad \text{where} \quad \hat{f}_t(\boldsymbol{x}) = \mathbf{E}\left[f_t(\boldsymbol{x} + h\tilde{\boldsymbol{\zeta}})\right] \ \forall \boldsymbol{x} \in \tilde{\Theta}, \tag{9}$$

with $\tilde{\boldsymbol{\zeta}}$ distributed uniformly on the unit ball $\mathbb{B}^d$. Importantly, $\hat{f}_t$ inherits the convexity properties of $f_t$ [71, Lemmas A.2–A.3]. Our approach differs from related works in OCO [22–24, 47, 48] that use exact gradients in two key ways: i) in each round, we query the perturbed point $\boldsymbol{z}_t = \boldsymbol{x}_t + h\boldsymbol{\zeta}_t$ rather than $\boldsymbol{x}_t$, accumulating regret at the perturbed point, and ii) we constrain $\boldsymbol{x}_t$ inside the clipped domain $\tilde{\Theta}$ to ensure all perturbed $\boldsymbol{z}_t$ remain feasible.

In our setting, under the high probability event $\Lambda_T = \left\{|\xi_t| \le 2\sigma\sqrt{\log(T+1)}, \ \forall t \in [T]\right\}$, we have

$$\|\boldsymbol{g}_t\| = (d/h)|f_t(\boldsymbol{x}_t + h\boldsymbol{\zeta}_t) + \xi_t| \le (d/h)\left(1 + 2\sigma\sqrt{\log(T+1)}\right) =: G, \quad \forall t \in [T]. \tag{10}$$

This implies a fundamental tradeoff in selecting the smoothing (and clipping) parameter $h$: larger values reduce $G$ (and the variance of $\boldsymbol{g}_t$), but increase both the approximation error between $\hat{f}_t$ and $f_t$ and the error due to clipping, while smaller values reduce bias at the cost of a higher variance in $\boldsymbol{g}_t$. In Theorem 1 and Corollary 1, we establish the optimal $h$ and the resulting regret guarantees.

---

**Algorithm 2** Expert algorithm $E(I, \eta)$: projected online gradient descent (OGD)

---

**Input:** $I = [r, s]$, $\eta$, $G$, clipped domain $\tilde{\Theta}$, and surrogate loss $\ell_t^\eta(\cdot)$ defined in (11) $\forall t \in \mathbb{N}^+$
**Initialize:** $\boldsymbol{x}_{r,I}^\eta$ be any point in $\tilde{\Theta}$

1: **for** $t = r, r + 1, \ldots, s$ **do**
2:     Send action $\boldsymbol{x}_{t,I}^\eta$ to Algorithm 1
3:     Receive meta-action $\boldsymbol{x}_t$ and $\boldsymbol{g}_t$ from Algorithm 1
4:     Update $\boldsymbol{x}_{t+1,I}^\eta = \Pi_{\tilde{\Theta}}\left(\boldsymbol{x}_{t,I}^\eta - \mu_t\nabla\ell_t^\eta(\boldsymbol{x}_{t,I}^\eta)\right)$, where $\mu_t = 1/(2\eta^2 G^2(t - r + 1))$
5: **end for**

---

**Design of expert algorithms and surrogate losses** We choose projected online gradient descent (OGD) as the expert algorithms (Algorithm 2), i.e., each expert $E(I, \eta)$ runs OGD during its lifetime $I$. In the full-information setting, where experts observe $f_t$ and gradients are evaluated at all of their actions, each expert could simply run OGD on the true loss functions. In contrast, for the bandit setting, with only one gradient estimate $\boldsymbol{g}_t$ of the smoothed loss $\hat{f}_t$ per round, we need to construct surrogate losses for the experts. The simplest option is the linear surrogate loss $\ell_t(\boldsymbol{x}) = -\boldsymbol{g}_t^\top(\boldsymbol{x}_t - \boldsymbol{x})$, but this fails to leverage curvature information and leads to a large $\widetilde{\mathcal{O}}(\sqrt{|I|})$ static regret for each expert, ultimately yielding linear adaptive regret.

To address these limitations, inspired by [22–24], we design the following strongly-convex surrogate loss $\ell_t^\eta : \mathbb{R}^d \to \mathbb{R}$:

$$\ell_t^\eta(\boldsymbol{x}) = -\eta \boldsymbol{g}_t^\top (\boldsymbol{x}_t - \boldsymbol{x}) + \eta^2 G^2 \|\boldsymbol{x}_t - \boldsymbol{x}\|^2, \quad \forall \boldsymbol{x} \in \mathbb{R}^d, \tag{11}$$

where $G$ is the upper bound (10) on $\|\boldsymbol{g}_t\|$, and $\eta$ is the learning rate of the expert. We highlight that our choice of the quadratic term in (11) differs from the $\eta^2\|\boldsymbol{g}_t\|^2\|\boldsymbol{x}_t - \boldsymbol{x}\|^2$ and $\eta^2(\boldsymbol{g}_t^\top(\boldsymbol{x}_t - \boldsymbol{x}))^2$ in [24] and [22]. The latter necessitates an additional condition relating $\mathbf{E}[\|\boldsymbol{g}_t\|]$ and $\mathbf{E}[\|\boldsymbol{g}_t\|^2]$ (or $\mathbf{E}[\boldsymbol{g}_t\boldsymbol{g}_t^\top]$) to be satisfied in the analysis, see e.g., [22, Theorem 2], and may yield suboptimal rates in dimension $d$ for strongly-convex losses, similar to [22]. Our choice of the quadratic term, similar to [23], eliminates these limitations and simplifies the proof.

For a comparator $\boldsymbol{u} \in \Theta$, (11) implies that the linearized regret associated with $\hat{f}_t$ on interval $I$ can be bounded as:

$$\sum_{t \in I} \langle \mathbf{E}[\boldsymbol{g}_t | \boldsymbol{x}_t, \Lambda_T], \boldsymbol{x}_t - \boldsymbol{u} \rangle \le \frac{1}{\eta} \underbrace{\sum_{t \in I} \mathbf{E}\left[\ell_t^\eta(\boldsymbol{x}_t) - \ell_t^\eta(\boldsymbol{u}) \mid \boldsymbol{x}_t, \Lambda_T\right]}_{:=\mathsf{A}} + \eta G^2 \sum_{t \in I} \|\boldsymbol{x}_t - \boldsymbol{u}\|^2. \tag{12}$$

Due to the strong-convexity of $\ell_t^\eta$, each expert attains only an $\mathcal{O}(\log|I|)$ static regret under OGD with an optimally tuned step size $\mu_t$ (see line 4 of Algorithm 2, and Lemma 6 in Appendix C.4). This ensures term $\mathsf{A}$ above is also of $\mathcal{O}(\log|I|)$. By the convexity of $\hat{f}_t$ we have

$$\sum_{t \in I} \mathbf{E}\left[\hat{f}_t(\boldsymbol{x}_t) - \hat{f}_t(\boldsymbol{u}) \mid \Lambda_T\right] \le \mathbf{E}\left[\frac{1}{\eta}\mathsf{A} + (\eta G^2 - \frac{\alpha}{2}) \sum_{t \in I} \|\boldsymbol{x}_t - \boldsymbol{u}\|^2 \mid \Lambda_T\right], \tag{13}$$

where $\alpha = 0$ for general convex $\hat{f}_t$ (and $f_t$), and $\alpha > 0$ for strongly-convex. Since both $\alpha$ and $\sum_{t \in I}\|\boldsymbol{x}_t - \boldsymbol{u}\|^2$ are unknown a priori, we use a deterministic exponential grid of $\eta$ values [19, 24], ensuring at least one expert covering $I$ effectively minimize the RHS of (13), ultimately yielding a sublinear adaptive regret w.r.t. $f_t$. We present this result in the following theorem.

**Theorem 1.** *For any $T \in \mathbb{N}^+$ and $\mathsf{B} \in [T]$, Algorithm 1 with $h = \min(\sqrt{d}\mathsf{B}^{-\frac{1}{4}}, r)$ satisfies*

$$R^{ada}(\mathsf{B}, T) \lesssim \sqrt{d}\mathsf{B}^{\frac{3}{4}} + d^2, \tag{14}$$

*and if $f_t$ is $\alpha$-strongly-convex with $\arg\min_{\boldsymbol{x} \in \mathbb{R}^d} f_t(\boldsymbol{x}) \in \Theta$ for all $t \in [T]$,[1] it furthermore holds that*

$$R^{ada}(\mathsf{B}, T) \lesssim \frac{d}{\alpha}\sqrt{\mathsf{B}} + \frac{1}{\alpha}d^2, \tag{15}$$

*where $\lesssim$ conceals polylogarithmic terms in $\mathsf{B}$ and $T$, independent of $d$ and $\alpha$.*

The proof of Theorem 1 can be found in Appendix C.2. We emphasize that TEWA-SE does not require knowledge of the strong-convexity parameter $\alpha$. This parameter is only used in the analysis and appear in the upper bound (15). Compared to the $\mathcal{O}(\sqrt{\mathsf{B}\log T})$ and $\mathcal{O}(\frac{1}{\alpha}\log T \log \mathsf{B})$ adaptive regrets in [24] for general convex and strongly-convex losses respectively, our bounds in Theorem 1 reflect the separation between online first-order and zeroth-order optimization. This mirrors the established gap in static regret analyses, see e.g. [74] vs. [72]. We further note that our bound for the strongly-convex case has a $\frac{1}{\alpha}$ dependency, which is suboptimal compared to the $\frac{1}{\sqrt{\alpha}}$ dependency in [33, 73] for static regret in BCO for $\alpha \lesssim 1$.

Applying Proposition 1, the adaptive regret bounds in Theorem 1 lead to the following bounds for $R^{swi}(T, S)$, $R^{dyn}(T, \Delta, S)$ and $R^{path}(T, P)$. In Corollary 1, for clarity we drop the $\lceil \cdot \rceil$ operators from the expressions for $\mathsf{B}$ and assume without loss of generality $\mathsf{B}$ is an integer (proof in Appendix C.5).

**Corollary 1.** *Consider any horizon $T \in \mathbb{N}^+$ and assume that, for all $t \in [T]$, the loss $f_t$ is convex, or strongly-convex with $\arg\min_{\boldsymbol{x} \in \mathbb{R}^d} f_t(\boldsymbol{x}) \in \Theta$. We refer to the second scenario as the strongly-convex (SC) case. Then, Algorithm 1 tuned with parameter $\mathsf{B}$ satisfies the following regret guarantees:*

---

[1]The assumption that loss minimizers lie inside $\Theta$ is common in zeroth-order optimization, see e.g., [7, 72, 73]. Without it, our upper bound analysis would have an extra term depending on the gradients at the minimizers.

**Switching.** $\quad \mathsf{B} = \frac{T}{S} \implies R^{swi}(T,S) \lesssim \begin{cases} \sqrt{d}S^{\frac{1}{4}}T^{\frac{3}{4}} + d^2 S \\ d\sqrt{ST} + d^2 S \quad (SC) \end{cases}$

**Dynamic.** $\begin{cases} \mathsf{B} = \frac{T}{S} \vee \left(\frac{\sqrt{d}T}{\Delta}\right)^{\frac{4}{5}} \Rightarrow R^{dyn}(T,\Delta,S) \lesssim R^{swi}(T,S) \wedge (d^{\frac{2}{5}}\Delta^{\frac{1}{5}}T^{\frac{4}{5}} + d^{\frac{8}{5}}\Delta^{\frac{4}{5}}T^{\frac{1}{5}}) \\ \mathsf{B} = \frac{T}{S} \vee \left(\frac{dT}{\Delta}\right)^{\frac{2}{3}} \Rightarrow R^{dyn}(T,\Delta,S) \lesssim R^{swi}(T,S) \wedge (d^{\frac{2}{3}}\Delta^{\frac{1}{3}}T^{\frac{2}{3}} + d^{\frac{4}{3}}\Delta^{\frac{2}{3}}T^{\frac{1}{3}}) \quad (SC) \end{cases}$

**Path-length.** $\begin{cases} \mathsf{B} = \left(\frac{r\sqrt{d}T}{P}\right)^{\frac{4}{5}} \Rightarrow R^{path}(T,P) \lesssim r^{-\frac{1}{5}}d^{\frac{2}{5}}P^{\frac{1}{5}}T^{\frac{4}{5}} + r^{-\frac{4}{5}}d^{\frac{8}{5}}P^{\frac{4}{5}}T^{\frac{1}{5}} \\ \mathsf{B} = \left(\frac{rdT}{P}\right)^{\frac{2}{3}} \Rightarrow R^{path}(T,P) \lesssim r^{-\frac{1}{3}}d^{\frac{2}{3}}P^{\frac{1}{3}}T^{\frac{2}{3}} + r^{-\frac{2}{3}}d^{\frac{4}{3}}P^{\frac{2}{3}}T^{\frac{1}{3}} \quad (SC) \, . \end{cases}$

## 2.1 Lower bound for strongly-convex loss functions

In this section, we derive a minimax lower bound on the dynamic regret and path-length regret, and discuss the optimality of TEWA-SE. To derive the lower bound for the dynamic regret, we adopt a standard minimax approach by constructing a class of hard functions, following [71, Theorem 6.1]. We assume that the adversary either (i) partitions the time horizon into $S$ segments and assigns a different function from this class to each segment, or (ii) selects a sequence of functions with total variation bounded by $\Delta$.

**Theorem 2.** *Let $\Theta = \mathbb{B}^d$. For $\alpha > 0$ denote by $\mathcal{F}_\alpha$ the class of $\alpha$-strongly convex and smooth functions. Let $\pi = \{z_t\}_{t=1}^T$ be any randomized algorithm (see Appendix D for a definition). Then there exists $T_0 > 0$ such that for all $T \geq T_0$ it holds that*

$$\sup_{f_1,\dots,f_T \in \mathcal{F}_\alpha} R^{dyn}(T,\Delta,S) \geq c_1 \cdot \left(d\sqrt{ST} \wedge d^{\frac{2}{3}}\Delta^{\frac{1}{3}}T^{\frac{2}{3}}\right) , \tag{16}$$

*where $c_1 > 0$ is a constant independent of $d, T, S$ and $\Delta$.*

We detail the proof in Appendix D. This lower bound establishes that TEWA-SE achieves the minimax-optimal dynamic regret (up to logarithms) for strongly convex and smooth functions w.r.t. $d$, $T$, $S$ and $\Delta$. We note that [7] derives a lower bound only in terms of $T$ and $\Delta$, matching (16), but it does not explicitly capture the dependence on $d$ nor does it address the interplay between $S$ and $\Delta$. In the special case where $S = 1$, Theorem 2 recovers the classical minimax static regret of order $d\sqrt{T}$ [71, 72]. Interestingly, for $d = 1$ the scaling of the lower bound as function of $T, S$ and $\Delta$ is the same as standard lower bounds in the non-stationary MAB literature [10, 11]. The proof of Theorem 2 can be readily adapted to consider only the measure $S$ with the switching regret, yielding a rate of $d\sqrt{ST}$ and thereby establishing the minimax optimality of TEWA-SE's switching regret bound.

We also derive a lower bound for path-length regret analogously to that for dynamic regret. In Theorem 4 in Appendix D we show that under the same assumptions as in the statement of Theorem 2,

$$\sup_{f_1,\dots,f_T \in \mathcal{F}_\alpha} R^{path}(T,P) \geq c_2 \cdot (d^2 P)^{\frac{2}{5}}T^{\frac{3}{5}} , \tag{17}$$

where $c_2 > 0$ is a constant independent of $d$, $T$ and $P$. Hence, TEWA-SE may not achieve the optimal regret rate for path-length. Additionally, Eq. (17) improves upon the only existing $d\sqrt{PT}$ lower bound from [13] in terms of the horizon $T$, by leveraging a different construction of a hard instance. This improvement comes from assuming $P = o(T)$, which is necessary for sublinear regret.

## 2.2 Parameter-free guarantees

In Corollary 1, we showed that the knowledge of the non-stationarity measures $S, \Delta$ and $P$ allows optimal tuning of TEWA-SE's parameter B. However, these measures can be hard to estimate. To obtain guarantees without such knowledge, we further analyze TEWA-SE under the *Bandit-over-Bandit* (BoB) framework from [27] (see Appendix C.6 for details), which divides the time horizon into epochs of suitable length $L$ and uses an adversarial bandit algorithm (e.g., EXP3) to select B for TEWA-SE in each epoch from the set $\mathcal{B} = \{2^i : i = 0, 1, \dots, \lfloor \log_2 T \rfloor\}$. In Corollary 3 in Appendix C.6, we adapt all the upper bounds in Corollary 1 to this framework, and show that this procedure costs an additional $d^{\frac{1}{3}}T^{\frac{5}{6}}$ term for the general convex case and $d^{\frac{1}{2}}T^{\frac{3}{4}}$ for the strongly-convex case. Our parameter-free path-length regret bound $P^{\frac{1}{5}}T^{\frac{4}{5}} + T^{\frac{5}{6}}$ for the general convex case improves on the $P^{\frac{1}{2}}T^{\frac{3}{4}}$ bound in [13] when $P = \Omega(T^{\frac{1}{6}})$.

Recent works on MAB [65–67, 75, 76] have proposed algorithms that achieve optimal dynamic regret without prior knowledge of $S$ and $\Delta$. However, they use procedures that crucially rely on the finiteness of the arm set, and are thus ill-suited for BCO. It remains open to determine if the minimax regret rate can be attained without such knowledge in the settings considered in this paper.

# 3 Clipped Exploration by Optimization

In this section, we propose a second algorithm (Algorithm 3) to improve upon the suboptimal rates for $R^{\text{dyn}}(T, \Delta, S)$ and $R^{\text{path}}(T, P)$ that TEWA-SE achieves for general convex loss functions. For ease of presentation, we assume in this section that the problem is noiseless, i.e., $\xi_t = 0$ for $t \in [T]$. We call this algorithm clipped Exploration by Optimization (cExO), which is built on Algorithm 8.3 (ExO) in [1]. The high level idea of ExO is to run exponential weights over a finite discretization of the feasible set, denoted by $\mathcal{C} \subset \Theta$. We assume the discretization $\mathcal{C}$ admits a worst-case error of $\varepsilon := \sup_{f \in \mathcal{F}_0} \min_{q \in \Delta(\mathcal{C})} \mathbf{E}_{z' \sim q} f(z') - \min_{z \in \Theta} f(z)$, where $\mathcal{F}_0$ denotes the class of convex and Lipschitz functions, and $\Delta(\mathcal{C})$ denotes the $(|\mathcal{C}| - 1)$-dimensional simplex.

With $q_0$ initialized as the uniform distribution, in each round $t$, given a loss estimate $\widehat{s}_t \in \mathbb{R}^{|\mathcal{C}|}$, ExO (in its mirror descent formulation) computes $q_t = \arg\min_{q \in \Delta(\mathcal{C})} \langle q, \widehat{s}_{t-1} \rangle + \frac{1}{\eta} \text{KL}(q \| q_{t-1})$, where KL is the Kullback-Leibler divergence $\text{KL}(q \| p) = \sum_{i=1}^{|\mathcal{C}|} q_i \log(q_i / p_i)$ for $q, p \in \Delta(\mathcal{C})$. The update rule in cExO departs from the vanilla ExO in this single step, by taking the minimum over the *clipped* simplex $\tilde{\Delta} = \Delta(\mathcal{C}) \cap [\gamma, 1]^{|\mathcal{C}|}$ where $\gamma \in (0, \frac{1}{|\mathcal{C}|})$ is a constant to be tuned, see line 2 of Algorithm 3. Clipping is a standard technique in mirror descent to ensure the algorithm does not commit too hard to any single action, and therefore detect changes in the environments more easily, yielding regret guarantees w.r.t. non-stationary measures [9, Chapter 31.1].

Given the reference distribution $q_t$, cExO selects a playing distribution $p_t \in \Delta(\mathcal{C})$ and an estimator function $E_t \in \mathcal{E}$ which returns an updated loss estimate for each action in $\mathcal{C}$, where $\mathcal{E}$ denotes the set of functions that map $\mathcal{C} \times [-1, 1]$ to $\mathbb{R}^{|\mathcal{C}|}$. It does so by solving an intractable optimization problem:[2]

$$\underset{p \in \Delta(\mathcal{C}), E \in \mathcal{E}}{\arg\min} \ \Lambda_\eta(q_t, p, E), \tag{18}$$

where, with $S_q(\eta \hat{s}) = \max_{q' \in \Delta(\mathcal{C})} \langle q - q', \eta \hat{s} \rangle - \text{KL}(q' \| q)$, the objective function is defined by

$$\Lambda_\eta(q, p, E) := \sup_{p^\star \in \Delta(\mathcal{C})} \sup_{f \in \mathcal{F}_0} \mathbf{E}_{z \sim p} \Big[ \langle p - p^\star, f \rangle + \langle p^\star - q, E(z, f(z)) \rangle + \frac{1}{\eta} S_q(\eta E(z, f(z))) \Big].$$

This optimization problem is intractable due to the large size of $\mathcal{E}$ and $\mathcal{F}_0$.[3] The role of this optimization problem is to tradeoff the worst-case cost of deviating from the desired distribution $q_t$ versus the gain of improved exploration (hence the name Exploration by Optimization). Finally, cExO samples an action $z_t$ according to $p_t$, observes the feedback $f(z_t)$ and constructs a loss estimate $\widehat{s}_t = E_t(z_t, f(z_t))$ to be used in the subsequent round. cExO achieves the adaptive regret guarantee stated in Theorem 3 below.

**Theorem 3.** *For $T \in \mathbb{N}^+$ and $\mathsf{B} \in [T]$, Algorithm 3 calibrated with $\varepsilon = \frac{1}{T}$, $\gamma = \frac{1}{T|\mathcal{C}|}$, $\eta = \sqrt{\log(\gamma^{-1})/(d^4 \log(dT)\mathsf{B})}$ and $\log |\mathcal{C}| = \mathcal{O}(d \log(dT^2))$ satisfies*

$$R^{\text{ada}}(\mathsf{B}, T) \lesssim d^{\frac{5}{2}} \sqrt{\mathsf{B}}. \tag{19}$$

We then use Proposition 1 to convert the bound of Theorem 3 into the following regret guarantees w.r.t. $S, \Delta$ and $P$. Like in Corollary 1, we omit $\lceil \cdot \rceil$ from the expressions for $\mathsf{B}$ for clarity.

**Corollary 2.** *For any horizon $T \in \mathbb{N}^+$, Algorithm 3 calibrated as in Theorem 3 and tuned with interval size $\mathsf{B}$ (which determines $\eta$) satisfies the following regret guarantees:*

> ***Switching:*** $\mathsf{B} = \frac{T}{S} \implies R^{\text{swi}}(T, S) \lesssim d^{\frac{5}{2}} \sqrt{ST}$,
>
> ***Dynamic:*** $\mathsf{B} = \frac{T}{S} \vee (d^{\frac{5}{2}} T/\Delta)^{\frac{2}{3}} \implies R^{\text{dyn}}(T, \Delta, S) \lesssim R^{\text{swi}}(T, S) \wedge d^{\frac{5}{3}} \Delta^{\frac{1}{3}} T^{\frac{2}{3}}$,
>
> ***Path-length:*** $\mathsf{B} = (rd^{\frac{5}{2}} T/P)^{\frac{2}{3}} \implies R^{\text{path}}(T, P) \lesssim r^{-\frac{1}{3}} d^{\frac{5}{3}} P^{\frac{1}{3}} T^{\frac{2}{3}}$.

---

[2]For detailed discussions about these functions, we refer the reader to [28].

[3]One can in theory bound the domain of $\mathcal{E}$ and discretize $\mathcal{E}$, $\mathcal{F}_0$ and $\Delta(\mathcal{C})$. The optimization problem is hence computable, though not in polynomial time.

**Algorithm 3** clipped Exploration by Optimization (cExO)

---

**Input:** $d, T, \mathrm{B}$, feasible set $\Theta$, a finite covering set $\mathcal{C} \subset \Theta$ of $\Theta$, discretization error $\varepsilon$, learning rate $\eta$, clipping parameter $\gamma \in (0, \frac{1}{|\mathcal{C}|})$, and $\tilde{\Delta} = \Delta(\mathcal{C}) \cap [\gamma, 1]^{|\mathcal{C}|}$

**Initialize:** $q_{0,i} = \frac{1}{|\mathcal{C}|} \ \forall i \in [|\mathcal{C}|]$.

1: **for** $t = 1, \dots, T$ **do**
2:      Compute $\boldsymbol{q}_t = \arg\min_{\boldsymbol{q} \in \tilde{\Delta}} \langle \boldsymbol{q}, \widehat{\boldsymbol{s}}_{t-1} \rangle + \frac{1}{\eta}\mathrm{KL}(\boldsymbol{q}||\boldsymbol{q}_{t-1})$
3:      Find distribution $\boldsymbol{p}_t \in \Delta(\mathcal{C})$ and $E_t \in \mathcal{E}$ s.t. $\Lambda_\eta(\boldsymbol{q}_t, \boldsymbol{p}_t, E_t) \le \inf_{\substack{\boldsymbol{p} \in \Delta(\mathcal{C}), \\ E \in \mathcal{E}}} \Lambda_\eta(\boldsymbol{q}_t, \boldsymbol{p}, E) + \eta d$
4:      Sample $\boldsymbol{z}_t \sim \boldsymbol{p}_t$ and observe $f_t(\boldsymbol{z}_t)$
5:      Compute $\widehat{\boldsymbol{s}}_t = E_t(\boldsymbol{z}_t, f_t(\boldsymbol{z}_t))$
6: **end for**

---

The proofs of Theorem 3 and Corollary 2 are presented in Appendix E. By comparing these results to the lower bounds in Section 2.1, we obtain that for known $S, \Delta$ and $P$, cExO achieves minimax-optimal rates in $T, S$ and $\Delta$, but remains suboptimal in $d$ (for all measures), and potentially for the path-length bound (see Eq. (17)). This suboptimal dependence on $d$ is unsurprising since even the best known static regret bounds of [31] and [34] suffer from similar dimensional dependence. Moreover, the gap between cExO's path-length regret bound and our minimax lower bound of order $d^{\frac{4}{5}}P^{\frac{2}{5}}T^{\frac{3}{5}}$ may stem from either (i) looseness in the lower bound, or (ii) sub-optimality of cExO, which runs OMD in distribution space rather than directly on the action set. The latter may allow us to bound path-length regret more directly and sharply.

To adapt to unknown non-stationarity measures, cExO equipped with the BoB framework yields the upper bounds in Corollary 2 with an additional $d^{\frac{5}{4}}T^{\frac{3}{4}}$ term (see Corollary 4 in Appendix E). Our path-length regret of $P^{\frac{1}{3}}T^{\frac{2}{3}}$ and $P^{\frac{1}{3}}T^{\frac{2}{3}} + T^{\frac{3}{4}}$ for known and unknown $P$, respectively, improves on the $P^{\frac{1}{4}}T^{\frac{3}{4}}$ and $P^{\frac{1}{2}}T^{\frac{3}{4}}$ rates in [13] in terms of $T$.

## 4 Conclusion

In this work, we develop and analyze two approaches for non-stationary Bandit Convex Optimization. For strongly convex losses, our polynomial-time TEWA-SE algorithm achieves minimax-optimal dynamic regret w.r.t. $S$ and $\Delta$ without knowing the strong-convexity parameter, but incurs a sub-optimal $T^{\frac{3}{4}}$ rate for general convex losses. To address this, we propose a second algorithm, cExO, which achieves minimax-optimality for $S$ and $\Delta$. However, this algorithm is not polynomial-time computable and has an increased dimension dependence. Our matching lower bounds confirm the optimality results, but also reveal potentially suboptimal guarantees w.r.t. the path-length $P$. This work highlights a central open challenge: designing algorithms that are simultaneously minimax-optimal and computationally efficient for general convex losses in non-stationary environments. A promising stepstone towards this goal is to incorporate second-order information, akin to the online Newton methods from [31, 34] that achieve state-of-the-art static regret guarantees for adversarial convex bandits. In particular, a restart criterion, similar to the one in line 15 of [35, Algorithm 1] or line 11 of [31, Algorithm 1], may enable tracking capabilities and lead to improved regret bounds.

## Acknowledgments and Disclosure of Funding

X. Liu, D. Baudry, P. Rebeschini and A. Akhavan were funded by UK Research and Innovation (UKRI) under the UK government's Horizon Europe funding guarantee [grant number EP/Y028333/1].

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

## A Definitions

**Definition 1.** *Let $\sigma > 0$. A random variable $\xi$ is $\sigma$-sub-Gaussian if for any $\lambda > 0$ we have $\mathbf{E}[\exp(\lambda \xi)] \leq \exp(\sigma^2 \lambda^2 / 2)$.*

**Definition 2.** *Let $\alpha > 0$. A differentiable function $f : \mathbb{R}^d \to \mathbb{R}$ is called $\alpha$-strongly convex, if for $\boldsymbol{x}, \boldsymbol{z} \in \mathbb{R}^d$, $f(\boldsymbol{z}) \geq f(\boldsymbol{x}) + \nabla f(\boldsymbol{x})^\top (\boldsymbol{z} - \boldsymbol{x}) + \frac{\alpha}{2} \|\boldsymbol{z} - \boldsymbol{x}\|^2$.*

**Definition 3.** *Let $\beta > 0$. Function $f : \mathbb{R}^d \to \mathbb{R}$ is called $\beta$-smooth, if it is continuously differentiable and for any $\boldsymbol{x}, \boldsymbol{z} \in \mathbb{R}^d$, $\|\nabla f(\boldsymbol{x}) - \nabla f(\boldsymbol{z})\| \leq \beta \|\boldsymbol{x} - \boldsymbol{z}\|$.*

**Definition 4.** *Let $K > 0$. Function $f : \mathbb{R}^d \to \mathbb{R}$ is called $K$-Lipschitz if for any $\boldsymbol{x}, \boldsymbol{z} \in \mathbb{R}^d$, $|f(\boldsymbol{x}) - f(\boldsymbol{z})| \leq K \|\boldsymbol{x} - \boldsymbol{z}\|$.*

## B Proof of Proposition 1

We start this section by restating the proposition, before detailing its proof.

**Proposition 1.** *Suppose that an algorithm can be calibrated to satisfy $R^{ada}(\mathsf{B}, T) \leq C\mathsf{B}^\kappa$, for any interval length $\mathsf{B} \in [T]$, for some factor $C > 0$ that is at most polynomial in $d$ and $\log(T)$, and $\kappa \in [0, 1)$.*

*Then, for any $S, S_\Delta, S_P \in [T]$, an appropriate choice of $\mathsf{B}$ yields the following regret guarantees:*

**Switching:** $\mathsf{B} = \left\lceil \frac{T}{S} \right\rceil$ *guarantees that* $R^{swi}(T, S) \leq 2^{1+\kappa} C S^{1-\kappa} T^\kappa$.

**Dynamic:** $\mathsf{B} = \left\lceil \frac{T}{S} \right\rceil \vee \left\lceil \frac{T}{S_\Delta} \right\rceil$ *yields* $R^{dyn}(T, \Delta, S) \leq R^{swi}(T, S) \wedge \left( R^{swi}(T, S_\Delta) + \Delta \left\lceil \frac{T}{S_\Delta} \right\rceil \right)$.

**Path-length:** $\mathsf{B} = \left\lceil \frac{T}{S_P} \right\rceil$ *ensures that* $R^{path}(T, P) \leq R^{swi}(T, S_P) + \frac{P}{r} \cdot \left\lceil \frac{T}{S_P} \right\rceil$.

*Proof of Proposition 1.* The proof follows two steps. First, we state in Lemma 1 the conversion between adaptive regret and switching regret. A similar conversion can be found in [19], but we detail the proof for completeness. Next, we prove in Lemma 2 that switching regret guarantees for appropriate number of switches convert into dynamic and path-length regret guarantees. $\qquad\square$

In the remainder of this section, we detail the two supporting lemmas and their proof.

**Lemma 1.** *Consider an algorithm that satisfies the adaptive regret guarantees of Proposition 1, then this algorithm calibrated with interval size $\mathsf{B} = \left\lceil \frac{T}{S} \right\rceil$ satisfies*

$$R^{swi}(T, S) \leq 2^{1+\kappa} C S^{1-\kappa} T^\kappa.$$

*Proof of Lemma 1.* Consider $\mathsf{B} = \left\lceil \frac{T}{S} \right\rceil$. Let $\boldsymbol{u}_{1:T} \in \Theta^T$ be a sequence of arbitrary comparators with at most $S$ switches. We divide the horizon into intervals of length $\mathsf{B}$ (the last interval may be shorter than $\mathsf{B}$), and further divide the intervals at the rounds where $\boldsymbol{u}_t \neq \boldsymbol{u}_{t-1}$. This ensures each of these intervals is associated with a constant comparator. By construction, these intervals are of length $\leq \mathsf{B}$ and the number of intervals is bounded by $2S$. Hence, we can apply the adaptive regret bound to each interval to obtain

$$R^{swi}(T, S) \leq 2S \cdot C\mathsf{B}^\kappa \leq 2CS \cdot \left( \frac{T}{S} + 1 \right)^\kappa$$
$$= 2C \cdot S^{1-\kappa} T^\kappa \cdot \left( 1 + \frac{S}{T} \right)^\kappa \leq 2^{1+\kappa} C \cdot S^{1-\kappa} T^\kappa.$$

$\qquad\square$

We now prove the conversion between switching regret and dynamic and path-length regrets.

**Lemma 2.** *Consider any fictitious number of switches $S' \in [T]$. Then the dynamic regret of environments constrained by $\Delta$ satisfies*

$$R^{dyn}(T, \Delta) \leq R^{swi}(T, S') + \Delta \left\lceil \tfrac{T}{S'} \right\rceil, \tag{20}$$

*and the path-length regret satisfies*

$$R^{path}(T, P) \leq R^{swi}(T, S') + \tfrac{P}{r} \left\lceil \tfrac{T}{S'} \right\rceil. \tag{21}$$

*Proof of Lemma 2.* For both upper bounds, the switching regret term comes from dividing the horizon $[T]$ into $S'$ intervals, denoted by $(I_s)_{s \in [S']}$, each of length at most $\lceil \frac{T}{S'} \rceil$ (defining them precisely is not important for the following arguments). Recall the definition of $R(T, \boldsymbol{u}_{1:T})$ from (2). For any sequence of actions $\boldsymbol{z}_{1:T} \in \Theta^T$ chosen by the given algorithm, and for any arbitrary comparator sequences $\boldsymbol{u}_{1:T} \in \Theta^T$ and $\boldsymbol{v}_{1:S'} \in \Theta^{S'}$, it holds that

$$
\begin{aligned}
R(T, \boldsymbol{u}_{1:T}) &= \sum_{t=1}^{T} \mathbf{E}\left[f_t(\boldsymbol{z}_t) - f_t(\boldsymbol{u}_t)\right] \\
&= \sum_{t=1}^{T} \sum_{s=1}^{S'} \mathbf{1}(t \in I_s) \cdot \left(\mathbf{E}\left[f_t(\boldsymbol{z}_t) - f_t(\boldsymbol{v}_s)\right] + (f_t(\boldsymbol{v}_s) - f_t(\boldsymbol{u}_t))\right) \\
&\leq R^{\mathsf{swi}}(T, S') + \sum_{s=1}^{S'} \underbrace{\sum_{t \in I_s} (f_t(\boldsymbol{v}_s) - f_t(\boldsymbol{u}_t))}_{=:V_s}, \quad (22)
\end{aligned}
$$

where the last step holds by the definition of the switching regret. It thus remains to choose a suitable $\boldsymbol{v}_s \in \Theta$ and upper bound the term $V_s$ for $s \in [S']$. We choose a different $\boldsymbol{v}_s$ for the proof of the dynamic regret bound vs. that of the path-length regret bound.

**Dynamic regret.** Consider the interval $I_s$ for $s \in [S']$. Let $L_s$ be its length and $\Delta_s = \sum_{t \in I_s} \max_{\boldsymbol{z} \in \Theta} |f_t(\boldsymbol{z}) - f_{t-1}(\boldsymbol{z})|$ be the total variation over this interval. Then, for any two time steps $t$ and $t'$ in $I_s$ and any $\boldsymbol{z} \in \Theta$, it holds that $f_t(\boldsymbol{z}) - f_{t'}(\boldsymbol{z}) \leq \Delta_s$ by definition of total variation. Let $\bar{f}_s$ denote the average of the functions over the interval $I_s$ and define $\boldsymbol{v}_s \in \arg\min_{\boldsymbol{z} \in \Theta} \bar{f}_s(\boldsymbol{z})$, then we have

$$
\forall s \in [S'], \quad V_s \leq \sum_{t \in I_s} (f_t(\boldsymbol{v}_s) - \bar{f}_s(\boldsymbol{u}_t) + \Delta_s) \leq \Delta_s L_s.
$$

Taking the sum over all intervals and using $L_s \leq \lceil \frac{T}{S'} \rceil$ completes the proof of (20).

**Path-length regret.** This proof proceeds similarly as that for the dynamic regret. Consider the interval $I_s$ for some $s \in [S']$, and denote by $L_s$ its length and $P_s = \sum_{t \in I_s} \|\boldsymbol{u}_t - \boldsymbol{u}_{t-1}\|$ the path-length of the comparator sequence on this interval. For the proof, we construct $\boldsymbol{v}_s \in \Theta$ differently from that in the proof of the dynamic regret. Before detailing the construction of $\boldsymbol{v}_s$, we first define a set of comparators $(\boldsymbol{u}'_t)_{t \in I_s} \in \Theta^{L_s}$ as follows: for some $\alpha_0 \in [0, 1]$ and any time $t \in I_s$, we define $\boldsymbol{u}'_t$ to satisfy $\boldsymbol{v}_s = \alpha_0 \boldsymbol{u}_t + (1 - \alpha_0)\boldsymbol{u}'_t$. Using this and by the convexity and boundedness of $f_t$, we can bound that

$$
f_t(\boldsymbol{v}_s) \leq f_t(\boldsymbol{u}_t) + (1 - \alpha_0)(f_t(\boldsymbol{u}'_t) - f_t(\boldsymbol{u}_t)) \leq f_t(\boldsymbol{u}_t) + 2(1 - \alpha_0).
$$

We then proceed by choosing a suitable $\boldsymbol{v}_s$ and $\alpha_0$ to make this bound depend on the path-length. Since the path-length is $P_s$, there exists an $\ell_2$-ball of radius $\frac{P_s}{2}$ that contains all the comparators $(\boldsymbol{u}_t)_{t \in I_s}$, and its center $\boldsymbol{c}_u$ lies in the feasible domain $\Theta$. By assumption (as we stated in Section 1), there also exists a ball with radius $r$ and center $\boldsymbol{c}_r$ within the domain. We can thus construct $\boldsymbol{v}_s$ to satisfy

$$
\boldsymbol{v}_s = \alpha_0 \boldsymbol{c}_u + (1 - \alpha_0)\boldsymbol{c}_r \in \Theta, \qquad \text{which yields} \quad \boldsymbol{u}'_t = \boldsymbol{c}_r + \frac{\alpha_0}{1 - \alpha_0}(\boldsymbol{c}_u - \boldsymbol{u}_t), \quad (23)
$$

where $\boldsymbol{v}_s \in \Theta$ due to the convexity of the domain. Our goal is then to choose $\alpha_0$ as large as possible (to make $1 - \alpha_0$ small) such that all the comparators $(\boldsymbol{u}'_t)_{t \in I_s}$ belong to $\Theta$. Eq. (23) implies that

$$
\|\boldsymbol{u}'_t - \boldsymbol{c}_r\| = \frac{\alpha_0}{1 - \alpha_0}\|\boldsymbol{c}_u - \boldsymbol{u}_t\| \leq \frac{\alpha_0}{1 - \alpha_0} \cdot \frac{P_s}{2} = \frac{\alpha_0 P_s}{2(1 - \alpha_0)},
$$

which by definition of the $r$-ball guarantees that $\boldsymbol{u}'_t \in \Theta$ as long as $\frac{\alpha_0 P_s}{2(1 - \alpha_0)} \leq r$. To satisfy this condition, we can thus pick $\alpha_0 = \frac{2r}{P_s + 2r}$, which guarantees by construction that

$$
\forall t \in I_s: \quad f_t(\boldsymbol{v}_s) \leq f_t(\boldsymbol{u}_t) + \frac{2P_s}{P_s + 2r} \leq f_t(\boldsymbol{u}_t) + \frac{P_s}{r}.
$$

The desired bound on $V_s$ in (22) directly follows. The final result (21) then comes by summation over all intervals $(I_s)_{s \in [S']}$. $\qquad \square$

# C   Details and proofs for TEWA-SE

In this appendix, we provide additional details on TEWA-SE in Section C.1 and establish its theoretical guarantees in Sections C.4–C.6. We present the proof of Theorem 1 in Section C.2, followed by the supporting lemmas in Sections C.3 and C.4. We then provide the proof of Corollary 1 in Section C.5, and the parameter-free guarantees in Section C.6.

## C.1   Additional details on TEWA-SE

As we described in Section 2, TEWA-SE handles non-stationary environments by employing the Geometric Covering (GC) scheme from [19] to schedule experts across different time intervals. Additionally, TEWA-SE assigns an exponential grid of learning rates to the multiple experts covering each GC interval, to adapt to the curvature of the loss functions. We first invoke the definition of GC intervals from [19].

**Definition 5** (Geometric Covering (GC) intervals [19]). *For $k \in \mathbb{N}$, define the set of intervals*

$$\mathcal{I}_k = \left\{ [i \cdot 2^k, (i+1) \cdot 2^k - 1] : i \in \mathbb{N}^+ \right\}, \tag{24}$$

*that is, $\mathcal{I}_k$ is a partition of $\mathbb{N}^+ \setminus [2^k - 1]$ into intervals of length $2^k$. Then we call $\mathcal{I} = \bigcup_{k \in \mathbb{N}} \mathcal{I}_k$ the set of Geometric Covering (GC) intervals.*

For any interval length $L \in \mathbb{N}^+$, we also define the exponential grid of learning rates as

$$\mathcal{S}(L) = \left\{ \frac{2^{-i}}{5GD} : i \in \left\{ 0, 1, \dots, \left\lceil \tfrac{1}{2} \log_2 L \right\rceil \right\} \right\}, \tag{25}$$

where $G$ is the uniform upper bound (10) on $\|\boldsymbol{g}_t\|$, and $D$ is the diameter of the feasible set $\Theta$. For each given GC interval $I = [r, s] \in \mathcal{I}$, TEWA-SE instantiates multiple experts in round $r$, each assigned a distinct learning rate $\eta \in \mathcal{S}(|I|)$ and surrogate loss $\ell_t^\eta$ as defined in (11). It removes these experts after round $s$. This scheduling scheme ensures at least one expert covering $I$ effectively minimizes the linearized regret $\sum_{t \in I} \langle \mathbf{E}[\boldsymbol{g}_t | \boldsymbol{x}_t, \Lambda_T], \boldsymbol{x}_t - \boldsymbol{u} \rangle$ associated with $\hat{f}_t$ on the interval $I$ (Lemma 5), ultimately yielding the regret guarantees in Theorem 1 and Corollary 1.

**Polylogarithmic computational complexity**   For $t \in \mathbb{N}^+$, we use $\mathcal{C}_t = \{ I \in \mathcal{I} : t \in I \}$ to denote the set of GC intervals covering time $t$. From Definition 5 it is easy to verify that $|\mathcal{C}_t| = 1 + \lfloor \log_2 t \rfloor$. The longest interval in $\mathcal{C}_t$ has length at most $t$, which is associated with at most $|\mathcal{S}(t)| = 1 + \lceil \tfrac{1}{2} \log_2 t \rceil$ experts. With $\mathcal{A}_t = \{ E(I, \eta) : t \in I \}$ representing the set of experts active in round $t$, the number of active experts in round $t$, denoted by $n_t = |\mathcal{A}_t|$ in Algorithm 1, satisfies

$$n_t \leq \left( 1 + \lfloor \log_2 t \rfloor \right) \cdot \left( 1 + \left\lceil \tfrac{1}{2} \log_2 t \right\rceil \right). \tag{26}$$

This ensures that the computational complexity of TEWA-SE is only $\mathcal{O}(\log^2 T)$ per round.

**Tilted Exponentially Weighted Average**   In each round $t$, TEWA-SE aggregates the actions proposed by the active experts $E(I, \eta) \in \mathcal{A}_t$ using exponential weights, tilted by their respective learning rates, by computing

$$\boldsymbol{x}_t = \frac{\sum_{E(I, \eta) \in \mathcal{A}_t} \eta \exp(-L_{t-1,I}^\eta) \boldsymbol{x}_{t,I}^\eta}{\sum_{E(\tilde{I}, \tilde{\eta}) \in \mathcal{A}_t} \tilde{\eta} \exp(-L_{t-1,\tilde{I}}^{\tilde{\eta}})}, \tag{27}$$

where for $I = [r, s]$ and $t \in [r+1, s]$, $L_{t-1,I}^\eta = \sum_{\tau=r}^{t-1} \ell_\tau^\eta(\boldsymbol{x}_{\tau,I}^\eta)$ represents the cumulative surrogate loss accrued by expert $E(I, \eta)$ over the interval $[r, t-1]$. Note that (27) is equivalent to line 7 of Algorithm 1, rewritten with notation better suited for our proof.

In what follows, we prove some theoretical guarantees for TEWA-SE.

## C.2   Proof of Theorem 1

In this section, we first restate Theorem 1 and provide its complete proof, which relies on several supporting lemmas. For clarity of exposition, we defer the statements and proofs of these supporting lemmas to the following sections.

**Theorem 1.** *For any $T \in \mathbb{N}^+$ and $\mathsf{B} \in [T]$, Algorithm 1 with $h = \min(\sqrt{d}\mathsf{B}^{-\frac{1}{4}}, r)$ satisfies*

$$R^{\mathsf{ada}}(\mathsf{B}, T) \lesssim \sqrt{d}\mathsf{B}^{\frac{3}{4}} + d^2, \tag{14}$$

*and if $f_t$ is $\alpha$-strongly-convex with $\arg\min_{\boldsymbol{x} \in \mathbb{R}^d} f_t(\boldsymbol{x}) \in \Theta$ for all $t \in [T]$,[4] it furthermore holds that*

$$R^{\mathsf{ada}}(\mathsf{B}, T) \lesssim \frac{d}{\alpha}\sqrt{\mathsf{B}} + \frac{1}{\alpha}d^2, \tag{15}$$

*where $\lesssim$ conceals polylogarithmic terms in $\mathsf{B}$ and $T$, independent of $d$ and $\alpha$.*

*Proof of Theorem 1.* We prove (14) for the general convex case and (15) for the strongly-convex case similarly. To bound $R^{\mathsf{ada}}(\mathsf{B}, T)$, we will uniformly bound $\sum_{t=p}^{q} \mathbf{E}[f_t(\boldsymbol{z}_t) - f_t(\boldsymbol{u})]$ across all comparators $\boldsymbol{u} \in \Theta$ and intervals $[p, q]$ shorter than $\mathsf{B}$.

**Common setup:** Invoking the event $\Lambda_T = \{|\xi_t| \leq 2\sigma\sqrt{\log(T+1)}, \forall t \in [T]\}$ defined above (10), since $\{\xi_t\}_{t=1}^T$ are $\sigma$-sub-Gaussian, we have

$$\mathbf{P}(\Lambda_T^c) \leq \sum_{t=1}^{T} \mathbf{P}\left(|\xi_t| > 2\sigma\sqrt{\log(T+1)}\right) \leq 2\sum_{t=2}^{T+1} T^{-2} = 2T^{-1}. \tag{28}$$

By the law of total expectation we can write for any $\boldsymbol{u} \in \Theta$,

$$\sum_{t=p}^{q} \mathbf{E}[f_t(\boldsymbol{z}_t) - f_t(\boldsymbol{u})] = \sum_{t=p}^{q} \mathbf{E}[f_t(\boldsymbol{z}_t) - f_t(\boldsymbol{u}) \mid \Lambda_T]\mathbf{P}(\Lambda_T) + \sum_{t=p}^{q} \mathbf{E}[\underbrace{f_t(\boldsymbol{z}_t) - f_t(\boldsymbol{u})}_{\leq 2} \mid \Lambda_T^c] \underbrace{\mathbf{P}(\Lambda_T^c)}_{\leq 2T^{-1}}$$

$$\leq \sum_{t=p}^{q} \mathbf{E}[f_t(\boldsymbol{z}_t) - f_t(\boldsymbol{u}) \mid \Lambda_T] + 4. \tag{29}$$

To bound the first term in the last display, we consider the following decomposition

$$\sum_{t=p}^{q} \mathbf{E}[f_t(\boldsymbol{z}_t) - f_t(\boldsymbol{u}) \mid \Lambda_T]$$

$$= \underbrace{\sum_{t=p}^{q} \mathbf{E}[f_t(\boldsymbol{z}_t) - f_t(\boldsymbol{x}_t) \mid \Lambda_T]}_{\text{term I}} + \underbrace{\sum_{t=p}^{q} \mathbf{E}[f_t(\boldsymbol{x}_t) - \hat{f}_t(\boldsymbol{x}_t) \mid \Lambda_T]}_{\text{term II}} +$$

$$\underbrace{\sum_{t=p}^{q} \mathbf{E}[\hat{f}_t(\boldsymbol{x}_t) - \hat{f}_t(\boldsymbol{u}) \mid \Lambda_T]}_{\text{term III}} + \underbrace{\sum_{t=p}^{q} \mathbf{E}[\hat{f}_t(\boldsymbol{u}) - f_t(\boldsymbol{u}) \mid \Lambda_T]}_{\text{term IV}}. \tag{30}$$

Since $f_t$ is convex, by Jensen's inequality we obtain that term II is negative (c.f. [71, Lemma A.2 (ii)]). In what follows, we bound terms I, III and IV in this decomposition separately for the general convex case and the strongly-convex case.

**General convex and Lipschitz case:** Recall that $(\boldsymbol{\zeta}_t)_{t=1}^T$ denote uniform samples from the unit sphere $\partial\mathbb{B}^d$, and $\tilde{\boldsymbol{\zeta}}$ denotes a uniform sample from the unit ball $\mathbb{B}^d$, while $\hat{f}_t(\boldsymbol{x}) = \mathbf{E}[f_t(\boldsymbol{x} + h\tilde{\boldsymbol{\zeta}})] \forall \boldsymbol{x} \in \tilde{\Theta}$. Since $(f_t)_{t=1}^T$ are $K$-Lipschitz, $\|\boldsymbol{\zeta}_t\| = 1$, and $\mathbf{E}[\|\tilde{\boldsymbol{\zeta}}\|] \leq 1$, we can bound term I and term IV by

$$\text{term I} = \sum_{t=p}^{q} \mathbf{E}\left[\mathbf{E}[f_t(\boldsymbol{x}_t + h\boldsymbol{\zeta}_t)|\boldsymbol{x}_t] - f_t(\boldsymbol{x}_t) \mid \Lambda_T\right] \leq K(q - p + 1)h, \tag{31}$$

$$\text{term IV} = \sum_{t=p}^{q} \mathbf{E}\left[\mathbf{E}[f_t(\boldsymbol{u} + h\tilde{\boldsymbol{\zeta}})] - f_t(\boldsymbol{u}) \mid \Lambda_T\right] \leq K(q - p + 1)h. \tag{32}$$

---

[4]The assumption that loss minimizers lie inside $\Theta$ is common in zeroth-order optimization, see e.g., [7, 72, 73]. Without it, our upper bound analysis would have an extra term depending on the gradients at the minimizers.

To bound term III, recall that $\boldsymbol{g}_t$ denotes the gradient estimate of $\hat{f}_t$ at $\boldsymbol{x}_t$. We use the convexity of $\hat{f}_t$ and apply Lemma 3 to obtain that for any $\boldsymbol{u} \in \Theta$,

$$\text{term III} \leq \mathbf{E}\left[\sum_{t=p}^{q} \langle \mathbf{E}[\boldsymbol{g}_t | \boldsymbol{x}_t, \Lambda_T], \boldsymbol{x}_t - \boldsymbol{u} \rangle \,\big|\, \Lambda_T \right]$$

$$\leq \mathbf{E}\left[10 G D a_{p,q} b_{p,q} + 3G\sqrt{a_{p,q} b_{p,q}} \sqrt{\sum_{t=p}^{q} \|\boldsymbol{x}_t - \boldsymbol{u}\|^2} \,\big|\, \Lambda_T \right]$$

$$\leq 10 G D a_{p,q} b_{p,q} + 3 G D \sqrt{a_{p,q} b_{p,q}} \sqrt{q - p + 1}\,, \tag{33}$$

where all constants are explicit in the statement of the lemma. By combining the bounds for all four terms in (30) with (29), and using $h = \min\left(\sqrt{d}\mathsf{B}^{-\frac{1}{4}}, r\right)$, $G = \frac{d}{h}(1 + 2\sigma\sqrt{\log(T+1)})$, $a_{p,q} = \frac{1}{2} + 2\log(2q) + \frac{1}{2}\log(q - p + 1) \leq 6\log(T+1)$, and $b_{p,q} = 2\lceil \log_2(q - p + 2)\rceil \leq 6\log(\mathsf{B}+1)$, we establish that

$$R^{\mathsf{ada}}(\mathsf{B}, T) \leq 10 G D a_{p,q} b_{p,q} + 3 G D \sqrt{a_{p,q} b_{p,q}} \sqrt{q - p + 1} + 2K(q - p + 1)h + 4$$

$$\leq \begin{cases} C\sqrt{d}\mathsf{B}^{\frac{3}{4}} + 4 & \text{if } h = \sqrt{d}\mathsf{B}^{-\frac{1}{4}} \\ C_1 d^2 + C_2 d + 4 & \text{if } h = r \end{cases}$$

$$\leq C\sqrt{d}\mathsf{B}^{\frac{3}{4}} + C_1 d^2 + C_2 d + 4\,, \tag{34}$$

where $C, C_1, C_2 > 0$ are polylogarithmic in $T$, independent of $d$, defined with $M_T = 1 + 2\sigma\sqrt{\log(T+1)}$ and $N_{T,\mathsf{B}} = \log(T+1)\log(\mathsf{B}+1)$ as

$$C = 18 D M_T\left(\sqrt{N_{T,\mathsf{B}}} + 20 N_{T,\mathsf{B}}\mathsf{B}^{-\frac{1}{2}}\right) + 2K \tag{35}$$

$$C_1 = \left(18 D M_T \sqrt{N_{T,\mathsf{B}}} + 2K\right)/r^3 \tag{36}$$

$$C_2 = 360 D M_T N_{T,\mathsf{B}}/r\,. \tag{37}$$

This concludes the proof of (14).

**Strongly-convex and smooth case:** Due to the $\beta$-smoothness of $f_t$ and the fact that $\mathbf{E}[\boldsymbol{\zeta}_t] = \mathbf{E}[\tilde{\boldsymbol{\zeta}}] = \boldsymbol{0}$ and $\mathbf{E}[\|\tilde{\boldsymbol{\zeta}}\|^2] \leq \mathbf{E}[\|\boldsymbol{\zeta}_t\|^2] = 1$, we can bound term I and term IV each by $\frac{\beta}{2}(q - p + 1)h^2$. When the $f_t$'s are strongly-convex, we can derive a tighter bound on term III than that in (33) by restricting the comparator $\boldsymbol{u}$ to the clipped domain $\tilde{\Theta}$ and using the fact that when $f_t$ is $\alpha$-strongly convex on $\Theta$, $\hat{f}_t$ is $\alpha$-strongly convex on $\tilde{\Theta}$ (c.f. [71, Lemma A.3]). That is, we have for any $\boldsymbol{u} \in \tilde{\Theta}$,

$$\text{term III} \leq \mathbf{E}\left[\sum_{t=p}^{q} \langle \mathbf{E}[\boldsymbol{g}_t | \boldsymbol{x}_t, \Lambda_T], \boldsymbol{x}_t - \boldsymbol{u} \rangle - \frac{\alpha}{2}\sum_{t=p}^{q} \|\boldsymbol{x}_t - \boldsymbol{u}\|^2 \,\big|\, \Lambda_T \right]$$

$$\leq \mathbf{E}\left[10 G D a_{p,q} b_{p,q} + \underbrace{3G\sqrt{a_{p,q} b_{p,q}} \sqrt{\sum_{t=p}^{q} \|\boldsymbol{x}_t - \boldsymbol{u}\|^2} - \frac{\alpha}{2}\sum_{t=p}^{q} \|\boldsymbol{x}_t - \boldsymbol{u}\|^2}_{=:\delta} \,\big|\, \Lambda_T \right]$$

$$\leq \left(10 G D + \tfrac{18}{\alpha} G^2\right) a_{p,q} b_{p,q}\,, \tag{38}$$

where the last inequality holds because term $\delta$ is uniformly bounded as follows:

$$\delta \leq \begin{cases} \frac{18}{\alpha} G^2 a_{p,q} b_{p,q} & \text{if } \sqrt{\sum_{t=p}^{q} \|\boldsymbol{x}_t - \boldsymbol{u}\|^2} \leq \frac{6}{\alpha} G\sqrt{a_{p,q} b_{p,q}}\,, \\ 0 & \text{otherwise.} \end{cases}$$

Combining (38) with (29)–(30) and simplifying yields for any $\boldsymbol{u} \in \tilde{\Theta}$,

$$\sum_{t=p}^{q} \mathbf{E}[f_t(\boldsymbol{z}_t) - f_t(\boldsymbol{u})] \leq \left(10 G D + \tfrac{18}{\alpha} G^2\right) a_{p,q} b_{p,q} + \beta(q - p + 1)h^2 + 4\,. \tag{39}$$

The final step is to handle the case where the comparator $\boldsymbol{u} \in \Theta \setminus \tilde{\Theta}$. Consider the worst case when the comparator is $\boldsymbol{u}^* \in \arg\min_{\boldsymbol{u} \in \Theta} \sum_{t=p}^{q} f_t(\boldsymbol{u})$ with $\boldsymbol{u}^* \in \Theta \setminus \tilde{\Theta}$. Let $\tilde{\boldsymbol{u}}^* = \Pi_{\tilde{\Theta}}(\boldsymbol{u}^*)$. If

$\arg\min_{\boldsymbol{x}\in\mathbb{R}^d} f_t(\boldsymbol{x}) \in \Theta \ \forall t \in [T]$, then by the $\beta$-smoothness of the $f_t$'s we have

$$\sum_{t=p}^{q} f_t(\tilde{\boldsymbol{u}}^*) - f_t(\boldsymbol{u}^*) \leq \sum_{t=p}^{q} \left[ \left\langle \underbrace{\nabla f_t(\boldsymbol{u}^*)}_{=0}, \tilde{\boldsymbol{u}}^* - \boldsymbol{u}^* \right\rangle + \frac{\beta}{2} \underbrace{\|\tilde{\boldsymbol{u}}^* - \boldsymbol{u}^*\|^2}_{\leq h^2} \right] = \frac{\beta}{2}(q - p + 1)h^2. \quad (40)$$

Combining (39) with (40) yields

$$\begin{aligned}
R^{\mathsf{ada}}(\mathsf{B}, T) &\leq \left( 10GD + \frac{18}{\alpha}G^2 \right) a_{p,q} b_{p,q} + \frac{3}{2}\beta(q - p + 1)h^2 + 4 \\
&\leq \begin{cases} C'd\sqrt{\mathsf{B}} + 4 & \text{if } h = \sqrt{d}\mathsf{B}^{-\frac{1}{4}} \\ C_1'd^2 + C_2'd + 4 & \text{if } h = r \end{cases} \\
&\leq C'\frac{d}{\alpha}\sqrt{\mathsf{B}} + C_1'\frac{1}{\alpha}d^2 + C_2'd + 4\,,
\end{aligned} \quad (41)$$

where $C', C_1', C_2' > 0$ are polylogarithmic in $T$ and B, independent of $d$, defined as

$$C' = 72 \left( 9M_T + 5\alpha D\mathsf{B}^{-\frac{1}{4}} \right) M_T N_{T,\mathsf{B}} + \frac{3}{2}\beta\alpha \quad (42)$$

$$C_1' = \left( 648M_T^2 N_{T,\mathsf{B}} + \beta\alpha \right)/r^2 \quad (43)$$

$$C_2' = 360DM_T N_{T,\mathsf{B}}/r\,. \quad (44)$$

This concludes the proof of (15). $\qquad\qquad\qquad\qquad\qquad\qquad\qquad\qquad\qquad\qquad\qquad\qquad$ $\square$

The proof above crucially relies on Lemma 3, which we state and prove in the following section.

### C.3 Upper bounds on linearized regret

Lemma 3 establishes an upper bound on the linearized regret associated with the smoothed loss $\hat{f}_t$ for any arbitrary interval $I = [p, q] \subseteq [1, T]$. This result builds on two key components: Lemma 4, which characterizes how a given arbitrary interval is covered by a sequence of GC intervals, and Lemma 5, which provides an upper bound on the linearized regret for each GC interval $I \in \mathcal{I}$. For clarity, we first present and prove Lemma 3, then proceed to detail the supporting Lemmas 4 and 5.

**Lemma 3** (Linearized regret on an arbitrary interval). *For an arbitrary interval $I = [p, q] \subseteq [1, T]$, Algorithm 1 satisfies for all $\boldsymbol{u} \in \Theta$,*

$$\sum_{t=p}^{q} \langle \mathbf{E}[\boldsymbol{g}_t | \boldsymbol{x}_t, \Lambda_T], \boldsymbol{x}_t - \boldsymbol{u} \rangle \leq 10GDa_{p,q}b_{p,q} + 3G\sqrt{a_{p,q}b_{p,q}}\sqrt{\sum_{t=p}^{q}\|\boldsymbol{x}_t - \boldsymbol{u}\|^2}, \quad (45)$$

*where $a_{p,q} = \frac{1}{2} + 2\log(2q) + \frac{1}{2}\log(q - p + 1)$ and $b_{p,q} = 2\lceil\log_2(q - p + 2)\rceil$.*

*Proof of Lemma 3.* This proof follows similar arguments to those used in proving the first part of [24, Theorem 2]. To begin, according to Lemma 4, any arbitrary interval $I = [p, q] \subseteq [1, T]$ can be covered by two sequences of consecutive and disjoint GC intervals, denoted by $I_{-m}, \ldots, I_0 \in \mathcal{I}$ and $I_1, \ldots, I_n \in \mathcal{I}$, where $n, m \in \mathbb{N}^+$ with $n \leq \lceil\log_2(q - p + 2)\rceil$ and $m + 1 \leq \lceil\log_2(q - p + 2)\rceil$. Note that negative indices correspond to GC intervals that precede $I_0$, while positive indices correspond to intervals that follow it. The indices indicate temporal ordering and are unrelated to the length of the intervals.

By applying the linearized regret bound from Lemma 5 to each GC interval, and noticing that $a_{r,s} \leq a_{p,q}$ for any subinterval $[r, s] \subseteq [p, q]$ (as evident from the definition of $a_{p,q}$ in (45)), we

establish for all $\boldsymbol{u} \in \Theta$,

$$
\begin{aligned}
\sum_{t=p}^{q} \langle \mathbf{E}[\boldsymbol{g}_t | \boldsymbol{x}_t, \Lambda_T], \boldsymbol{x}_t - \boldsymbol{u} \rangle &= \sum_{i=-m}^{n} \sum_{t \in I_i} \langle \mathbf{E}[\boldsymbol{g}_t | \boldsymbol{x}_t, \Lambda_T], \boldsymbol{x}_t - \boldsymbol{u} \rangle \\
&\leq \sum_{i=-m}^{n} \left[ 3G \sqrt{a_{p,q} \sum_{t \in I_i} \|\boldsymbol{x}_t - \boldsymbol{u}\|^2} + 10GDa_{p,q} \right] \\
&= 10GDa_{p,q}(n+m+1) + 3G\sqrt{a_{p,q}} \sum_{i=-m}^{n} \sqrt{\sum_{t \in I_i} \|\boldsymbol{x}_t - \boldsymbol{u}\|^2} \\
&\leq 10GDa_{p,q}(n+m+1) + 3G\sqrt{a_{p,q}} \sqrt{(n+m+1) \sum_{i=-m}^{n} \sum_{t \in I_i} \|\boldsymbol{x}_t - \boldsymbol{u}\|^2} \\
&\leq 10GDa_{p,q}b_{p,q} + 3G\sqrt{a_{p,q}b_{p,q}} \sqrt{\sum_{t=p}^{q} \|\boldsymbol{x}_t - \boldsymbol{u}\|^2} \,, \quad (46)
\end{aligned}
$$

where the last step uses $n + m + 1 \leq 2 \lceil \log_2(q - p + 2) \rceil =: b_{p,q}$. $\qquad \square$

We now present Lemmas 4 and 5 which we used to prove Lemma 3 above.

**Lemma 4** (Covering property of GC intervals)**.** *Any arbitrary interval $I = [p, q] \subseteq \mathbb{N}^+$ can be partitioned into two finite sequences of consecutive and disjoint GC intervals, denoted by $I_{-m}, \ldots, I_0 \in \mathcal{I}$ and $I_1, \ldots, I_n \in \mathcal{I}$, where $I = \bigcup_{i=-m}^{n} I_i$, such that*

$$
\frac{|I_{-i}|}{|I_{-i+1}|} \leq \tfrac{1}{2} \quad \forall i \geq 1, \quad \text{and} \quad \frac{|I_i|}{|I_{i-1}|} \leq \tfrac{1}{2}, \quad \forall i \geq 2 \,, \quad (47)
$$

*with*

$$
n \leq \lceil \log_2(q - p + 2) \rceil, \quad \text{and} \quad m + 1 \leq \lceil \log_2(q - p + 2) \rceil \,. \quad (48)
$$

*Proof of Lemma 4.* Eq. (47) directly comes from [19, Lemma 1.2]. To prove (48), suppose for contradiction $n > \lceil \log_2(q - p + 2) \rceil$, then we have

$$
\sum_{i=1}^{n} |I_i| \geq \sum_{i=1}^{n} 2^{i-1} = 2^n - 1 > q - p + 1 = |I| \,, \quad (49)
$$

contradicting the fact that $\bigcup_{i=-m}^{n} I_i = I$. By the same reasoning, we have $m + 1 \leq \lceil \log_2(q - p + 2) \rceil$. $\qquad \square$

**Lemma 5** (Linearized regret on a GC interval)**.** *For any GC interval $I = [r, s] \in \mathcal{I}$, Algorithm 1 satisfies for all $\boldsymbol{u} \in \Theta$,*

$$
\sum_{t=r}^{s} \langle \mathbf{E}[\boldsymbol{g}_t | \boldsymbol{x}_t, \Lambda_T], \boldsymbol{x}_t - \boldsymbol{u} \rangle \leq 3G \sqrt{a_{r,s} \sum_{t=r}^{s} \|\boldsymbol{x}_t - \boldsymbol{u}\|^2} + 10GDa_{r,s}, \quad (50)
$$

*where $a_{r,s} = \tfrac{1}{2} + 2\log(2s) + \tfrac{1}{2}\log(s - r + 1)$.*

*Proof of Lemma 5.* This proof is similar to that of [24, Lemma 12]. For any GC interval $I = [r, s] \in \mathcal{I}$ and learning rate $\eta \in \mathcal{S}(s - r + 1)$, we can apply the definition of surrogate loss $\ell_t^\eta$ from (11),

noticing that $\ell_t^\eta(\boldsymbol{x}_t) = 0$, to obtain for all $\boldsymbol{u} \in \Theta$,

$$\sum_{t=r}^{s} \eta \left\langle \mathbf{E}[\boldsymbol{g}_t | \boldsymbol{x}_t, \Lambda_T], \boldsymbol{x}_t - \boldsymbol{u} \right\rangle - \sum_{t=r}^{s} \eta^2 G^2 \|\boldsymbol{x}_t - \boldsymbol{u}\|^2$$

$$= \sum_{t=r}^{s} \mathbf{E}[-\ell_t^\eta(\boldsymbol{u}) \mid \boldsymbol{x}_t, \Lambda_T]$$

$$= \underbrace{\sum_{t=r}^{s} \mathbf{E}\left[ \ell_t^\eta(\boldsymbol{x}_t) - \ell_t^\eta(\boldsymbol{x}_{t,I}^\eta) \mid \boldsymbol{x}_t, \Lambda_T \right]}_{\text{meta-regret} \leq 2\log(2s)} + \underbrace{\sum_{t=r}^{s} \mathbf{E}\left[ \ell_t^\eta(\boldsymbol{x}_{t,I}^\eta) - \ell_t^\eta(\boldsymbol{u}) \mid \boldsymbol{x}_t, \Lambda_T \right]}_{\text{expert-regret} \leq \frac{1}{2} + \frac{1}{2}\log(s - r + 1)}$$

$$\leq 2\log(2s) + \tfrac{1}{2} + \tfrac{1}{2}\log(s - r + 1) =: a_{r,s}, \tag{51}$$

where the last step applies the upper bound on the expert-regret established in Lemma 6 and the upper bound on the meta-regret in Lemma 7, both of which we defer to Section C.4. Eq. (51) can be rearranged into

$$\sum_{t=r}^{s} \left\langle \mathbf{E}[\boldsymbol{g}_t | \boldsymbol{x}_t, \Lambda_T], \boldsymbol{x}_t - \boldsymbol{u} \right\rangle \leq \frac{a_{r,s}}{\eta} + \eta G^2 \sum_{t=r}^{s} \|\boldsymbol{x}_t - \boldsymbol{u}\|^2. \tag{52}$$

The optimal value of $\eta$ that minimizes the RHS of (52) is

$$\eta^* = \sqrt{\frac{a_{r,s}}{G^2 \sum_{t=r}^{s} \|\boldsymbol{x}_t - \boldsymbol{u}\|^2}}. \tag{53}$$

Note that since $a_{r,s} \geq \frac{1}{2}$, $\eta^* \geq \frac{1}{GD\sqrt{2(s-r+1)}}$ for all $\boldsymbol{x} \in \Theta$. The next step is to select a value $\eta$ from the set $\mathcal{S}(s - r + 1) = \left\{ \frac{2^{-i}}{5GD} : i \in \left\{ 0, 1, \dots, \left\lceil \frac{1}{2}\log_2(s - r + 1) \right\rceil \right\} \right\}$ that best approximates $\eta^*$. Two cases arises:

i) If $\eta^* \leq \frac{1}{5GD}$, there must exist an $\eta \in \mathcal{S}(s - r + 1)$ such that $\frac{\eta^*}{2} \leq \eta \leq \eta^*$. Substituting this choice of $\eta$ into (52) gives

$$\sum_{t=r}^{s} \left\langle \mathbf{E}[\boldsymbol{g}_t | \boldsymbol{x}_t, \Lambda_T], \boldsymbol{x}_t - \boldsymbol{u} \right\rangle \leq \frac{2a_{r,s}}{\eta^*} + \eta^* G^2 \sum_{t=r}^{s} \|\boldsymbol{x}_t - \boldsymbol{u}\|^2 = 3G \sqrt{a_{r,s} \sum_{t=r}^{s} \|\boldsymbol{x}_t - \boldsymbol{u}\|^2}. \tag{54}$$

ii) If $\eta^* > \frac{1}{5GD}$, then the best choice of $\eta \in \mathcal{S}(s - r + 1)$ is $\eta = \frac{1}{5GD}$, which leads to

$$\sum_{t=r}^{s} \left\langle \mathbf{E}[\boldsymbol{g}_t | \boldsymbol{x}_t, \Lambda_T], \boldsymbol{x}_t - \boldsymbol{u} \right\rangle \leq a_{r,s} \cdot 5GD \cdot 2 = 10GDa_{r,s}. \tag{55}$$

Combining (54)–(55) concludes the proof. $\qquad\square$

The proof of Lemma 5 above relied on the upper bounds on the expert-regret and meta-regret from Lemmas 6 and 7. We present and prove these lemmas in the following section.

## C.4 Upper bounds on expert-regret and meta-regret

**Lemma 6** (Expert-regret). *For any GC interval $I = [r, s] \in \mathcal{I}$ and learning rate $\eta \in \mathcal{S}(s - r + 1)$, Algorithm 1 satisfies for all $\boldsymbol{u} \in \tilde{\Theta}$,*

$$\sum_{t=r}^{s} \mathbf{E}\left[ \ell_t^\eta(\boldsymbol{x}_{t,I}^\eta) - \ell_t^\eta(\boldsymbol{u}) \mid \boldsymbol{x}_t, \boldsymbol{x}_{t,I}^\eta, \Lambda_T \right] \leq \tfrac{1}{2} + \tfrac{1}{2}\log(s - r + 1). \tag{56}$$

*Proof of Lemma 6.* The proof follows standard convergence analysis of projected online gradient descent for strongly convex objective functions, see e.g., [74, Theorem 1]. For any time step $t \in I$, the surrogate loss $\ell_t^\eta$ associated with the expert with learning rate $\eta$ and lifetime $I = [r, s]$ serves as our strongly-convex objective function. By applying the definition of $\ell_t^\eta$, we have for all $x \in \Theta$,

$$\mathbf{E}\left[\|\nabla \ell_t^\eta(\boldsymbol{x})\|^2 \mid \boldsymbol{x}_t, \Lambda_T\right] = \mathbf{E}\left[\|\eta \boldsymbol{g}_t + 2\eta^2 G^2(\boldsymbol{x} - \boldsymbol{x}_t)\|^2 \mid \boldsymbol{x}_t, \Lambda_T\right]$$
$$\leq \mathbf{E}\left[\left(\|\eta \boldsymbol{g}_t\| + \|2\eta^2 G^2(\boldsymbol{x} - \boldsymbol{x}_t)\|\right)^2 \mid \boldsymbol{x}_t, \Lambda_T\right] \quad \leq (G')^2, \quad (57)$$

where we introduced $G' = \eta G + 2\eta^2 G^2 D$. By the update rule of our projected online gradient descent with step size $\mu_t$ (line 4 of Algorithm 2), we have for all $\boldsymbol{u} \in \tilde{\Theta}$,

$$\|\boldsymbol{x}_{t+1,I}^\eta - \boldsymbol{u}\|^2 = \left\|\Pi_{\tilde{\Theta}}\left(\boldsymbol{x}_{t,I}^\eta - \mu_t \nabla \ell_t^\eta(\boldsymbol{x}_{t,I}^\eta)\right) - \boldsymbol{u}\right\|^2$$
$$\leq \left\|\boldsymbol{x}_{t,I}^\eta - \mu_t \nabla \ell_t^\eta(\boldsymbol{x}_{t,I}^\eta) - \boldsymbol{u}\right\|^2$$
$$= \|\boldsymbol{x}_{t,I}^\eta - \boldsymbol{u}\|^2 + \mu_t^2\|\nabla \ell_t^\eta(\boldsymbol{x}_{t,I}^\eta)\|^2 - 2\mu_t(\boldsymbol{x}_{t,I}^\eta - \boldsymbol{u})^\top \nabla \ell_t^\eta(\boldsymbol{x}_{t,I}^\eta),$$

which can be rearranged into

$$2(\boldsymbol{x}_{t,I}^\eta - \boldsymbol{u})^\top \nabla \ell_t^\eta(\boldsymbol{x}_{t,I}^\eta) \leq \frac{\|\boldsymbol{x}_{t,I}^\eta - \boldsymbol{u}\|^2 - \|\boldsymbol{x}_{t+1,I}^\eta - \boldsymbol{u}\|^2}{\mu_t} + \mu_t\|\nabla \ell_t^\eta(\boldsymbol{x}_{t,I}^\eta)\|^2. \quad (58)$$

Define shorthand $\lambda \equiv 2\eta^2 G^2$ and recall $\mu_t = 1/(\lambda(t - r + 1))$, then Eq. (58) implies that

$$2\sum_{t=r}^s (\boldsymbol{x}_{t,I}^\eta - \boldsymbol{u})^\top \nabla \ell_t^\eta(\boldsymbol{x}_{t,I}^\eta) - \lambda\sum_{t=r}^s \|\boldsymbol{x}_{t,I}^\eta - \boldsymbol{u}\|^2$$
$$\leq \sum_{t=r}^s \frac{\|\boldsymbol{x}_{t,I}^\eta - \boldsymbol{u}\|^2 - \|\boldsymbol{x}_{t+1,I}^\eta - \boldsymbol{u}\|^2}{\mu_t} + \sum_{t=r}^s \mu_t\|\nabla \ell_t^\eta(\boldsymbol{x}_{t,I}^\eta)\|^2 - \lambda\sum_{t=r}^s \|\boldsymbol{x}_{t,I}^\eta - \boldsymbol{u}\|^2$$
$$= \sum_{t=r+1}^s \|\boldsymbol{x}_{t,I}^\eta - \boldsymbol{u}\|^2 \underbrace{\left(\frac{1}{\mu_t} - \frac{1}{\mu_{t-1}} - \lambda\right)}_{=0} + \|\boldsymbol{x}_{r,I}^\eta - \boldsymbol{u}\|^2 \underbrace{\left(\frac{1}{\mu_r} - \lambda\right)}_{=0} - \underbrace{\frac{\|\boldsymbol{x}_{s+1,I}^\eta - \boldsymbol{u}\|^2}{\mu_s}}_{\geq 0}$$
$$+ \sum_{t=r}^s \mu_t\|\nabla \ell_t^\eta(\boldsymbol{x}_{t,I}^\eta)\|^2 \leq \sum_{t=r}^s \mu_t\|\nabla \ell_t^\eta(\boldsymbol{x}_{t,I}^\eta)\|^2. \quad (59)$$

Noticing that with any given $\boldsymbol{x}_t \in \mathbb{R}^d$, $\ell_t^\eta$ is $\lambda$-strongly-convex, we apply (59) to obtain that for all $\boldsymbol{u} \in \tilde{\Theta}$,

$$2\sum_{t=r}^s \mathbf{E}\left[\ell_t^\eta(\boldsymbol{x}_{t,I}^\eta) - \ell_t^\eta(\boldsymbol{u}) \mid \boldsymbol{x}_t, \boldsymbol{x}_{t,I}^\eta, \Lambda_T\right]$$
$$\leq \mathbf{E}\left[2\sum_{t=r}^s (\boldsymbol{x}_{t,I}^\eta - \boldsymbol{u})^\top \nabla \ell_t^\eta(\boldsymbol{x}_{t,I}^\eta) - \lambda\sum_{t=r}^s \|\boldsymbol{x}_{t,I}^\eta - \boldsymbol{u}\|^2 \,\bigg|\, \{\boldsymbol{x}_t, \boldsymbol{x}_{t,I}^\eta\}_{t=r}^s, \Lambda_T\right]$$
$$\leq \sum_{t=r}^s \mu_t \mathbf{E}\left[\|\nabla \ell_t^\eta(\boldsymbol{x}_{t,I}^\eta)\|^2 \mid \boldsymbol{x}_t, \boldsymbol{x}_{t,I}^\eta, \Lambda_T\right]$$
$$\overset{(i)}{\leq} (G')^2 \sum_{t=r}^s \mu_t \overset{(ii)}{\leq} \frac{(G')^2}{\lambda}\left(1 + \log(s - r + 1)\right) \overset{(iii)}{\leq} 1 + \log(s - r + 1), \quad (60)$$

where (i) is a result of (57), (ii) uses the bound $\sum_{k=1}^n \frac{1}{k} \leq 1 + \log n$ for any $n \in \mathbb{N}^+$, and (iii) uses the fact that given $\eta \leq \frac{1}{5GD}$ it holds that

$$(G')^2 = \left(\eta G + 2\eta^2 G^2 D\right)^2 = \eta^2 G^2 + 4\eta^3 G^3 D + 4\eta^4 G^4 D^2 \leq \eta^2 G^2 + \tfrac{4}{5}\eta^2 G^2 + \tfrac{4}{25}\eta^2 G^2 \leq \lambda.$$

$\square$

**Lemma 7** (Meta-regret). *For any GC interval $I = [r, s] \in \mathcal{I}$ and learning rate $\eta \in \mathcal{S}(s - r + 1)$, Algorithm 1 satisfies*

$$\sum_{t=r}^s \mathbf{E}\left[\ell_t^\eta(\boldsymbol{x}_t) - \ell_t^\eta(\boldsymbol{x}_{t,I}^\eta) \mid \boldsymbol{x}_t, \boldsymbol{x}_{t,I}^\eta, \Lambda_T\right] = -\sum_{t=r}^s \mathbf{E}\left[\ell_t^\eta(\boldsymbol{x}_{t,I}^\eta) \mid \boldsymbol{x}_t, \boldsymbol{x}_{t,I}^\eta, \Lambda_T\right] \leq 2\log(2s). \quad (61)$$

*Proof of Lemma 7.* The proof is similar to that of [24, Lemma 6]. By Jensen's inequality and the convexity of norms, we have for all $\boldsymbol{x} \in \Theta$,

$$|\langle \mathbf{E}[\boldsymbol{g}_t|\boldsymbol{x}_t, \Lambda_T], \boldsymbol{x}_t - \boldsymbol{x}\rangle| \le \|\mathbf{E}[\boldsymbol{g}_t|\boldsymbol{x}_t, \Lambda_T]\|\|\boldsymbol{x}_t - \boldsymbol{x}\|$$
$$\le \mathbf{E}\left[\|\boldsymbol{g}_t\| \mid \boldsymbol{x}_t, \Lambda_T\right]\|\boldsymbol{x}_t - \boldsymbol{x}\| \le GD\,, \tag{62}$$

which, given $\eta \le \frac{1}{5GD}$, implies that

$$\eta\langle\mathbf{E}[\boldsymbol{g}_t|\boldsymbol{x}_t, \Lambda_T], \boldsymbol{x}_t - \boldsymbol{x}\rangle \ge -\eta GD \ge -\frac{1}{5}\,. \tag{63}$$

Using (62)–(63) and applying the inequality $\ln(1 + z) \ge z - z^2$ for any $z \ge -\frac{2}{3}$ with $z = \eta\langle\mathbf{E}[\boldsymbol{g}_t|\boldsymbol{x}_t, \Lambda_T], \boldsymbol{x}_t - \boldsymbol{x}\rangle$, we obtain for all $\boldsymbol{x} \in \Theta$,

$$\exp\left(-\mathbf{E}[\ell_t^\eta(\boldsymbol{x}) \mid \boldsymbol{x}_t, \Lambda_T]\right) = \exp\left(\eta\langle\mathbf{E}[\boldsymbol{g}_t|\boldsymbol{x}_t, \Lambda_T], \boldsymbol{x}_t - \boldsymbol{x}\rangle - \eta^2 G^2\|\boldsymbol{x}_t - \boldsymbol{x}\|^2\right)$$
$$\le \exp\left(\eta\langle\mathbf{E}[\boldsymbol{g}_t|\boldsymbol{x}_t, \Lambda_T], \boldsymbol{x}_t - \boldsymbol{x}\rangle - \eta^2\langle\mathbf{E}[\boldsymbol{g}_t|\boldsymbol{x}_t, \Lambda_T], \boldsymbol{x}_t - \boldsymbol{x}\rangle^2\right)$$
$$\le 1 + \eta\langle\mathbf{E}[\boldsymbol{g}_t|\boldsymbol{x}_t, \Lambda_T], \boldsymbol{x}_t - \boldsymbol{x}\rangle\,. \tag{64}$$

Define shorthand $\mathbb{F}_{t,I}^\eta = \{\boldsymbol{x}_t, \boldsymbol{x}_{t,I}^\eta, \Lambda_T\}$, and $\mathbb{H}_{t,I}^\eta = \cup_{\tau \in [t]}\mathbb{F}_{\tau,I}^\eta$ for $t \in [T]$. Using (64), we can write for every $t \in [T]$,

$$\sum_{E(I,\eta)\in\mathcal{A}_t} \exp\left(-\mathbf{E}[L_{t,I}^\eta \mid \mathbb{H}_{t,I}^\eta]\right)$$

$$= \sum_{E(I,\eta)\in\mathcal{A}_t} \exp\left(-\mathbf{E}[L_{t-1,I}^\eta \mid \mathbb{H}_{t-1,I}^\eta]\right)\exp\left(-\mathbf{E}[\ell_t^\eta(\boldsymbol{x}_{t,I}^\eta) \mid \mathbb{F}_{t,I}^\eta]\right)$$

$$\le \sum_{E(I,\eta)\in\mathcal{A}_t} \exp\left(-\mathbf{E}[L_{t-1,I}^\eta \mid \mathbb{H}_{t-1,I}^\eta]\right)\left[1 + \eta\left\langle\mathbf{E}[\boldsymbol{g}_t|\boldsymbol{x}_t, \Lambda_T], \boldsymbol{x}_t - \boldsymbol{x}_{t,I}^\eta\right\rangle\right]. \tag{65}$$

The second term on the RHS can be bounded as follows:

$$\sum_{E(I,\eta)\in\mathcal{A}_t} \exp(-\mathbf{E}[L_{t-1,I}^\eta \mid \mathbb{H}_{t-1,I}^\eta])\left[\eta\left\langle\mathbf{E}[\boldsymbol{g}_t|\boldsymbol{x}_t, \Lambda_T], \boldsymbol{x}_t - \boldsymbol{x}_{t,I}^\eta\right\rangle\right]$$

$$= \left\langle\mathbf{E}[\boldsymbol{g}_t|\boldsymbol{x}_t, \Lambda_T], \sum_{E(I,\eta)\in\mathcal{A}_t}\eta\exp(-\mathbf{E}[L_{t-1,I}^\eta \mid \mathbb{H}_{t-1,I}^\eta])(\boldsymbol{x}_t - \boldsymbol{x}_{t,I}^\eta)\right\rangle$$

$$\overset{(i)}{\le} \left\langle\mathbf{E}[\boldsymbol{g}_t|\boldsymbol{x}_t, \Lambda_T], \sum_{E(I,\eta)\in\mathcal{A}_t}\eta\mathbf{E}\left[\exp(-L_{t-1,I}^\eta) \mid \mathbb{H}_{t-1,I}^\eta\right](\boldsymbol{x}_t - \boldsymbol{x}_{t,I}^\eta)\right\rangle$$

$$= \left\langle\mathbf{E}[\boldsymbol{g}_t|\boldsymbol{x}_t, \Lambda_T], \mathbf{E}\Big[\underbrace{\sum_{E(I,\eta)\in\mathcal{A}_t}\eta\exp(-L_{t-1,I}^\eta)(\boldsymbol{x}_t - \boldsymbol{x}_{t,I}^\eta)}_{=0}\Big|\mathbb{H}_{t-1,I}^\eta\Big]\right\rangle \overset{(ii)}{=} 0\,, \tag{66}$$

where (i) applies Jensen's inequality, and (ii) is due to the update rule of $\boldsymbol{x}_t$ in (27). Combining (65)–(66) yields

$$\sum_{E(I,\eta)\in\mathcal{A}_t} \exp\left(-\mathbf{E}[L_{t,I}^\eta \mid \mathbb{H}_{t,I}^\eta]\right) \le \sum_{E(I,\eta)\in\mathcal{A}_t} \exp\left(-\mathbf{E}[L_{t-1,I}^\eta \mid \mathbb{H}_{t-1,I}^\eta]\right). \tag{67}$$

By summing both sides of (67) over $t = 1, \ldots, s$ and rewriting, we obtain

$$\sum_{E(I,\eta)\in\mathcal{A}_s} \exp\left(-\mathbf{E}[L_{s,I}^\eta \mid \mathbb{H}_{s,I}^\eta]\right) + \sum_{t=1}^{s-1}\sum_{E(I,\eta)\in\mathcal{A}_t\backslash\mathcal{A}_{t+1}} \exp\left(-\mathbf{E}[L_{t,I}^\eta \mid \mathbb{H}_{t,I}^\eta]\right) +$$

$$\sum_{t=1}^{s-1}\sum_{E(I,\eta)\in\mathcal{A}_t\cap\mathcal{A}_{t+1}} \exp\left(-\mathbf{E}[L_{t,I}^\eta \mid \mathbb{H}_{t,I}^\eta]\right)$$

$$\le \sum_{E(I,\eta)\in\mathcal{A}_1} \exp\left(-\mathbf{E}[L_{0,I}^\eta]\right) + \sum_{t=2}^{s}\sum_{E(I,\eta)\in\mathcal{A}_t\backslash\mathcal{A}_{t-1}} \exp\left(-\mathbf{E}[L_{t-1,I}^\eta \mid \mathbb{H}_{t-1,I}^\eta]\right) +$$

$$\sum_{t=2}^{s}\sum_{E(I,\eta)\in\mathcal{A}_t\cap\mathcal{A}_{t-1}} \exp\left(-\mathbf{E}[L_{t-1,I}^\eta \mid \mathbb{H}_{t-1,I}^\eta]\right). \tag{68}$$

Canceling the equivalent last terms on both sides of (68) and noting that $L_{\tau,I}^{\eta} = 0$ for $\tau = \min\{t : t \in I\} - 1$ by construction (see line 4 of Algorithm 1), we obtain for $s \geq 1$,

$$\sum_{E(I,\eta)\in\mathcal{A}_s} \exp\left(-\mathbf{E}[L_{s,I}^{\eta} \mid \mathbb{H}_{s,I}^{\eta}]\right) + \sum_{t=1}^{s-1}\left[\sum_{E(I,\eta)\in\mathcal{A}_t\backslash\mathcal{A}_{t+1}} \exp\left(-\mathbf{E}[L_{t,I}^{\eta} \mid \mathbb{H}_{t,I}^{\eta}]\right)\right]$$

$$\leq \sum_{E(I,\eta)\in\mathcal{A}_1} \exp\left(\mathbf{E}[\underbrace{L_{0,I}^{\eta}}_{=0}]\right) + \sum_{t=2}^{s}\left[\sum_{E(I,\eta)\in\mathcal{A}_t\backslash\mathcal{A}_{t-1}} \exp\left(-\mathbf{E}[\underbrace{L_{t-1,I}^{\eta}}_{=0} \mid \mathbb{H}_{t-1,I}^{\eta}]\right)\right]$$

$$= \sum_{E(I,\eta)\in\mathcal{A}_1} \exp(0) + \sum_{t=2}^{s}\sum_{E(I,\eta)\in\mathcal{A}_t\backslash\mathcal{A}_{t-1}} \exp(0)$$

$$= |\mathcal{A}_1| + \sum_{t=2}^{s}|\mathcal{A}_t \setminus \mathcal{A}_{t-1}| \leq \sum_{t=1}^{s}|\mathcal{A}_t|$$

$$\overset{(i)}{\leq} \sum_{t=1}^{s}(1 + \lfloor\log_2 t\rfloor)\cdot\left(1 + \lceil\tfrac{1}{2}\log_2 t\rceil\right) \leq \sum_{t=1}^{s}(1 + \log_2 t)^2 \overset{(ii)}{\leq} 4s^2\,, \tag{69}$$

where (i) applies (26), and (ii) is due to $1 + \log_2 t \leq 2\sqrt{t} \; \forall t \geq 1$. Since $\exp(x) > 0$ for $x \in \mathbb{R}$, Eq. (69) implies that for any GC interval $I = [r, s] \in \mathcal{I}$ and learning rate $\eta \in \mathcal{S}(|I|)$,

$$\exp\left(-\mathbf{E}[L_{s,I}^{\eta} \mid \mathbb{H}_{s,I}^{\eta}]\right) = \exp\left(-\sum_{t=r}^{s}\mathbf{E}[\ell_t^{\eta}(\boldsymbol{x}_{t,I}^{\eta}) \mid \mathbb{F}_{t,I}^{\eta}]\right) \leq 4s^2. \tag{70}$$

Taking the logarithm of both sides completes the proof. $\qquad\square$

## C.5   Proof of Corollary 1

We first restate Corollary 1 and then provide the proof. Recall that for clarity we drop the $\lceil\cdot\rceil$ operators from the expressions for B and assume without loss of generality the expressions take integer values.

**Corollary 1.** *Consider any horizon $T \in \mathbb{N}^+$ and assume that, for all $t \in [T]$, the loss $f_t$ is convex, or strongly-convex with $\arg\min_{\boldsymbol{x}\in\mathbb{R}^d} f_t(\boldsymbol{x}) \in \Theta$. We refer to the second scenario as the strongly-convex (SC) case. Then, Algorithm 1 tuned with parameter B satisfies the following regret guarantees:*

**Switching.** $\quad \mathsf{B} = \frac{T}{S} \Longrightarrow R^{swi}(T, S) \lesssim \begin{cases} \sqrt{d}S^{\frac{1}{4}}T^{\frac{3}{4}} + d^2 S \\ d\sqrt{ST} + d^2 S \quad (SC) \end{cases}$

**Dynamic.** $\begin{cases} \mathsf{B} = \frac{T}{S} \vee \left(\frac{\sqrt{d}T}{\Delta}\right)^{\frac{4}{5}} \Rightarrow R^{dyn}(T,\Delta,S) \lesssim R^{swi}(T,S) \wedge (d^{\frac{2}{5}}\Delta^{\frac{1}{5}}T^{\frac{4}{5}} + d^{\frac{8}{5}}\Delta^{\frac{4}{5}}T^{\frac{1}{5}}) \\ \mathsf{B} = \frac{T}{S} \vee \left(\frac{dT}{\Delta}\right)^{\frac{2}{3}} \Rightarrow R^{dyn}(T,\Delta,S) \lesssim R^{swi}(T,S) \wedge (d^{\frac{2}{3}}\Delta^{\frac{1}{3}}T^{\frac{2}{3}} + d^{\frac{4}{3}}\Delta^{\frac{2}{3}}T^{\frac{1}{3}}) \quad (SC) \end{cases}$

**Path-length.** $\begin{cases} \mathsf{B} = \left(\frac{r\sqrt{d}T}{P}\right)^{\frac{4}{5}} \Rightarrow R^{path}(T,P) \lesssim r^{-\frac{1}{5}}d^{\frac{2}{5}}P^{\frac{1}{5}}T^{\frac{4}{5}} + r^{-\frac{4}{5}}d^{\frac{8}{5}}P^{\frac{4}{5}}T^{\frac{1}{5}} \\ \mathsf{B} = \left(\frac{rdT}{P}\right)^{\frac{2}{3}} \Rightarrow R^{path}(T,P) \lesssim r^{-\frac{1}{3}}d^{\frac{2}{3}}P^{\frac{1}{3}}T^{\frac{2}{3}} + r^{-\frac{2}{3}}d^{\frac{4}{3}}P^{\frac{2}{3}}T^{\frac{1}{3}} \quad (SC)\,. \end{cases}$

*Proof of Corollary 1.* We begin by applying the first result in Proposition 1 with the adaptive regret guarantees in Theorem 1 to obtain switching regret guarantees. For known $S$, Algorithm 1 with parameter $\mathsf{B} = \frac{T}{S}$ achieves in the general convex case,

$$R^{swi}(T, S) \leq 2C\sqrt{d}S^{\frac{1}{4}}T^{\frac{3}{4}} + 2(C_1 + \tfrac{C_2}{d} + \tfrac{4}{d^2})d^2 S\,, \tag{71}$$

and in the case where $f_t$ is $\alpha$-strongly-convex and $\arg\min_{\boldsymbol{x}\in\mathbb{R}^d} f_t(\boldsymbol{x}) \in \Theta$ for all $t \in [T]$,

$$R^{swi}(T, S) \leq 2C'd\sqrt{ST} + 2(C_1' + \tfrac{C_2'}{d} + \tfrac{4}{d^2})d^2 S\,, \tag{72}$$

where $C, C_1, C_2, C', C_1', C_2' > 0$ are the terms defined in (35)–(37) and (42)–(44) which are polylog-arithmic in $T$ and B. When $S$ and $\Delta$ are both known, we use (71)–(72) and apply the second result in Proposition 1 to bound $R^{dyn}(T, \Delta, S)$. Specifically, for general convex losses, Algorithm 1 with $\mathsf{B} = \frac{T}{S} \vee \left(\frac{\sqrt{d}T}{\Delta}\right)^{\frac{4}{5}}$ yields

$$R^{dyn}(T, \Delta, S) \leq R^{swi}(T, S) \wedge F^{dyn}(T, \Delta)\,,$$

where $F^{\mathsf{dyn}}(T,\Delta) := (2C+1)d^{\frac{2}{5}}\Delta^{\frac{1}{5}}T^{\frac{4}{5}} + 2(C_1 + \frac{C_2}{d} + \frac{4}{d^2})d^{\frac{8}{5}}\Delta^{\frac{4}{5}}T^{\frac{1}{5}}$. For strongly-convex losses with minimizers inside $\Theta$, Algorithm 1 with $\mathsf{B} = \frac{T}{S} \vee \left(\frac{dT}{\Delta}\right)^{\frac{2}{3}}$ gives

$$R^{\mathsf{dyn}}(T,\Delta,S) \leq R^{\mathsf{swi}}(T,S) \wedge F^{\mathsf{dyn}}_{\mathsf{sc}}(T,\Delta).$$

where $F^{\mathsf{dyn}}_{\mathsf{sc}}(T,\Delta) := (2C'+1)d^{\frac{2}{3}}\Delta^{\frac{1}{3}}T^{\frac{2}{3}} + 2(C'_1 + \frac{C'_2}{d} + \frac{4}{d^2})d^{\frac{4}{3}}\Delta^{\frac{2}{3}}T^{\frac{1}{3}}$. Finally, for known $P$ we use (71)–(72) and apply the third result in Proposition 1 to bound $R^{\mathsf{path}}(T,P)$. For the general convex case, taking $\mathsf{B} = \left(\frac{r\sqrt{d}T}{P}\right)^{\frac{4}{5}}$ gives

$$R^{\mathsf{path}}(T,P) \leq (2C+1)r^{-\frac{1}{5}}d^{\frac{2}{5}}P^{\frac{1}{5}}T^{\frac{4}{5}} + 2(C_1 + \frac{C_2}{d} + \frac{4}{d^2})r^{-\frac{4}{5}}d^{\frac{8}{5}}P^{\frac{4}{5}}T^{\frac{1}{5}}.$$

For strongly-convex losses with minimizers inside $\Theta$, taking $\mathsf{B} = \left(\frac{rdT}{P}\right)^{\frac{2}{3}}$ yields

$$R^{\mathsf{path}}(T,P) \leq (2C'+1)r^{-\frac{1}{3}}d^{\frac{2}{3}}P^{\frac{1}{3}}T^{\frac{2}{3}} + 2(C'_1 + \frac{C'_2}{d} + \frac{4}{d^2})r^{-\frac{2}{3}}d^{\frac{4}{3}}P^{\frac{2}{3}}T^{\frac{1}{3}}.$$

$\square$

### C.6  Parameter-free upper bounds

Corollary 1 presents the optimal choice of parameter $\mathsf{B}$ for TEWA-SE when $S$, $\Delta$ and $P$ are known. When the non-stationarity measures are unknown, the optimal $\mathsf{B}$ cannot be directly computed, and we therefore employ the Bandit-over-Bandit (BoB) framework from [27] to adaptively select $\mathsf{B}$ from a prespecified set $\mathcal{B} = \{2^i : i = 0, 1, \ldots, \lfloor \log_2 T \rfloor\}$. BoB has been used in [66] in a similar fashion to obtain parameter-free algorithms. Specifically, BoB divides the time horizon into $E = \lceil T/L \rceil$ epochs each with length $L$, denoted by $(I_e)_{e=1}^{E}$ (where the last epoch may be shorter than $L$). In the first epoch, it runs TEWA-SE with $\mathsf{B} = \mathsf{B}_1$ which is randomly selected from $\mathcal{B}$. For subsequent epochs, it uses the cumulative empirical loss on the current epoch $e-1$ to select $\mathsf{B}_e \in \mathcal{B}$ for the next epoch via EXP3 [59]. That is, BoB computes

$$p_{e,i} = (1-\gamma)\frac{s_{e,i}}{\sum_{i' \in [|\mathcal{B}|]} s_{e,i'}} + \frac{\gamma}{|\mathcal{B}|} \quad \forall i \in [|\mathcal{B}|], \quad \text{with} \quad \gamma = 1 \wedge \sqrt{\frac{|\mathsf{B}|\ln(|\mathcal{B}|)}{(\mathrm{e}-1)E}}, \quad (73)$$

where e denotes the base of the exponential function, and then samples $i_e = i$ with probability $p_{e,i}$ yielding $\mathsf{B}_e = 2^{i_e-1}$.[5] For $i \in [|\mathcal{B}|]$, initialized with $s_{0,i} = 1$, the quantity $s_{e,i}$ for $e \in \mathbb{N}^+$ is updated by computing

$$s_{e+1,i} = s_{e,i}\exp\left(\frac{\gamma}{|\mathcal{B}|}\widehat{r}_{e,i}\right), \quad (74)$$

where with $M_T = 1 + 2\sigma\sqrt{\log(T+1)}$, the importance-weighted reward $\widehat{r}_{e,i}$ takes the form

$$\widehat{r}_{e,i} = \begin{cases} \left(\frac{1}{2} + \frac{1}{2LM_T}\sum_{t \in I_e}(1-y_t)\right)/p_{e,i} & \text{if } i = i_e \\ 0 & \text{otherwise}. \end{cases} \quad (75)$$

Note that conditioned on the event $\Lambda_T = \left\{|\xi_t| \leq 2\sigma\sqrt{\log(T+1)}, \forall t \in [T]\right\}$ defined above (10), the absolute total reward in each epoch is bounded by $Q := \max_{e \in [E]}\left|\sum_{t \in I_e}(1-y_t)\right| \leq LM_T$, which ensures the rescaled reward $\frac{1}{2} + \frac{1}{2LM_T}\sum_{t \in I_e}(1-y_t)$ in (75) remains bounded within $[0,1]$. The pseudo-code for TEWA-SE equipped with BoB is provided in Algorithm 4, with theoretical guarantees detailed in Corollary 3.

**Corollary 3** (TEWA-SE with BoB). *Consider any horizon $T \in \mathbb{N}^+$ and assume that, for all $t \in [T]$, the loss $f_t$ is convex, or strongly-convex with $\arg\min_{\boldsymbol{x} \in \mathbb{R}^d} f_t(\boldsymbol{x}) \in \Theta$ (referred to as the strongly-convex (SC) case).*

*Then, for the general convex case, Algorithm 4 with epoch size $L = (dT)^{\frac{2}{3}}$ attains all the regret bounds from Corollary 1 plus an additional term of $d^{\frac{1}{3}}T^{\frac{5}{6}} + d^{\frac{4}{3}}T^{\frac{1}{3}}$. For the SC case, Algorithm 4 with epoch size $L = d\sqrt{T}$ satisfies all the regret bounds from Corollary 1 plus an additional term of $d^{\frac{1}{2}}T^{\frac{3}{4}} + d\sqrt{T}$. Both results omitted polylogarithmic factors.*

---

[5]We adopt clipping (by $\gamma$) following [27, 59], though $\gamma = 0$ suffices as discussed in [77] and [9, Section 11.6].

---

**Algorithm 4** TEWA equipped with Bandit-over-Bandit (BoB)

---

**Input:** $d, T, L, E = \lceil T/L \rceil, (I_e)_{e=1}^E, \mathcal{B} = \{2^i : i = 0, 1, \ldots, \lfloor \log_2 T \rfloor\}$, and $\gamma \in (0, 1)$ as defined in (73)

**Initialize:** $s_{0,i} = 1 \; \forall i \in [|\mathcal{B}|]$

1: **for** $e = 1, 2, \ldots, E$ **do**
2:      Compute $p_{e,i}$ according to (73) $\forall i \in [|\mathcal{B}|]$
3:      Sample $i_e = i$ with probability $p_{e,i}$, and select $\mathsf{B}_e = 2^{i_e - 1} \in \mathcal{B}$
4:      **for** $t \in I_e$ **do**
5:          Run TEWA-SE with $\mathsf{B} = \mathsf{B}_e$ to select action $\boldsymbol{z}_t$ and observe losses $y_t = f_t(\boldsymbol{z}_t) + \xi_t$
6:      **end for**
7:      Update $s_{e+1,i}$ according to (74) $\forall i \in [|\mathcal{B}|]$
8: **end for**

---

*Proof of Corollary 3.* For brevity, we suppress terms that are polylogarithmic in $T$ using $\lesssim$ in this proof. For all $\mathsf{B}^\dagger \in \mathcal{B}$, we have

$$
R^{\mathsf{dyn}}(T) = \sum_{t=1}^T \mathbf{E}\left[f_t(\boldsymbol{z}_t) - \min_{\boldsymbol{z} \in \Theta} f_t(\boldsymbol{z})\right]
$$

$$
= \sum_{t=1}^T \mathbf{E}\left[f_t(\boldsymbol{z}_t) - f_t\left(\boldsymbol{z}_t(\mathsf{B}^\dagger)\right)\right] + \sum_{t=1}^T \mathbf{E}\left[f_t\left(\boldsymbol{z}_t(\mathsf{B}^\dagger)\right) - \min_{\boldsymbol{z} \in \Theta} f_t(\boldsymbol{z})\right]
$$

$$
= \underbrace{\sum_{e=1}^E \sum_{t \in I_e} \mathbf{E}\left[f_t\left(\boldsymbol{z}_t(\mathsf{B}_e)\right) - f_t\left(\boldsymbol{z}_t(\mathsf{B}^\dagger)\right)\right]}_{\text{term I}} + \underbrace{\sum_{e=1}^E \sum_{t \in I_e} \mathbf{E}\left[f_t\left(\boldsymbol{z}_t(\mathsf{B}^\dagger)\right) - \min_{\boldsymbol{z} \in \Theta} f_t(\boldsymbol{z})\right]}_{\text{term II}},
$$

$$
\tag{76}
$$

where $\boldsymbol{z}_t(\mathsf{B}_e)$ represents the actual action taken by TEWA-SE in round $t$ of epoch $e$, and $\boldsymbol{z}_t(\mathsf{B}^\dagger)$ denotes the hypothetical action that TEWA-SE would have chosen had its $\mathsf{B}$ parameter been set to $\mathsf{B}^\dagger$. Term I in (76) can be bounded by applying the classical analysis of EXP3 from [59, Corollary 3.2], combined with (28), as follows

$$
\forall \mathsf{B}^\dagger \in \mathcal{B}: \quad \text{term I} \leq 4\sqrt{e-1}\sqrt{E|\mathcal{B}|\log|\mathcal{B}|} \cdot \mathbf{E}[Q]
$$

$$
\lesssim \sqrt{E} \cdot \left(\mathbf{P}(\Lambda_T)\mathbf{E}[Q \mid \Lambda_T] + \mathbf{P}(\Lambda_T^c)\mathbf{E}[Q \mid \Lambda_T^c]\right)
$$

$$
\lesssim \sqrt{T/L} \cdot \left(LM_T + \tfrac{2}{T} \cdot L\right) \lesssim \sqrt{TL}.
$$

$$
\tag{77}
$$

To bound term II, we introduce shorthand $F^{\mathsf{ada}}(\mathsf{B}, T)$ to refer to the upper bound on $R^{\mathsf{ada}}(\mathsf{B}, T)$ in Theorem 1, and $F^{\mathsf{swi}}(T, S), F^{\mathsf{dyn}}(T, \Delta, S)$ and $F^{\mathsf{path}}(T, P)$ to refer to the upper bounds on $R^{\mathsf{swi}}(T, S), R^{\mathsf{dyn}}(T, \Delta, S)$ and $R^{\mathsf{path}}(T, P)$ in Corollary 1 for known $S, \Delta$ and $P$. We also use $S_e = 1 + \sum_{t \in I_e} \mathbf{1}(f_t \neq f_{t-1})$. By choosing $\mathsf{B}^\dagger = 2^{i^\dagger}$ in the analysis with $i^\dagger = \lfloor \log_2 \tfrac{T}{S} \rfloor \wedge \lfloor \log_2 L \rfloor$, term II can be bounded in terms of the number of switches $S$ by

$$
\text{term II} \leq \sum_{e=1}^E \left(\lceil \tfrac{L}{\mathsf{B}^\dagger} \rceil + S_e\right) R^{\mathsf{ada}}(\mathsf{B}^\dagger, L) \leq \left(\tfrac{T}{\mathsf{B}^\dagger} + S + E\right) R^{\mathsf{ada}}(\mathsf{B}^\dagger, L)
$$

$$
\leq F^{\mathsf{swi}}(T, S) + \tfrac{T}{L} F^{\mathsf{ada}}(L, L).
$$

$$
\tag{78}
$$

Combining (77) and (78), we obtain

$$
R^{\mathsf{swi}}(T, S) \lesssim F^{\mathsf{swi}}(T, S) + \left[\tfrac{T}{L} F^{\mathsf{ada}}(L, L) + \sqrt{TL}\right]
$$

$$
\lesssim F^{\mathsf{swi}}(T, S) + \begin{cases} d^{\frac{1}{3}}T^{\frac{5}{6}} + d^{\frac{4}{3}}T^{\frac{1}{3}} \\ d^{\frac{1}{2}}T^{\frac{3}{4}} + d\sqrt{T} \quad (\mathsf{SC}), \end{cases}
$$

$$
\tag{79}
$$

where we used $L = (dT)^{\frac{2}{3}}$ for the general convex case, and $L = d\sqrt{T}$ for the strongly-convex case. Following similar steps, by choosing $\mathsf{B}^\dagger = 2^{i^\dagger}$ in the analysis with $i^\dagger = (\lfloor \log_2 \tfrac{T}{S} \rfloor \vee \lfloor \log_2(\mathsf{B}_\Delta) \rfloor) \wedge$

$\lfloor \log_2 L \rfloor$ where $\mathsf{B}_\Delta = \left(\frac{\sqrt{dT}}{\Delta}\right)^{\frac{4}{5}}$ for the general convex case or $\mathsf{B}_\Delta = \left(\frac{dT}{\Delta}\right)^{\frac{2}{3}}$ for the strongly-convex case, we obtain

$$R^{\mathsf{dyn}}(T, \Delta, S) \lesssim F^{\mathsf{dyn}}(T, \Delta, S) + \begin{cases} d^{\frac{1}{3}} T^{\frac{5}{6}} + d^{\frac{4}{3}} T^{\frac{1}{3}} \\ d^{\frac{1}{2}} T^{\frac{3}{4}} + d\sqrt{T} \quad (\mathsf{SC}) \, . \end{cases} \tag{80}$$

The bound on $R^{\mathsf{path}}(T, P)$ can be established analogously. $\qquad\square$

## D   Proofs of lower bounds

We call $\pi = \{\boldsymbol{z}_t\}_{t=1}^\infty$ a randomized procedure if $\boldsymbol{z}_t = \Phi_t(\{\boldsymbol{z}_k\}_{k=1}^{t-1}, \{y_k\}_{k=1}^{t-1})$ where $\Phi_t$ are Borel functions, and $\boldsymbol{z}_1 \in \mathbb{R}^d$ is deterministic. We emphasize that, throughout this section, we assume the noise variables $\{\xi_t\}_{t=1}^T$ are independent with cumulative distribution function $F$ satisfying the condition

$$\int \log\left(\, \mathrm{d}F\left(u\right) / \mathrm{d}F\left(u + v\right)\right) \mathrm{d}F(u) \le I_0 v^2, \qquad |v| < v_0, \tag{81}$$

for some $0 < I_0 < \infty, 0 < v_0 \le \infty$. This condition holds, for instance, if $F$ has a sufficiently smooth density with finite Fisher information. In the special case where $F$ is Gaussian, the inequality (81) holds with $v_0 = \infty$. Note that Gaussian noise also satisfies our sub-Gaussian noise assumption in Section 1, which is used in the proof of the upper bounds.

We first restate and prove Theorem 2, which establishes a lower bound on $R^{\mathsf{dyn}}(T, \Delta, S)$, and then present and prove Theorem 4, which establishes a lower bound on $R^{\mathsf{path}}(T, P)$.

**Theorem 2.** *Let $\Theta = \mathbb{B}^d$. For $\alpha > 0$ denote by $\mathcal{F}_\alpha$ the class of $\alpha$-strongly convex and smooth functions. Let $\pi = \{\boldsymbol{z}_t\}_{t=1}^T$ be any randomized algorithm (see Appendix D for a definition). Then there exists $T_0 > 0$ such that for all $T \ge T_0$ it holds that*

$$\sup_{f_1, \ldots, f_T \in \mathcal{F}_\alpha} R^{\mathsf{dyn}}(T, \Delta, S) \ge c_1 \cdot \left( d\sqrt{ST} \wedge d^{\frac{2}{3}} \Delta^{\frac{1}{3}} T^{\frac{2}{3}} \right), \tag{16}$$

*where $c_1 > 0$ is a constant independent of $d, T, S$ and $\Delta$.*

*Proof of Theorem 2.* Let $\eta_0 : \mathbb{R} \to \mathbb{R}$ be an infinitely many times differentiable function that satisfies

$$\eta_0(x) \begin{cases} = 1 & \text{if } |x| \le 1/4 \, , \\ \in (0, 1) & \text{if } 1/4 < |x| < 1 \, , \\ = 0 & \text{if } |x| \ge 1 \, . \end{cases}$$

Denote by $\Omega = \{-1, 1\}^d$ the set of binary sequences of length $d$, and let $\eta(x) = \int_{-\infty}^x \eta_0(u) \, \mathrm{d}u$. Consider the set of functions $f_{\boldsymbol{\omega}} : \mathbb{R}^d \to \mathbb{R}$ with $\boldsymbol{\omega} = (\omega_1, \ldots, \omega_d) \in \{-1, 1\}^d$ such that:

$$f_{\boldsymbol{\omega}}(\boldsymbol{x}) = \alpha \|\boldsymbol{x}\|^2 + \iota h^2 \left( \sum_{i=1}^d \omega_i \eta \left( \frac{x_i}{h} \right) \right), \qquad \boldsymbol{x} = (x_1, \ldots, x_d), \tag{82}$$

where $h = \min\left(d^{-\frac{1}{2}}, \left(\frac{T}{S}\right)^{-\frac{1}{4}}, \left(\frac{dT}{\Delta}\right)^{-\frac{1}{6}}\right)$, and $\iota > 0$ is to be assigned later. Let $L' = \max_{x \in \mathbb{R}} |\eta''(x)|$. By [78, Lemma 10] we have that if $\iota \le \min\left(1/2\eta(1), \alpha/L'\right)$ then $f_{\boldsymbol{\omega}} \in \mathcal{F}_\alpha$. Moreover, if $\iota \le \alpha/2$, the equation $\nabla f_{\boldsymbol{\omega}}(\boldsymbol{x}) = 0$ has the solution

$$\boldsymbol{x}^*(\boldsymbol{\omega}) = (x_1^*(\boldsymbol{\omega}), \ldots, x_d^*(\boldsymbol{\omega})), \quad \text{with} \quad x_i^*(\boldsymbol{\omega}) = -\frac{h\iota\omega_i}{2\alpha} \quad \text{for } 1 \le i \le d \, . \tag{83}$$

This is the unique minimizer of $f_{\boldsymbol{\omega}}$ and belongs to $\boldsymbol{x}^*(\boldsymbol{\omega}) \in \Theta = \mathbb{B}^d$ because

$$\|\boldsymbol{x}^*(\boldsymbol{\omega})\|^2 \le \frac{h^2 \iota^2 d}{4\alpha^2} \le \frac{1}{16}.$$

We consider the following adversarial protocol. At the beginning of the game, the adversary selects $N_c = \min(S, (T\Delta^2/d^2)^{\frac{1}{3}})$ points from $\Omega$, sampled uniformly at random with replacement. Here

without loss of generality we assumed that $(T\Delta^2)^{\frac{1}{3}}$ is a positive integer. Denote these points by $\{\boldsymbol{\omega}_k\}_{k=1}^{N_c}$, and then for each $k = 1, 2, \ldots, N_c$, let

$$f_{(k-1)T/N_c+1} = \cdots = f_{kT/N_c} = f_{\boldsymbol{\omega}_k}.$$

For any $\boldsymbol{\omega}, \boldsymbol{\omega}' \in \Omega$ let $\rho(\boldsymbol{\omega}, \boldsymbol{\omega}') = \sum_{i=1}^d \mathbb{1}(\omega_i \neq \omega_i')$ be the Hamming distance between $\boldsymbol{\omega}$ and $\boldsymbol{\omega}'$, with $\boldsymbol{\omega} = (\omega_1, \ldots, \omega_d)$ and $\boldsymbol{\omega}' = (\omega_1', \ldots, \omega_d')$. By construction, $N_c \leq S$ and

$$\sum_{k=2}^{N_c} \max_{\boldsymbol{x} \in \Theta} |f_{\boldsymbol{\omega}_{k-1}}(\boldsymbol{x}) - f_{\boldsymbol{\omega}_k}(\boldsymbol{x})| \leq 2\iota h^2 \eta(1) \sum_{k=2}^{N_c} \rho(\boldsymbol{\omega}_{k-1}, \boldsymbol{\omega}_k) \leq \Delta.$$

For any fixed $\boldsymbol{\omega}_1, \ldots, \boldsymbol{\omega}_{N_c} \in \Omega$, and $1 \leq t \leq T$, denote $\Gamma = [\boldsymbol{\omega}_1 \mid \ldots \mid \boldsymbol{\omega}_{N_c}]$ as the matrix whose columns are the $\boldsymbol{\omega}_k$'s. Denote by $\mathbf{P}_{\Gamma,t}$ the probability measure corresponding to the joint distribution of $\{\boldsymbol{z}_k, y_k\}_{k=1}^t$ where $y_k = f_k(\boldsymbol{z}_k) + \xi_k$ with independent identically distributed $\xi_k$'s such that (81) holds and $\boldsymbol{z}_k$'s are chosen by the algorithm $\pi$. We have

$$\mathrm{d}\mathbf{P}_{\Gamma,t}(\boldsymbol{z}_{1:t}, y_{1:t}) = \mathrm{d}F(y_1 - f_1(\boldsymbol{z}_1)) \prod_{\tau=2}^t \mathrm{d}F(y_\tau - f_\tau(\Phi_\tau(\boldsymbol{z}_{1:\tau-1}, y_{1:\tau-1})))$$

$$= \mathrm{d}F(y_1 - f_{\boldsymbol{\omega}_1}(\boldsymbol{z}_1)) \prod_{\tau=2}^t \mathrm{d}F(y_\tau - f_{\boldsymbol{\omega}_{k_\tau}}(\Phi_\tau(\boldsymbol{z}_{1:\tau-1}, y_{1:\tau-1}))), \quad (84)$$

where $k_\tau = \lfloor (\tau - 1)N_c/T \rfloor + 1$. (We omit explicit mention of the dependence of $\mathbf{P}_{\Gamma,t}$ and $\Phi_\tau$ on $\boldsymbol{z}_2, \ldots, \boldsymbol{z}_{\tau-1}$, since $\boldsymbol{z}_\tau$ for $\tau \geq 2$ is a Borel function of $\boldsymbol{z}_1, y_1, \ldots, \boldsymbol{z}_{\tau-1}, y_{\tau-1}$.) Let $\mathbf{E}_{\Gamma,t}$ denote the expectation w.r.t. $\mathbf{P}_{\Gamma,t}$.

Note that by $\alpha$-strong convexity of $f_{\boldsymbol{\omega}}$ and the fact that $\boldsymbol{x}^*(\boldsymbol{\omega}) \in \arg\min_{\boldsymbol{x} \in \mathbb{R}^d} f_{\boldsymbol{\omega}}(\boldsymbol{x})$ from (83), we have

$$\sum_{t=1}^T \mathbf{E}_{\Gamma,t}\left[f_{\boldsymbol{\omega}_{k_t}}(\boldsymbol{z}_t) - \min_{\boldsymbol{x} \in \Theta} f_{\boldsymbol{\omega}_{k_t}}(\boldsymbol{x})\right] \geq \frac{\alpha}{2} \sum_{t=1}^T \mathbf{E}_{\Gamma,t}\left[\|\boldsymbol{z}_t - \boldsymbol{x}^*(\boldsymbol{\omega}_{k_t})\|^2\right]. \quad (85)$$

Define the nearest-neighbour estimator

$$\hat{\boldsymbol{\omega}}_t \in \arg\min_{\boldsymbol{\omega} \in \Omega} \|\boldsymbol{z}_t - \boldsymbol{x}^*(\boldsymbol{\omega})\|.$$

Using this combined with the triangle inequality, we have $\|\boldsymbol{x}^*(\hat{\boldsymbol{\omega}}_t) - \boldsymbol{x}^*(\boldsymbol{\omega}_{k_t})\| \leq \|\boldsymbol{z}_t - \boldsymbol{x}^*(\hat{\boldsymbol{\omega}}_t)\| + \|\boldsymbol{z}_t - \boldsymbol{x}^*(\boldsymbol{\omega}_{k_t})\| \leq 2\|\boldsymbol{z}_t - \boldsymbol{x}^*(\boldsymbol{\omega}_{k_t})\|$. Together with (83) this implies that

$$\mathbf{E}_{\Gamma,t}\left[\|\boldsymbol{z}_t - \boldsymbol{x}^*(\boldsymbol{\omega}_{k_t})\|^2\right] \geq \frac{1}{4}\mathbf{E}_{\Gamma,t}\left[\|\boldsymbol{x}^*(\hat{\boldsymbol{\omega}}_t) - \boldsymbol{x}^*(\boldsymbol{\omega}_{k_t})\|^2\right] = \frac{h^2\iota^2}{4\alpha^2}\mathbf{E}_{\Gamma,t}\left[\rho(\hat{\boldsymbol{\omega}}_t, \boldsymbol{\omega}_{k_t})\right].$$

Summing over $1, \ldots, T$, then taking the maximum over $\Gamma = [\boldsymbol{\omega}_1 \mid \ldots \mid \boldsymbol{\omega}_{N_c}]$ and then the minimum over all estimators $\hat{\boldsymbol{\omega}}_1, \ldots, \hat{\boldsymbol{\omega}}_T$ with values in $\Omega$, we get

$$\max_{\Gamma \in \Omega^{N_c}} \sum_{t=1}^T \mathbf{E}_{\Gamma,t}\left[\|\boldsymbol{z}_t - \boldsymbol{x}^*(\boldsymbol{\omega}_{k_t})\|^2\right] \geq \frac{h^2\iota^2}{4\alpha^2} \underbrace{\min_{\hat{\boldsymbol{\omega}}_1, \ldots, \hat{\boldsymbol{\omega}}_T \in \Omega} \max_{\Gamma \in \Omega^{N_c}} \sum_{t=1}^T \sum_{i=1}^d \mathbf{E}_{\Gamma,t}\left[\mathbb{1}(\hat{\omega}_{t,i} \neq \omega_{k_t,i})\right]}_{\text{term I}}.$$

$$(86)$$

For term I, lower bounding the maximum with the average we can write

$$\text{term I} \geq 2^{-dN_c} \min_{\hat{\boldsymbol{\omega}}_1, \ldots, \hat{\boldsymbol{\omega}}_T \in \Omega} \sum_{t=1}^T \sum_{\Gamma \in \Omega^{N_c}} \sum_{i=1}^d \mathbf{E}_{\Gamma,t}\left[\mathbb{1}(\hat{\omega}_{t,i} \neq \omega_{k_t,i})\right]$$

$$\geq 2^{-dN_c} \sum_{t=1}^T \sum_{\Gamma \in \Omega^{N_c}} \sum_{i=1}^d \min_{\hat{\omega}_{t,i} \in \{-1,1\}} \mathbf{E}_{\Gamma,t}\left[\mathbb{1}(\hat{\omega}_{t,i} \neq \omega_{k_t,i})\right].$$

Next, for each $i = 1, \ldots, d$, define $\Gamma_i^{k_t} = \{[\boldsymbol{\omega}_1|\ldots|\boldsymbol{\omega}_{N_c}] : \boldsymbol{\omega}_1, \ldots, \boldsymbol{\omega}_{N_c} \in \Omega, \omega_{k_t,i} = 1\}$. Given any $\Gamma \in \Gamma_i^{k_t}$, let $\bar{\Gamma} = [\bar{\boldsymbol{\omega}}_1|\ldots|\bar{\boldsymbol{\omega}}_{N_c}]$ such that $\bar{\omega}_{k,j} = \omega_{k,j}$ for any $k \neq k_t$, and let $\bar{\omega}_{k_t,i} = -1$ and $\bar{\omega}_{k_t,j} = \omega_{k_t,j}$ for $j \neq i$. Hence,

$$
\begin{aligned}
\text{term I} &\geq 2^{-dN_c} \sum_{t=1}^{T} \sum_{\Gamma \in \Omega^{N_c}} \sum_{i=1}^{d} \min_{\hat{\omega}_{t,i} \in \{-1,1\}} \left( \mathbf{E}_{\Gamma,t} \left[ \mathbb{1}\left(\hat{\omega}_{t,i} \neq 1\right) \right] + \mathbf{E}_{\bar{\Gamma},t} \left[ \mathbb{1}\left(\hat{\omega}_{t,i} \neq -1\right) \right] \right) \\
&\geq \frac{1}{2} \sum_{t=1}^{T} \sum_{i=1}^{d} \min_{\Gamma \in \Gamma_i^{k_t}} \min_{\hat{\omega}_{t,i} \in \{-1,1\}} \left( \mathbf{E}_{\Gamma,t} \left[ \mathbb{1}\left(\hat{\omega}_{t,i} \neq 1\right) \right] + \mathbf{E}_{\bar{\Gamma},t} \left[ \mathbb{1}\left(\hat{\omega}_{t,i} \neq -1\right) \right] \right).
\end{aligned}
$$

Thus, we can write

$$
\begin{aligned}
\text{KL}\left(\mathbf{P}_{\Gamma,t} || \mathbf{P}_{\bar{\Gamma},t}\right) &= \int \log\left(\frac{d\mathbf{P}_{\Gamma,t}}{d\mathbf{P}_{\bar{\Gamma},t}}\right) d\mathbf{P}_{\Gamma,t} \\
&= \int \left[ \log\left(\frac{dF(y_1 - f_{\boldsymbol{\omega}_1}(\boldsymbol{z}_1))}{dF(y_1 - f_{\bar{\boldsymbol{\omega}}_1}(\boldsymbol{z}_1))}\right) + \right. \\
&\qquad \left. + \sum_{\tau=2}^{t} \log\left(\frac{dF(y_\tau - f_{\boldsymbol{\omega}_{k_\tau}}(\Phi_\tau(\boldsymbol{z}_{1:\tau-1}, y_{1:\tau-1})))}{dF(y_\tau - f_{\bar{\boldsymbol{\omega}}_{k_\tau}}(\Phi_\tau(\boldsymbol{z}_{1:\tau-1}, y_{1:\tau-1})))}\right) \right] \\
&\qquad dF\left(y_1 - f_{\boldsymbol{\omega}_1}(\boldsymbol{z}_1)\right) \prod_{\tau=2}^{t} dF\left(y_\tau - f_{\boldsymbol{\omega}_{k_\tau}}(\Phi_\tau(\boldsymbol{z}_{1:\tau-1}, y_{1:\tau-1}))\right) \\
&\leq I_0 \sum_{\tau=1}^{t} \max_{\boldsymbol{x} \in \Theta} |f_{\boldsymbol{\omega}_{k_\tau}}(\boldsymbol{x}) - f_{\bar{\boldsymbol{\omega}}_{k_\tau}}(\boldsymbol{x})|^2 \leq 4TN_c^{-1} I_0 \iota^2 h^4 \eta^2(1).
\end{aligned}
$$

Since $h \leq \min\left(\left(\frac{S}{T}\right)^{\frac{1}{4}}, \left(\frac{\Delta}{dT}\right)^{\frac{1}{6}}\right)$, and by choosing $\iota \leq \sqrt{\log(2)/(4I_0 \eta^2(1))}$, we have $\text{KL}(\mathbf{P}_{\Gamma,t} || \mathbf{P}_{\bar{\Gamma},t}) \leq \log(2)$. Hence, Theorem 2.12 of [79] gives

$$
\text{term I} \geq \frac{Td}{4} \exp(-\log(2)) = \frac{Td}{8}.
$$

Substituting this into (86) and our overall bound (85) yields

$$
\max_{\Gamma \in \Omega^{N_c}} \sum_{t=1}^{T} \mathbf{E}_{\Gamma,t} \left[ f_{\boldsymbol{\omega}_{k_t}}(\boldsymbol{z}_t) - \min_{\boldsymbol{x} \in \Theta} f_{\boldsymbol{\omega}_{k_t}}(\boldsymbol{x}) \right] \geq \frac{\alpha}{2} \cdot \frac{h^2 \iota^2}{4\alpha^2} \cdot \frac{Td}{8} = \frac{h^2 \iota^2 Td}{64\alpha}.
$$

Finally, substituting the definition of $h$ and noting that $\iota$ is independent of $d, T, S$ and $\Delta$ completes the proof. $\qquad\square$

**Theorem 4.** *Let $\Theta = \mathbb{B}^d$. For $\alpha > 0$ denote by $\mathcal{F}_\alpha$ the class of $\alpha$-strongly convex and smooth functions. Let $\pi = \{\boldsymbol{z}_t\}_{t=1}^{T}$ be any randomized algorithm. Then there exists $T_0 > 0$ such that for all $T \geq T_0$ it holds that*

$$
\sup_{f_1,\ldots,f_T \in \mathcal{F}_\alpha} R^{\text{path}}(T, P) \geq c_2 \cdot (d^2 P)^{\frac{2}{5}} T^{\frac{3}{5}},
$$

*where $c_2 > 0$ is a constant indepedent of $d, T$ and $P$.*

*Proof of Theorem 4.* The proof uses the same notation and follows the same steps as in the proof of Theorem 2, but with different choices for the parameters $h$ and $N_c$. Define the set of functions $f_{\boldsymbol{\omega}} : \mathbb{R}^d \to \mathbb{R}$ with $\boldsymbol{\omega} \in \{-1, 1\}^d$ as they are defined in (82), and choose $h = \min(d^{-\frac{1}{2}}, \frac{P}{N_c\sqrt{d}})$ and $N_c = \lfloor P^{\frac{4}{5}} T^{\frac{1}{5}} d^{-\frac{2}{5}} \rfloor$. Then we have that

$$
\sum_{k=2}^{N_c} \|\boldsymbol{x}^*(\boldsymbol{\omega}_{k-1}) - \boldsymbol{x}^*(\boldsymbol{\omega}_k)\| = \frac{h\iota}{\alpha} \sum_{k=2}^{N_c} \sqrt{\rho(\boldsymbol{\omega}_{k-1}, \boldsymbol{\omega}_k)} \leq \frac{h\iota}{\alpha} \sqrt{d} N_c \leq P, \tag{87}
$$

for any $\iota \leq \frac{\alpha}{2}$. Following similar steps as in the proof of Theorem 2 for large enough $T$ (when $h = \frac{P}{N_c\sqrt{d}}$) we get

$$\max_{\Gamma \in \Omega^{N_c}} \sum_{t=1}^{T} \mathbf{E}_{\Gamma,t} \left[ f_{\boldsymbol{\omega}_{k_t}}(\boldsymbol{z}_t) - \min_{\boldsymbol{x} \in \Theta} f_{\boldsymbol{\omega}_{k_t}}(\boldsymbol{x}) \right] \geq \frac{h^2 \iota^2 T d}{64\alpha} \geq c_2 (d^2 P)^{\frac{2}{5}} T^{\frac{3}{5}} ,$$

where $c_2 > 0$ is independent of $d, T$ and $P$. $\qquad\square$

# E   Proofs for clipped Exploration by Optimization

We restate and prove Theorem 3 which establishes an adaptive regret guarantee for cExO. In this section, we use $\langle \boldsymbol{p}, f_t \rangle = \mathbf{E}_{\boldsymbol{z} \sim \boldsymbol{p}}[f_t(\boldsymbol{z})]$ where $\boldsymbol{p}$ belongs to a probability simplex.

**Theorem 3.** *For* $T \in \mathbb{N}^+$ *and* $\mathsf{B} \in [T]$, *Algorithm 3 calibrated with* $\varepsilon = \frac{1}{T}$, $\gamma = \frac{1}{T|\mathcal{C}|}$, $\eta = \sqrt{\log(\gamma^{-1})/(d^4 \log(dT)\mathsf{B})}$ *and* $\log|\mathcal{C}| = \mathcal{O}(d \log(dT^2))$ *satisfies*

$$R^{ada}(\mathsf{B}, T) \lesssim d^{\frac{5}{2}} \sqrt{\mathsf{B}} . \tag{19}$$

*Proof of Theorem 3.* Consider an arbitrary interval $[a, b]$ of length $b - a + 1 \leq \mathsf{B}$, and notice that for any $\boldsymbol{q}^\star \in \tilde{\Delta}$,

$$\max_{\boldsymbol{u} \in \Theta} \sum_{t=a}^{b} \mathbf{E}[f_t(\boldsymbol{z}_t) - f_t(\boldsymbol{u})] = \underbrace{\sum_{t=a}^{b} \langle \boldsymbol{p}_t - \boldsymbol{q}^\star, f_t \rangle}_{\text{term I}} + \underbrace{\sum_{t=a}^{b} \mathbf{E}_{\boldsymbol{z} \sim \boldsymbol{q}^\star}[f_t(\boldsymbol{z})] - \min_{\boldsymbol{u} \in \Theta} \sum_{t=a}^{b} f_t(\boldsymbol{u})}_{\text{term II}} . \tag{88}$$

In what follows, we choose a suitable $\boldsymbol{q}^\star$ and bound term I and term II separately.

Recall that the covering set $\mathcal{C}$ is assumed in Section 3 to have a discretization error of $\varepsilon$, implying that there exists a $\boldsymbol{u}_\mathcal{C} \in \mathcal{C}$ such that $\sum_{t=a}^{b} f_t(\boldsymbol{u}_\mathcal{C}) - \min_{\boldsymbol{u} \in \Theta} \sum_{t=a}^{b} f_t(\boldsymbol{u}) \leq \varepsilon \mathsf{B}$. Define $\boldsymbol{q}^\star \in \tilde{\Delta}$ to be the distribution with probability mass given by

$$q^\star(\boldsymbol{z}) = \begin{cases} 1 - \gamma(|\mathcal{C}| - 1) & \text{if } \boldsymbol{z} = \boldsymbol{u}_\mathcal{C} \\ \gamma & \text{otherwise} . \end{cases} \tag{89}$$

This construction ensures that

$$\text{term II} \leq (\varepsilon + 2\gamma|\mathcal{C}|)\mathsf{B} . \tag{90}$$

To bound term I, we first apply Lemma 8 to the sequence of Online Mirror Descent (OMD) updates $\boldsymbol{q}_t \in \tilde{\Delta}$ and the sequence of loss estimates $\widehat{\boldsymbol{s}}_t$ to obtain

$$\sum_{t=a}^{b} \langle \boldsymbol{q}_t - \boldsymbol{q}^\star, \widehat{\boldsymbol{s}}_t \rangle \leq \frac{1}{\eta} \left( \text{KL}(\boldsymbol{q}^\star \| \boldsymbol{q}_a) + \sum_{t=a}^{b} S_{\boldsymbol{q}_t}(\eta \widehat{\boldsymbol{s}}_t) \right) , \tag{91}$$

where by the definition of $q^\star(\cdot)$ in (89), we have

$$\begin{aligned}
\text{KL}(\boldsymbol{q}^\star \| \boldsymbol{q}_a) &= \sum_{\boldsymbol{z} \in \mathcal{C}} q^\star(\boldsymbol{z}) \log \left( \frac{q^\star(\boldsymbol{z})}{q_a(\boldsymbol{z})} \right) \\
&= (1 - \gamma(|\mathcal{C}| - 1)) \log \left( \frac{1 - \gamma(|\mathcal{C}| - 1)}{q_a(\boldsymbol{u}_\mathcal{C})} \right) + \sum_{\boldsymbol{z} \in \mathcal{C} \setminus \{\boldsymbol{u}_\mathcal{C}\}} \gamma \log \left( \frac{\gamma}{q_a(\boldsymbol{z})} \right) \\
&\leq \log(\gamma^{-1}) . 
\end{aligned} \tag{92}$$

Then applying (91) and (92), we have

$$
\begin{aligned}
\text{term I} &= \sum_{t=a}^{b} \left[ \langle \boldsymbol{q}_t - \boldsymbol{q}^\star, \widehat{\boldsymbol{s}}_t \rangle + \langle \boldsymbol{p}_t - \boldsymbol{q}^\star, f_t \rangle + \langle \boldsymbol{q}^\star - \boldsymbol{q}_t, \widehat{\boldsymbol{s}}_t \rangle \right] \\
&\leq \frac{\log(\gamma^{-1})}{\eta} + \sum_{t=a}^{b} \left[ \langle \boldsymbol{p}_t - \boldsymbol{q}^\star, f_t \rangle + \langle \boldsymbol{q}^\star - \boldsymbol{q}_t, \widehat{\boldsymbol{s}}_t \rangle + \frac{1}{\eta} S_{\boldsymbol{q}_t}(\eta \widehat{\boldsymbol{s}}_t) \right] \\
&\overset{\text{(i)}}{\leq} \frac{\log(\gamma^{-1})}{\eta} + \mathsf{B} \left( \inf_{\substack{\boldsymbol{p} \in \Delta(\mathcal{C}), \\ E \in \mathcal{E}}} \Lambda_\eta(\boldsymbol{q}_t, \boldsymbol{p}, E) + \eta d \right) \\
&\overset{\text{(ii)}}{\leq} \frac{\log(\gamma^{-1})}{\eta} + \mathsf{B} \left( \eta \kappa d^4 \log(dT) + \eta d \right) ,
\end{aligned}
\tag{93}
$$

where (i) follows from the update rule (18) and the precision level assumed for solving the minimization problem (18) (see line 3 of Algorithm 3), and (ii) uses [1, Theorems 8.19 and 8.21] which establish that there exists a universal constant $\kappa$ such that

$$
\sup_{\boldsymbol{q} \in \tilde{\Delta}} \inf_{\substack{\boldsymbol{p} \in \Delta(\mathcal{C}), \\ E \in \mathcal{E}}} \frac{1}{\eta} \Lambda_\eta(\boldsymbol{q}, \boldsymbol{p}, E) \leq \kappa d^4 \log(dT) .
$$

Finally, combining (90) and (93) we obtain

$$
\begin{aligned}
\max_{\boldsymbol{u} \in \Theta} \sum_{t=a}^{b} \mathbf{E}[f_t(\boldsymbol{z}_t) - f_t(\boldsymbol{u})] &= \sum_{t=a}^{b} \langle \boldsymbol{p}_t - \boldsymbol{q}^\star, f_t \rangle + \sum_{t=a}^{b} \mathbf{E}_{\boldsymbol{z} \sim \boldsymbol{q}^\star}[f_t(\boldsymbol{z})] - \min_{\boldsymbol{u} \in \Theta} \sum_{t=a}^{b} f_t(\boldsymbol{u}) \\
&\leq (\varepsilon + 2\gamma|\mathcal{C}|)\mathsf{B} + \frac{\log(\gamma^{-1})}{\eta} + \mathsf{B}\left(\eta \kappa d^4 \log(dT) + \eta d\right) \\
&\lesssim \frac{\mathsf{B}}{T} + \sqrt{\mathsf{B}d^4 \log(dT) \log(T|\mathcal{C}|)} \lesssim d^{\frac{5}{2}} \sqrt{\mathsf{B}} ,
\end{aligned}
$$

where (i) applies $\varepsilon = \frac{1}{T}$, $\gamma = \frac{1}{T|\mathcal{C}|}$ and $\eta = \sqrt{\log(\gamma^{-1})/(d^4 \log(dT)\mathsf{B})}$, and (ii) is by selecting the covering set $\mathcal{C}$ such that $\log|\mathcal{C}| \leq d \log(1 + 16dT^2)$ (existence given by [1, Definition 8.12 and Exercise 8.13]). $\qquad\square$

The proof of Theorem 3 above relied on Lemma 8, which we present and prove below.

**Lemma 8.** *Consider Online Mirror Descent (OMD) with KL divergence regularization and fixed learning rate $\eta > 0$ applied to a sequence of loss estimates $\boldsymbol{s}_t \in \mathbb{R}^n$ for $t \in \mathbb{N}^+$. When run over a convex and complete domain $\tilde{\Delta} \subseteq \Delta^{n-1}$, the algorithm produces a sequence of updates $\boldsymbol{q}_t \in \tilde{\Delta}$ for $t \in \mathbb{N}^+$. For any comparator in $\boldsymbol{q}^\star \in \tilde{\Delta}$ and time interval $\{t \in \mathbb{N}^+ : a \leq t \leq b\}$, it holds that*

$$
\sum_{t=a}^{b} \langle \boldsymbol{q}_t - \boldsymbol{q}^\star, \boldsymbol{s}_t \rangle \leq \frac{1}{\eta} \left( KL(\boldsymbol{q}^\star \| \boldsymbol{q}_a) + \sum_{t=a}^{b} S_{\boldsymbol{q}_t}(\eta \boldsymbol{s}_t) \right) ,
$$

*where $S_{\boldsymbol{q}}(\eta \boldsymbol{s}) = \max_{\boldsymbol{q}' \in \Delta(\mathcal{C})} \langle \boldsymbol{q} - \boldsymbol{q}', \eta \boldsymbol{s} \rangle - KL(\boldsymbol{q}' \| \boldsymbol{q})$.*

*Proof of Lemma 8.* The proof is standard and included for completeness. Let $F$ denote the negentropy $F(\boldsymbol{q}) = \sum_{i=1}^{n} q_i \log(q_i)$ for $\boldsymbol{q} \in \Delta^{n-1}$, and note that

$$
KL(\boldsymbol{p} \| \boldsymbol{q}) = F(\boldsymbol{p}) - \langle \boldsymbol{p} - \boldsymbol{q}, \nabla F(\boldsymbol{q}) \rangle - F(\boldsymbol{q}) \quad \forall\, \boldsymbol{p}, \boldsymbol{q} \in \Delta^{n-1} .
\tag{94}
$$

Consider the update rule of the OMD defined in the lemma:

$$
\boldsymbol{q}_{t+1} = \arg\min_{\boldsymbol{q} \in \tilde{\Delta}} \langle \boldsymbol{q}, \eta \boldsymbol{s}_t \rangle + KL(\boldsymbol{q} \| \boldsymbol{q}_t) = \arg\min_{\boldsymbol{q} \in \tilde{\Delta}} \langle \boldsymbol{q}, \eta \boldsymbol{s}_t \rangle + F(\boldsymbol{q}) - \boldsymbol{q} \nabla F(\boldsymbol{q}_t) ,
$$

which implies by the first order optimality condition [9, Proposition 26.14] that, for any $\boldsymbol{q}^\star \in \tilde{\Delta}$ and time $t$,

$$
\langle \boldsymbol{q}^\star - \boldsymbol{q}_{t+1}, \eta \boldsymbol{s}_t + \nabla F(\boldsymbol{q}_{t+1}) - \nabla F(\boldsymbol{q}_t) \rangle \geq 0 .
\tag{95}
$$

Rearranging (95) and applying (94) we obtain

$$
\begin{aligned}
\langle \boldsymbol{q}_{t+1} - \boldsymbol{q}^\star, \boldsymbol{s}_t \rangle &\le \frac{1}{\eta} \langle \boldsymbol{q}^\star - \boldsymbol{q}_{t+1}, \nabla F(\boldsymbol{q}_{t+1}) - \nabla F(\boldsymbol{q}_t) \rangle \\
&= \frac{1}{\eta} \left( \mathrm{KL}(\boldsymbol{q}^\star \| \boldsymbol{q}_t) - \mathrm{KL}(\boldsymbol{q}^\star \| \boldsymbol{q}_{t+1}) - \mathrm{KL}(\boldsymbol{q}_{t+1} \| \boldsymbol{q}_t) \right) \\
&\le - \langle \boldsymbol{q}_t - \boldsymbol{q}_{t+1}, \boldsymbol{s}_t \rangle + \frac{1}{\eta} S_{\boldsymbol{q}_t}(\eta \boldsymbol{s}_t) + \frac{1}{\eta} \left( \mathrm{KL}(\boldsymbol{q}^\star \| \boldsymbol{q}_t) - \mathrm{KL}(\boldsymbol{q}^\star \| \boldsymbol{q}_{t+1}) \right) . \quad (96)
\end{aligned}
$$

Rearranging (96) and summing over $t \in [a, b]$ yields

$$
\begin{aligned}
\sum_{t=a}^{b} \langle \boldsymbol{q}_t - \boldsymbol{q}^\star, \boldsymbol{s}_t \rangle &= \sum_{t=a}^{b} \left( \langle \boldsymbol{q}_{t+1} - \boldsymbol{q}^\star, \boldsymbol{s}_t \rangle + \langle \boldsymbol{q}_t - \boldsymbol{q}_{t+1}, \boldsymbol{s}_t \rangle \right) \\
&\le \frac{1}{\eta} \sum_{t=a}^{b} \left( \mathrm{KL}(\boldsymbol{q}^\star \| \boldsymbol{q}_t) - \mathrm{KL}(\boldsymbol{q}^\star \| \boldsymbol{q}_{t+1}) + S_{\boldsymbol{q}_t}(\eta \boldsymbol{s}_t) \right) \\
&= \frac{1}{\eta} \left( \mathrm{KL}(\boldsymbol{q}^\star \| \boldsymbol{q}_a) - \mathrm{KL}(\boldsymbol{q}^\star \| \boldsymbol{q}_{b+1}) + \sum_{t=a}^{b} S_{\boldsymbol{q}_t}(\eta \boldsymbol{s}_t) \right) ,
\end{aligned}
$$

which combined with non-negativity of the KL divergence completes the proof. $\qquad \square$

Finally, we apply Theorem 3 to prove the bounds on $R^{\mathsf{swi}}(T, S)$, $R^{\mathsf{dyn}}(T, \Delta, S)$ and $R^{\mathsf{path}}(T, P)$ in Corollary 2, as well as the parameter-free guarantees in Corollary 4.

**Corollary 2.** *For any horizon $T \in \mathbb{N}^+$, Algorithm 3 calibrated as in Theorem 3 and tuned with interval size* B *(which determines $\eta$) satisfies the following regret guarantees:*

> ***Switching:*** $\quad \mathsf{B} = \frac{T}{S} \implies R^{\mathsf{swi}}(T, S) \lesssim d^{\frac{5}{2}} \sqrt{ST}$,
>
> ***Dynamic:*** $\quad \mathsf{B} = \frac{T}{S} \vee (d^{\frac{5}{2}} T / \Delta)^{\frac{2}{3}} \implies R^{\mathsf{dyn}}(T, \Delta, S) \lesssim R^{\mathsf{swi}}(T, S) \wedge d^{\frac{5}{3}} \Delta^{\frac{1}{3}} T^{\frac{2}{3}}$,
>
> ***Path-length:*** $\quad \mathsf{B} = (r d^{\frac{5}{2}} T / P)^{\frac{2}{3}} \implies R^{\mathsf{path}}(T, P) \lesssim r^{-\frac{1}{3}} d^{\frac{5}{3}} P^{\frac{1}{3}} T^{\frac{2}{3}}$.

*Proof of Corollary 2.* We prove these results by applying the adaptive regret guarantee from Theorem 3 and the conversions results from Proposition 1, similarly to the proof of Corollary 1. $\qquad \square$

**Corollary 4** (cExO with BoB)**.** *Let $T \in \mathbb{N}^+$. By partitioning the time horizon $[T]$ into epochs of length $L = d^{\frac{5}{2}} \sqrt{T}$, and employing Bandit-over-Bandit to select cExO's parameter* B *for each epoch from the set $\mathcal{B} = \{2^i : i = 0, 1, \dots, \lfloor \log_2 T \rfloor\}$, this algorithm achieves all regret bounds in Corollary 2 with an additional term of $d^{\frac{5}{4}} T^{\frac{3}{4}}$ (up to polylogarithmic factors).*

*Proof of Corollary 4.* The proof is similar to that of Corollary 3 and is therefore omitted. $\qquad \square$

