# OpenReview forum: "Non-stationary Bandit Convex Optimization: A Comprehensive Study"
_NeurIPS.cc/2025/Conference — NeurIPS 2025 poster_

### Official Review · Reviewer_JbAB · 2025-06-15

**Clarity:** 4
**Significance:** 2
**Originality:** 2
**Rating:** 4
**Confidence:** 3

**Summary:**

This paper studies a non-stationary bandit convex optimization problem where the non-stationarity is measured by the number of switches $S$, the total variation $\Delta$ of the loss functions, and the path-length $P$ of the comparator sequence. The authors propose two algorithms, a polynomial-time TEWA-SE and a non-polynomial-time cExO. By establishing the regret upper and lower bounds for both general convex and strongly-convex losses, TEWA-SE is shown to achieve minimax-optimal regret bounds with respect to known $S$ and $\Delta$ in strongly convex cases, while not knowing the strong-convexity parameter $\alpha$. In contrast, TEWA-SE is sub-optimal in the case of general convex losses. Furthermore, the cExO algorithm is shown to achieve minimax-optimal regret with respect to known $S$ and $\Delta$, and improve on the best existing bounds with respect to $P$. The two approaches can be both extended to the environments with unknown non-stationarity measures via Bandit-over-Bandit framework.

**Questions:**

See Strengths And Weaknesses

**Ethical Concerns:**

["NO or VERY MINOR ethics concerns only"]

**Final Justification:**

I have carefully reviewed all the comments and all the authors' responses. The discussion is very comprehensive and address most of my concerns. I believe all these discussions make the contribution more clearly. Therefore, I will raise my score to 4.

**Limitations:**

See Strengths And Weaknesses

**Quality:**

3

**Strengths And Weaknesses:**

**Strengths**:
1. The paper is well-written, the authors clearly convey their messages, stating the claims and validating via complete analysis.
2. The paper clearly and concisely discusses the advantages and limitations of the two algorithms. Combining with the regret lower bounds, this makes the reader clearly understand the difficulty of the problem.

**Weaknesses and Questions**:
1. The paper studies a relatively standard problem in the literature. There are some existing works considering the same problem with little difference (like different regret or different non-stationarity measure). Therefore, I am afraid that the novelty and contribution to this problem is limited. The paper should provide more discussions on the difference between this paper and the existing ones that study the almost same problem (like with different regret and with different non-stationarity measure), and showcase why the approaches in this paper is more worthy of attention.
2. Although the discussion of TEWA-SE is comprehensive and the result itself is interesting (although not minimax optimal in general case), I feel that both the design and analysis for TEWA-SE largely follows the standard approach in the OCO literature with adaptive learning rates. I understand that in OCO literature many approaches look similar, but it still makes it less interesting. Throughout reading Section 2 and Appendix C.1, I think the authors should discuss more on its difference between TEWA-SE and previous algorithms mentioned in the paper.
3. For the bound of TEWA-SE in the general convex losses cases, although I don't feel it is really a problem (to be published), I still wonder why it is hard to achieve the minimax-optimal regret bound with polynomial-time algorithm, when the non-stationarity measure is known. For example, in multi-armed bandit case, [1] achieves the minimax regret bounds in the case when non-stationarity (variation budget) is known, and sub-optimal otherwise.

**Comments**: Given the discussions above, I think the novelty and significance for this paper is not enough to be accepted by neurips. However, I would be happy to change my mind and increase my score if I miss some important points.

[1] Cheung, W. C., Simchi-Levi, D., & Zhu, R. (2019, April). Learning to optimize under non-stationarity. In The 22nd International Conference on Artificial Intelligence and Statistics (pp. 1079-1087). PMLR.

---

> ### Author Rebuttal · Authors · 2025-07-29
>
> ### **1. Novelty and contributions of this work, and differences from existing work**
>
> Thank you for this constructive feedback. While non-stationary Bandit Convex Optimization has been studied before [7, 13, 14, 33], our work provides the following meaningful advances:
>
> **Unifying study:** Attaining optimal performance for general non-stationary Convex Bandit Optimization is hard (see point 3 below), and the field currently lacks a comprehensive, unified treatment. Existing works study different aspects of the problem in isolation: [7, 14] focus on dynamic regret $ R^{\text{dyn}}(T, \Delta)$, while [13, 33] address path-length regret $R^{\text{path}}(T, P).$ As we illustrate in Fig. 1 and formalize in Proposition 1, the dynamic regret and path-length regret can both be upper bounded using the switching regret $R^{\text{swi}}(T, S)$, but the reverse does not necessarily hold, suggesting that our results may be more fundamental. This work is the first to systematically unify and extend previously scattered results, establishing a complete picture of the state-of-the-art. We also prove lower bounds and derive  parameter-free guarantees by applying the Bandit-over-Bandit framework, which offer baselines for future research on this topic.
>
> **Technical contributions:** Our proposed algorithms represent rigorous adaptations of established techniques, as detailed in lines 115-127 for TEWA-SE and lines 128-135 for cExO. Both algorithms build upon long lines of prior work.
> - **TEWA-SE:** We adapt sleeping experts and MetaGrad-type adaptivity techniques from Online Convex Optimization [21-23] to the bandit setting. Key modifications include replacing exact gradients with one-point bandit gradient estimates, and designing a quadratic surrogate function (see point 2 below). These adaptations require non-trivial technical work, including careful bias-variance trade-offs and detailed analysis of tilted exponentially weighted averaging with noisy gradient estimates.
> - **cExO:** The cExO algorithm builds on the exploration by optimization approach from [27] by adding a novel clipping step. Though not polynomial-time computable, cExO achieves the first minimax-optimal result for switching regret and dynamic regret w.r.t. $T, S$ and $\Delta$.
>
> Moreover, to clarify how our work differs from prior work, we would like to emphasize the following points:
> - The reduction from $R^{\text{path}}(T, P)$ to $R^{\text{swi}}(T, S)$ in Proposition 1 is new and employs simple geometric arguments (see Lemma 1 and its proof from line 863). To better position our work relative to existing work, we will add a remark discussing this reduction technique and referencing [17], noting that while our unified approach through $R^{\text{swi}}(T, S)$ simplifies the analysis of $R^{\text{path}}(T, P)$, it may yield slightly looser bounds on $R^{\text{path}}(T, P)$ compared to direct analysis.
> - We detail in point 2 below how the quadratic surrogate loss we tailored for TEWA-SE differs from its direct precursor in [23]. We will also emphasize this in the revised manuscript.
> - To our knowledge, the $d^{\frac{4}{5}}P^{\frac{2}{5}}T^{\frac{3}{5}}$ lower bound for path-length regret in Theorem 4 is new  and improves upon the only existing result, the $d\sqrt{PT}$ lower bound from [13, Theorem 5], w.r.t. $T$. This is attained by leveraging a different construction of a hard instance of loss functions.
>
> ### **2. Differences between TEWA-SE and prior work**
>
> Thank you for this comment. You are right that TEWA-SE builds on Zhang et al. [23] from Online Convex Optimization, which allows access to exact gradients. Apart from replacing exact gradients with one-point gradient estimates, we have also made a technical adaptation in the surrogate loss for TEWA-SE which we will explain more explicitly in the revised manuscript. While Zhang et al. [23] and the original MetaGrad paper [21] use $\eta^2||\boldsymbol{g}_t||^2||\boldsymbol{x}_t-\boldsymbol{x}||^2$ and $\eta^2(\boldsymbol{g}_t^\top(\boldsymbol{x}_t-\boldsymbol{x}))^2$ as the quadratic term for the surrogate loss, we use $\eta^2 G^2||\boldsymbol{x}_t-\boldsymbol{x}||^2$ in our surrogate loss $\ell_t^\eta(\boldsymbol{x})=-\eta\boldsymbol{g}_t^\top(\boldsymbol{x}_t-\boldsymbol{x})+\eta^2G^2||\boldsymbol{x}_t-\boldsymbol{x}||^2$ in (11), where $G$ is a high probability upper bound on $||\boldsymbol{g}_t||$ as in (10).
>
> The surrogate loss from [23] or [21] necessitates an additional condition relating $\mathbb{E}[||\boldsymbol{g}_t||]$ and $\mathbb{E}[||\boldsymbol{g}_t||^2]$ (or $\mathbb{E}[\boldsymbol{g}_t\boldsymbol{g}_t^\top]$) to be satisfied in the analysis, see e.g. [21, Theorem 2]. When working with tilted exponentially weighted average with noisy gradient estimates, directly using their quadratic term would complicate the proof. Our choice of the quadratic term, inspired by [22], simplifies this particular aspect of the proof.
>
> ### **3. Why harder to achieve optimality in non-stationary convex bandits than in multi-armed bandits**
>
> Thank you for this insightful comment. As we detail in lines 136-141, the techniques that achieve optimal regret in multi-armed bandits under non-stationarity (with known non-stationarity measures) cannot be directly applied to Bandit Convex Optimization (BCO). The continuous action space in BCO requires fundamentally different approaches.
>
> Achieving optimal time-dependence in BCO is challenging even for static regret, as noted in lines 130-132 and in point 2 of the review by Reviewer NBQt. The current state-of-the-art Fokkema el al. (2024) and Suggala et al. (2024) use the bandit version of the online Newton step for stochastic unconstrained BCO from Lattimore and György (2023) to achieve the optimal $\sqrt{T}$ time dependence (modulo polylogarithms). It remains unclear whether first-order methods, including our proposed TEWA-SE algorithm, can achieve the optimal rate, especially in the more challenging non-stationary environments. We will add a brief  discussion on this before the "Conclusion" section in the revised manuscript.
>
> To handle non-stationary environments, an interesting future direction is to combine TEWA-SE's aggregation mechanism with second-order expert algorithms similar to that in Fokkema et al. (2024) and Suggala et al. (2024). Another idea is to explore whether incorporating a restart criterion, similar to line 11 of Algorithm 1 in Fokkema et al. (2024) or line 15 of Algorithm 1 in Suggala et al. (2021), could enable change-detection and  improve regret guarantees compared to TEWA-SE.
>
> **References:**
> - Hidde Fokkema, Dirk van der Hoeven, Tor Lattimore, and Jack J Mayo. Online newton method for bandit convex optimisation. In *Conference on Learning Theory*, pages 1713–1714. PMLR, 2024.
> - Arun Suggala, Y Jennifer Sun, Praneeth Netrapalli, and Elad Hazan. Second order methods for bandit optimization and control. In *Conference on Learning Theory*, pages 4691–4763. PMLR, 2024.
> - Tor Lattimore and András György. A second-order method for stochastic bandit convex optimisation. In *Conference on Learning Theory*, pages 2067–2094. PMLR, 2023.
> - Arun Sai Suggala, Pradeep Ravikumar, and Praneeth Netrapalli. Efficient bandit convex optimization: Beyond linear losses. In *Conference on Learning Theory*, pages 4008–4067. PMLR, 2021.
> - (Numbered references above correspond to the bibliography in the submitted manuscript.)

---

> > ### Comment · Reviewer_JbAB · 2025-08-05
> >
> > Thank you for your valuable response. I have carefully reviewed all the comments and all the authors' responses. The discussion is very comprehensive. I believe all these discussions make the contribution more clearly. Therefore, I will raise my score to 4.

---

### Official Review · Reviewer_NBQt · 2025-06-26

**Clarity:** 3
**Significance:** 2
**Originality:** 3
**Rating:** 4
**Confidence:** 4

**Summary:**

This paper addresses the problem of adversarial convex bandits in a nonstationary setting. While nonstationary online convex optimization (OCO) and finite-armed bandits have been extensively studied, the bandit framework with a continuous action space remains underexplored.

The authors consider four notions of regret to measure nonstationarity: adaptive, switching, dynamic, and path-length regret. They first show that dynamic regret implies bounds for all the other notions. Consequently, the paper focuses primarily on dynamic regret, with results for the other measures derived as corollaries.

The authors then introduce the TEWA-SE algorithm, inspired by MetaGrad, which combines sleeping experts that rely on gradient estimations. While many of these techniques are standard in nonstationary online learning, the key novel idea lies in the design of a strongly convex surrogate loss, which allows all experts to share the same gradient estimator. This design enables the combination of expert predictions rather than bandit algorithm outputs. Assuming knowledge of the nonstationarity measure, the algorithm achieves an adaptive regret bound that is optimal (up to logarithmic factors) for strongly convex losses, and yields optimal or improved regret bounds for the various notions of nonstationarity.

However, this algorithm has two drawbacks: (1) it is suboptimal for general convex losses, and (2) it requires prior knowledge of the nonstationarity measure.

To address the first issue, the authors propose a second algorithm, based on a discretized and clipped version of exponential weights. While this algorithm is not computationally tractable, it achieves optimal adaptive regret bounds for general convex losses.

Adaptivity to the nonstationarity measure in both algorithms is obtained via a Bandit-over-Bandit technique. However, this comes at the cost of deteriorated regret guarantees.

**Questions:**

Remarks:

1. What is the optimal dependence on the strong convexity parameter $\alpha$? Would it be possible to derive a lower bound that shows an explicit dependence on $\alpha$, or is it possible that no such dependence appears in the optimal rate?

2. For smooth convex functions, could a better regret rate be achieved by optimizing $h$ differently in equation (34)? At first glance, choosing $h$ of order $B^{-1/6}$ appears to yield an adaptive regret of order $B^{2/3}$ instead of $B^{3/4}$. This would align with existing results in bandit convex optimization.

3. If I understand correctly, the only parameter that requires prior knowledge of the nonstationarity is $h$ in Algorithm 1. What regret bound do you obtain if $h$ is selected without knowing $B$ (e.g., by using a fixed schedule or data-driven tuning)?
More precisely, by choosing $h = T^{-1/6}$ in (34), independent of $B$, one could obtain an adaptive regret bound of order $B^{1/2} T^{1/6}$, thus removing the need for prior knowledge of the smoothness.
Similarly, if we avoid optimizing $h$ and instead analyze path-length regret, we might obtain (though I may have made some errors) a bound of the form $T h^2 + P^{1/3} T^{2/3} h^{-1/3}$. Optimizing $h$ independently of $P$ would then yield a path-length regret of order $P^{1/3} T^{5/7}$.
How does this simpler tuning strategy compare to your Bandit-over-Bandit (BoB) approach? At first glance, it seems to improve performance when $P$ is small and $B$ is large. A discussion comparing these approaches—both in the convex and strongly convex settings—would be valuable.

4. The paper states that TEWA-SE is polynomial-time, but as far as I can tell, the per-round computational complexity is $O(\log(T)^2)$, correct?


Minor remark:

The notation $B^d$ for the unit ball and $B$ for the nonstationarity measure may be confusing. It might help to disambiguate these symbols.

**Ethical Concerns:**

["NO or VERY MINOR ethics concerns only"]

**Final Justification:**

I read other reviews and rebuttal. The authors address well my most of my concerns. Some weaknesses are yet remaining (known nonstationary measure or non efficient algorithms in some settings). I lean toward acceptance and maintain my score.

**Limitations:**

Yes

**Quality:**

3

**Strengths And Weaknesses:**

Strengths:

The paper is clearly written and addresses an important problem. The authors provide several new regret upper bounds that either improve upon or match the optimal rates in nonstationary bandit convex optimization. These results take into account various notions of nonstationarity, both strongly convex and convex losses, and adaptivity to different parameters (e.g., nonstationarity and strong convexity). Overall, I believe this work makes a significant contribution to the field.

Weaknesses:

1. The technical novelty is limited. Most results are derived by combining well-established techniques—such as sleeping experts, MetaGrad, gradient estimation, and Bandit-over-Bandit methods—and the analysis largely follows from known arguments. That said, this should not diminish the value of the contributions: leveraging known tools in a clean, effective way is itself valuable, and simplicity in analysis can be a strength.

2. The algorithm that achieves optimal rates for convex losses is not computationally tractable. However, this limitation is inherent to the problem, which remains challenging even in the static setting.

3. Optimal regret rates are achieved only when the nonstationarity measure is known in advance.

---

> ### Author Rebuttal · Authors · 2025-07-29
>
> ### **1. Optimal dependence on strong-convexity parameter $\alpha$**
>
> Thank you for this insightful question. To the best of our knowledge, the optimal dependence on the strong convexity parameter $\alpha$ remains an open question with some intriguing gaps between upper and lower bounds.
>
> For static regret in adversarial Bandit Convex Optimization, the two tightest lower bounds we are aware of are [68, Theorem 7] by Shamir, which gives a $d\sqrt{T}$ lower bound with no dependence on $\alpha$, and [67, Theorem 6.1] by Akhavan et al. (with Hölder exponent equals 2), which implies a lower bound on the order of $d\sqrt{T}/\max(\alpha,1)$.
> - For $\alpha\lesssim 1$, the latter reduces to $d\sqrt{T}$ independent of $\alpha$, while existing upper bounds from [32, 69] achieve $d\sqrt{T}/\sqrt{\alpha}$. This suggests that the $1/\alpha$ dependence in our adaptive regret upper bound (15) in Theorem 1 is likely suboptimal.
> - For $\alpha\gtrsim 1$, our $d\sqrt{\textsf{B}}/\alpha$ adaptive regret upper bound in (15) appears to match the $d\sqrt{T}/\alpha$ lower bound from [67]. However, this comparison is not definitive because both results suppress dependence on the smoothness parameter, which could affect the true relationship between these bounds.
>
> To improve clarity, we will update the sentence starting on line 214 to read: "We further note that our bound for the strongly-convex case has a $\tfrac{1}{\alpha}$ dependency, which is suboptimal compared to the $ \frac{1}{\sqrt{\alpha}}$ dependency in [32, 69] for static regret in BCO for $\alpha\lesssim 1$."
>
> ### **2. Adaptive regret of $\mathcal{O}(\textsf{B}^{\frac{2}{3}})$ via alternative choice of $h$ in (34) for convex and smooth functions**
>
> Thank you for this important comment. We appreciate the opportunity to clarify the assumptions and scope of our results. Our results for TEWA-SE address two cases: (i) general convex and Lipschitz functions, and (ii) the special case of strongly-convex  and smooth functions. Importantly, the presented results for the convex case do *not* exploit smoothness assumptions. We will clarify this in the main text of the revised manuscript, rather than relegating it to footnote 1 on p.3.
>
> **Convex and Lipschitz functions:** For this setting, $\sqrt{d}T^{\frac{3}{4}}$ is a classical static regret rate in bandit optimization, see e.g., [24, Theorem 3.3]. Since our TEWA-SE algorithm employs the one-point gradient estimate from [24], our adaptive regret upper bound in (34) naturally inherits a leading term of order $\sqrt{d}\textsf{B}^{\frac{3}{4}}$.
>
> For brevity, our current manuscript proves the upper bounds for TEWA-SE for both the general convex case and the strongly-convex case using smoothness assumptions, with footnote 1 noting that the convex case guarantee can alternatively be obtained using only Lipschitz-continuity. In the revised manuscript, we will restructure our proofs to match the minimal assumptions: (i) prove the general convex case directly using only Lipschitz-continuity, and (ii) separately prove the strongly-convex case using smoothness.
>
> In the revised proof for the convex case, the smoothness-dependent term $\beta(q-p+1)h^2$ in (34) will be replaced by $2L(q-p+1)h$ under the Lipschitz-only assumption, where $L$ denotes the Lipschitz constant. Using our current parameter choice  $h=\sqrt{d}\textsf{B}^{-\frac{1}{4}}$, we will have  $2L(q-p+1)h\lesssim\sqrt{d}\textsf{B}^{\frac{3}{4}}$, which preserves the same $\sqrt{d}\textsf{B}^{\frac{3}{4}}$ rate from the original analysis.
>
> **Convex and smooth functions:** To avoid confusion, the revised manuscript will focus exclusively on cases (i) and (ii) described above. However, to answer your separate questions about convex and smooth functions:
> - **Clarification on $\textsf{B}^{\frac{2}{3}}$ rate:** You are correct that choosing $h=d^{\frac{1}{3}}\textsf{B}^{-\frac{1}{6}}$ in (34) which currently assumes convexity and smoothness would yield an adaptive regret of order $d^{\frac{2}{3}}\textsf{B}^{\frac{2}{3}}$. The $\textsf{B}^{\frac{2}{3}}$ term aligns with the time-dependence in the static regret upper bound of [29, Theorem 3].
> - **Clarification on choice of $h$:** TEWA-SE is designed to adapt to unknown curvature of the loss functions. Therefore, it is not feasible to choose different $h$ a priori for different functions, e.g., $h=d^{\frac{1}{3}}\textsf{B}^{-\frac{1}{6}}$ if the functions are convex and smooth, or $h=\sqrt{d}\textsf{B}^{-\frac{1}{4}}$ if they are strongly-convex and smooth. Our current proof uses $h=\sqrt{d}\textsf{B}^{-\frac{1}{4}}$ throughout. While fixing $h=d^{\frac{1}{3}}\textsf{B}^{-\frac{1}{6}}$ would yield a $d^{\frac{2}{3}}\textsf{B}^{\frac{2}{3}}$ adaptive regret for the convex case, this would compromise the minimax-optimality we achieve for the strongly-convex case in switching regret and dynamic regret.
>
> ### **3. Alternative choice of $h$ for parameter-free adaptive regret and path-length regret**
>
> Thank you for this detailed question. We respond to your question point-to-point:
> - You are correct that $h$ is the only parameter of TEWA-SE that is tuned using the non-stationarity measures such as $S, \Delta$ and $P$. As you note, choosing $h$ to be order of $T^{-\frac{1}{6}}$ independently of $\textsf{B}$ would yield an adaptive regret of $\textsf{B}^{\frac{1}{2}}T^{\frac{1}{6}}$ in (34) for the convex and smooth case, and $\textsf{B}T^{-\frac{1}{3}}+T^{\frac{1}{3}}$ in (41) for the strongly-convex and smooth case. However, this approach sacrifices the minimax-optimality of switching regret and dynamic regret in the strongly-convex setting.
> - We should clarify that our tuning of $h$ does not require knowledge of the smoothness parameter $\beta$.
> - Regarding your derivation of the parameter-free path-length regret, thank you for working through this analysis. If we understand correctly, your expressions (modulo a typo) were obtained through the following steps:
> \begin{align}
> R^{\text{path}}(T, P)
> &\stackrel{\text{(a)}}{\le}R^{\text{swi}}(T, S')+\tfrac{P}{r}\big\lceil\tfrac{T}{S'}\big\rceil\\\\
> &\stackrel{\text{(b)}}{\le} 2 S'\cdot R^{\text{ada}}(\big\lceil \tfrac{T}{S'}\big\rceil, T) + \tfrac{P}{r}\big\lceil \tfrac{T}{S'}\big\rceil\\\\
> &\stackrel{\text{(c)}}{\lesssim}  S'\cdot \Big(\tfrac{d}{h}\sqrt{\tfrac{T}{S'}} + \tfrac{T}{S'}h^2\Big) + \tfrac{P}{r}\tfrac{T}{S'}\\\\
> &\stackrel{\text{(d)}}{\lesssim}Th^2+P^{\frac{1}{3}}T^{\frac{2}{3}}h^{-\frac{2}{3}},
> \end{align}
> where (a) applies Proposition 1, (b) converts adaptive to switching regret as we do on line 860 (Lemma 1), (c) substitutes the adaptive regret bound from (34), and (d) optimizes by choosing $S'=P^{\frac{2}{3}}T^{\frac{1}{3}}h^{\frac{2}{3}}$. Finally, setting $h$ to be of order $T^{-\frac{1}{8}}$ simplifies the bound above into $P^{\frac{1}{3}}T^{\frac{3}{4}}$. While this approach avoids tuning $h$ based on $P$, it still requires knowledge of $P$ to set $S'$. In contrast, the presented Bandit-over-Bandit (BoB) approach is truly parameter-free by using an adversarial bandit algorithm to adaptively select $h$ for TEWA-SE in each epoch, without prior knowledge of any non-stationarity measures.
> - If we instead set $S'=T^{\frac{1}{3}}h^{\frac{2}{3}}$ independent of $P$ in step (d), the resulting bound becomes $PT^{\frac{3}{4}}$, which scales linearly in $P$. When $P$ is small, this bound is tighter than our $P^{\frac{1}{5}}T^{\frac{4}{5}}+T^{\frac{5}{6}}$ bound obtained using BoB. However, it becomes vacuous when $P\gtrsim T^{\frac{1}{4}}$. Our goal is to provide a non-vacuous bound for all $P=o(T)$, which BoB achieves.
>
> ### **4. Polynomial-time complexity of TEWA-SE**
>
> Yes, the computational complexity of TEWA-SE is $\mathcal{O}(\log^2 T)$ per round, as we reasoned in the paragraph containing equation (26) and stated below it. We describe this as polynomial-time because the overall computational complexity across the entire time horizon $T$ is polynomial in $T$.
>
> ### **5. Response to remark on $B^d$ vs. $\textsf{B}$ notation**
>
> Thank you. To distinguish from the parameter $\textsf{B}$ in TEWA-SE, we will use $\mathbb{B}^d$ and $\partial \mathbb{B}^d$ to denote the unit ball and unit sphere in the revised manuscript.
>
> **References:** Numbered references above correspond to the bibliography in the submitted manuscript.

---

### Official Review · Reviewer_m61f · 2025-06-28

**Clarity:** 4
**Significance:** 3
**Originality:** 3
**Rating:** 5
**Confidence:** 4

**Summary:**

The paper deals with the setting of Bandit Convex Optimisation (BCO) [1], whereby a learner is tasked with optimising over a sequence of functions $ f_{1},\dots,f_{T} $, which may very adversarially, given only noisy point-wise zero-order evaluations. Namely, given an action played at time $t\in[T]$, the action $x_{t} \in \Theta \subset \mathbb{R}^{d}$ ( $\Theta$ being convex) incurs the loss $f_{t}(x_{t})$, but only the random variable $y_{t}=f_{t}(x_{t})+\zeta_{t}$, whose expectation coincides with the true feedback, is available to the learner. Coinciding with recent interest in this problem, the present work studies the case where, in place of the usual notion of regret ($R_{T}=\min_{x\in \Theta}\left[\sum_{t=1}^{T}f_{t}(x_{t})-f_{t}(x)\right]$), several non-stationary variants are considered. These non-stationary variants constitute forms of the dynamic or path regret under constraints which render the problem tractable.

The authors provide an algorithm, TEWA-SE, based on MetaGrad [2] and its adaptation due to Zhang et al. [3], which efficiently attains the minimax rate in two of three cases for strongly convex functions, but is suboptimal in all cases for general convex losses. Also provided is an inefficient algorithm, cExO, which is based on the Exploration by Optimisation framework due to Lattimore & Gyorgi [4], and attains minimax rates in two out of three cases for general convex losses. Their results are extended to unknown non-stationary measures using the Bandit-over-bandit framework.

To verify optimality of the cases where $f_{t}$ is $\alpha$-strongly-convex (i.e. $f_t \in$ $\mathcal{F}^\alpha$ ) for all $ t \in [T] $, the authors have provided several new lower bounds on the dynamic and path regret using similar techniques to [5]. Namely, by way of constructing a sequence of hard functions within the class $\mathcal{F}^{\alpha}$.

[1] Bandit convex optimisation.Tor Lattimore. arXiv:2402.06535, 2024.

[2] Metagrad: Adaptation using multiple learning rates in online learning. Tim van Erven, Wouter M. Koolen, and Dirk van der Hoeven. Journal of Machine Learning Research, 22(161): 1–61, 2021.

[3] Dual adaptivity: a universal algorithm for minimizing the adaptive regret of convex functions. Lijun Zhang, Guanghui Wang, Wei-Wei Tu, Wei Jiang, and Zhi-Hua Zhou. In Proceedings of the 35th International Conference on Neural Information Processing Systems. Curran Associates Inc., 2021.

[4] Mirror descent and the information ratio. Tor Lattimore and Andras Gyorgy. In Conference on Learning Theory, pages 2965–2992. PMLR, 2021.

[5] Exploiting higher order smoothness in derivative-free optimization and continuous bandits. Arya Akhavan, Massimiliano Pontil, and Alexandre Tsybakov. Advances in Neural Information Processing Systems, 33:9017–9027, 2020.

**Questions:**

1. I would be very curious to know if the authors believe that the slack in the path-length dependent regret is on the algorithmic side, or that some slack is introduced in the additional bound in Theorem 4. I would expect the exploration-by-optimization framework to afford the minimax result, but I cannot quite see where there would be slack in the lower bound.

2. I am curious whether or not one can expect similar performance achieved by cExO from a similar adaptation of MetaGrad, as the key difference seems to be the inclusion of properly estimated second-order information. MetaGrad relies on gradient outer products to adapt, but the current work relies on a course gradient bound based on variation of the loss estimates). This second-order information which can be estimated using single-point feedback with the convolution-based estimator proposed in [1] (and has found recent use in the works [2,3,4]), so I'm curious whether or not the authors suspect that the quadratic surrogate of this estimator would be sufficient to achieve a better rate.

3. Would the authors suspect that second-order methods might be adaptable to the case of dynamic & path-dependent regret, as those based on variants of FTRL should already have something of a tracking property. This question may indicate a limitation in my own understanding of the degree of adaptation needed to furnish rates for the dynamic & path-dependent regret.

[1] Kernel-based methods for bandit convex optimization. Sébastien Bubeck, Yin Tat Lee, and Ronen Eldan. 2017. In Proceedings of the 49th Annual ACM SIGACT Symposium on Theory of Computing (STOC 2017). Association for Computing Machinery, New York, NY, USA, 72–85. https://doi.org/10.1145/3055399.3055403

[3] Improved Regret for Zeroth-Order Stochastic Convex Bandits. Tor Lattimore, Andras Gyorgy Proceedings of Thirty Fourth Conference on Learning Theory, PMLR 134:2938-2964, 2021.

[2] A Second-Order Method for Stochastic Bandit Convex Optimisation. Tor Lattimore, András György Proceedings of Thirty Sixth Conference on Learning Theory, PMLR 195:2067-2094, 2023.

[4] Online newton method for bandit convex optimisation. Hidde Fokkema, Dirk van der Hoeven, Tor Lattimore, and Jack J Mayo. In Proceedings of Thirty Seventh Conference on Learning Theory, volume 247 of Proceedings of Machine Learning Research, pages 1713–1714. PMLR, 2024.

**Ethical Concerns:**

["NO or VERY MINOR ethics concerns only"]

**Final Justification:**

The authors have addressed my questions and those of the other reviewers, transparently delineating their contributions, degree of novelty, and have provided clear motivation for the approach. Overall, I believe this work constitutes a solid theoretical contribution, with several new results closing or partially closing gaps in the BCO literature. A downside of the work is the degree of novelty of the algorithms themselves, which inherit much of there structure from existing works, particularly MetaGrad and ExO [2], including the computational burden of the latter. Consequently, it is difficult to justify a higher score than 5. Nevertheless, this work represents an important stepping stone towards more flexible and efficient algorithms for this setting, and I believe that the insights gained through its development will be of interest to the community.

[1] Metagrad: Adaptation using multiple learning rates in online learning. Tim van Erven, Wouter M. Koolen, and Dirk van der Hoeven. Journal of Machine Learning Research, 22(161): 1–61, 2021.
[2] Mirror descent and the information ratio. Tor Lattimore and Andras Gyorgy. In Conference on Learning Theory, pages 2965–2992. PMLR, 2021.

**Limitations:**

In general yes. A minor comment would be that in line 108 - “and further developed in subsequent studies such as [28–32]. ” - there are a few relatively recent references missing; I realise that this comment is not meant as exhaustive, but the two papers by Suggula and co. (see below) are a relatively important part of the story. Particularly the latter, which coincided historically with the results of Fokkema et al., and constitute state-of-art rates for a class of $\kappa$-convex functions (basically just functions with well-conditioned hessians).

Here are the references:

Efficient Bandit Convex Optimization: Beyond Linear Losses; Arun Sai Suggala, Pradeep Ravikumar, Praneeth Netrapalli Proceedings of Thirty Fourth Conference on Learning Theory, PMLR 134:4008-4067, 2021.

Second Order Methods for Bandit Optimization and Control; Arun Suggala, Y Jennifer Sun, Praneeth Netrapalli, Elad Hazan Proceedings of Thirty Seventh Conference on Learning Theory, PMLR 247:4691-4763, 2024.

**Paper Formatting Concerns:**

No issues.

**Quality:**

4

**Strengths And Weaknesses:**

Strengths:
- The work does indeed provide a comprehensive study of non-stationary BCO, and closes several conspicuous gaps in the literature with new (close to) minimax optimal bounds. These constitute a set of results and ideas of significant interest to the community.
- The nuanced lower bounds illustrating clearly the (near) optimality of the results close some slack in previous results.
- The motivation and intuition behind algorithmic choices are extremely clear, and open up several new avenues for development and follow-up.
- The paper is well written, contributions are clearly delineated, and the overall structure serves as a template for what such a broad study of such a problem setting should entail.
- The proofs are easy enough to follow at a high level, are firmly grounded in existing analyses, and thus I have very little doubt as to their correctness.

Weaknesses:
- Although the results themselves constitute new and interesting points on the pareto frontier of possibilities in dynamic environments, I have had a hard time finding algorithmic ideas which weren't already derived elsewhere.
- It’s somewhat low hanging fruit (and a rather minor weakness) to say that the results are comprehensive, but not quite complete, as there still remain questions on the achievability of some of the results of cExO constructively (or rather, with an efficient algorithm).
- In complement to the above, some of the state-of-art techniques in bandit convex optimisation haven’t been discussed substantively, or alluded to in what might be an appropriate context, and it strikes me that some of these techniques might have afforded a more efficient approach than the cExO algorithm. Or at least it would be instructive to know why these techniques weren't alluded to when developing the cExO algorithm. Nevertheless, I see this weakness as very minor.

---

> ### Author Rebuttal · Authors · 2025-07-29
>
> ### **1. Reasons for slack between upper and lower bounds for path-length regret for cExO**
>
> Thank you for this insightful comment. As far as we can see, there are three potential sources of slack between our upper and lower bounds for cExO: **(i)** our lower bound construction in Theorem 4 may not be tight, **(ii)** the cExO algorithm design itself may be suboptimal, or **(iii)** our upper bound analysis may have introduced looseness through the reduction result in Proposition 1, which bounds path-length regret $R^{\text{path}}(T, P)$ through switching regret $R^{\text{swi}}(T, S)$. We examine each in turn:
> - **(i)** Though our $d^{\frac{4}{5}}P^{\frac{2}{5}}T^{\frac{3}{5}}$ lower bound shows better $T$-dependence than the $d\sqrt{PT}$ from [13, Theorem 5] (the only other result in the literature), we are unsure about the true minimax rate. While we also don't see slack in the lower bound, a $T^\frac{3}{5}$ rate is uncommon and we wouldn't be surprised if the cExO upper bound is tight.
> - **(ii)** Alternatively, a potential source of sub-optimality in cExO stems from the fact that cExO performs OMD over the distribution space rather than directly over the action set. The latter would allow us to derive path-length regret bounds more directly.
> - **(iii)** Our Proposition 1 unifies the regret analysis by bounding  $R^{\text{path}}(T, P)$ and $R^{\text{dyn}}(T, \Delta)$ through $R^{\text{swi}}(T, S)$. However, this simple approach may yield slightly looser bounds for $R^{\text{path}}(T, P)$ compared to direct analysis, as noted in [17].
>
> We will include a brief discussion on the above in the revised manuscript.
>
> ### **2. MetaGrad combined with convolution-based estimator from Bubeck et al. (2017)**
>
> Thank you for this thoughtful suggestion. As you note, MetaGrad is a first-order algorithm, where the quadratic term $\eta^2(\boldsymbol{g}_t^\top(\boldsymbol{x}_t-\boldsymbol{x}))^2$ in the surrogate loss uses only a simple outer product of the gradient estimate $\boldsymbol{g}_t\in\mathbb{R}^d$. Our TEWA-SE adaptation of MetaGrad inherits this first-order property, which is likely the reason why it achieves optimal rates for strongly-convex functions, but cannot attain optimality for general convex functions without incorporating second-order information. We will add a brief discussion on this before the "Conclusion" section in the revised manuscript.
>
> The kernel-based (or convolution-based) estimator from Bubeck et al. (2017) offers a promising approach to address this limitation. It adaptively exploits second-order structure through the covariance of the current exponential weights distribution, essentially capturing the geometry of where the algorithm currently believes the optimum lies. Replacing the simple gradient outer product with this adaptive kernel-based estimator as the quadratic term in TEWA-SE's surrogate loss could in theory improve our regret bounds. However, implementing this integration would require careful analysis of how the kernel-based estimates interact with TEWA-SE's sleeping experts framework. This is an interesting direction for future work.
>
> More broadly, your suggestion of leveraging second-order methods is very promising. Given that the online Newton methods from Fokkema et al. (2024) and Suggala et al. (2024) provide the state-of-the-art static regret for adversarial convex bandits, a natural future direction is to extend such methods to non-stationary environments.
>
> ### **3. Second-order methods and tracking capabilities**
>
> Thank you for this insightful question. In a setting similar to ours, we are not aware of any existing work that uses second-order methods to track changes in the environment for the purpose of minimizing switching, dynamic, or path-length regret. We would be grateful if you could point us to any such work. Nevertheless, a promising direction could be to incorporate a restart criterion, similar to the one used in line 15 of Algorithm 1 in Suggala et al. (2021), or line 11 of Algorithm 1 in Fokkema et al. (2024), which may enable tracking capabilities and potentially yield improved regret guarantees.
>
> ### **4. Response to remark on missing references**
>
> Thank you for pointing out two missing references in Bandit Convex Optimization (BCO). We will include both papers at line 108 in the revised manuscript.
>
> Indeed, Suggala et al. (2024) appeared concurrently with Fokkema et al. (2024) and presents an online Newton step algorithm that achieves optimal $\sqrt{T}$ static regret for a class of convex functions with well-conditioned Hessians. Both Suggala et al. (2024) and Fokkema et al. (2024) build upon the bandit version of the online Newton step for stochastic unconstrained BCO from Lattimore and György (2023), which we will also reference at line 108. Finally, the restart condition in Fokkema et al. (2024) is derived from Suggala et al. (2021), making the latter a key reference for understanding the historical connections.
>
> **References:**
> - Sébastien Bubeck, Yin Tat Lee, and Ronen Eldan. Kernel-based methods for bandit convex optimization. In *Proceedings of the 49th Annual ACM SIGACT Symposium on Theory of Computing*, pages 72–85, 2017.
> - Hidde Fokkema, Dirk van der Hoeven, Tor Lattimore, and Jack J Mayo. Online newton method for bandit convex optimisation. In *Conference on Learning Theory*, pages 1713–1714. PMLR, 2024.
> - Arun Suggala, Y Jennifer Sun, Praneeth Netrapalli, and Elad Hazan. Second order methods for bandit optimization and control. In *Conference on Learning Theory*, pages 4691–4763. PMLR, 2024.
> - Tor Lattimore and András György. A second-order method for stochastic bandit convex optimisation. In *Conference on Learning Theory*, pages 2067–2094. PMLR, 2023.
> - Arun Sai Suggala, Pradeep Ravikumar, and Praneeth Netrapalli. Efficient bandit convex optimization: Beyond linear losses. In *Conference on Learning Theory*, pages 4008–4067. PMLR, 2021.
> - (Numbered references above correspond to the bibliography in the submitted manuscript.)

---

> > ### Comment · Reviewer_m61f · 2025-08-05
> >
> > I thank the authors for the detailed and thoughtful responses to my questions.
> >
> > 1. Thanks for the discussion here. This additional explanation would give some very useful context in the manuscript.
> > 2. I see - thanks a lot for the clarification; I note at least one issue with the suggestion. Indeed, the analysis of MetaGrad hinges crucially on controlling the surrogate regret (Lemma 23 in van Erven et al. (2021)) using the multiple rate weighting scheme, and this guarantees positivity of the mixing term, in turn allowing the surrogate to be bounded. Using a second-order estimate creates an additional Jensen gap before the use of the "prod" inequality, and this is not trivial to resolve immediately.
> > 3. Interesting idea! Since the restart criteria for Suggala et al. (2024) and Fokkema et al. (2024) involves negative static regret (detection that the present optimum has left the confidence region), I would not be surprised if a version of constrained dynamic regret could be similarly controlled.
> >
> > I maintain my score.

---

> > > ### Author Response · Authors · 2025-08-05
> > >
> > > Thank you for your detailed and thoughtful response. We greatly appreciate the productive discussion during the rebuttal period.
> > >
> > > Regarding point 2, you raise an important technical concern. Indeed, applying the "prod" inequality in Eq. (64) of the submitted manuscript would likely be significantly more challenging with second-order estimates and could result in looser bounds.
> > >
> > > Thank you for sharing intuition about point 3. More broadly, your suggestion to explore second-order methods represents a very promising direction for improving regret guarantees in non-stationary convex bandits, and is likely the stepstone toward achieving optimality.

---

### Official Review · Reviewer_TTmz · 2025-07-02

**Clarity:** 2
**Significance:** 3
**Originality:** 2
**Rating:** 5
**Confidence:** 1

**Summary:**

The authors study the nonstationary convex bandit setting. They work with a “total variation” notion of non-stationarity and two forms of adaptive comparators.

They focus on several notions of regret for this nonstationary setting and different assumptions on environment parameter knowledge, ultimately proposing two different algorithms with a range of different theoretical properties.

Their first contribution is a polynomial-time algorithm based off of the sleeping experts framework that achieves $\mathcal{O}(\sqrt{d}B^{3/4})$ adaptive regret for general convex losses (where $B$ is the length of an interval of time), $\mathcal{O}(\sqrt{d} S^{1/4}T^{3/4})$ switching regret. They also show that if the losses are strongly convex, an if $\Delta$ is known, their algorithm achieves a dynamic regret of $\mathcal{O}(\min\\{ d\sqrt{ST}, d^{2/3}\Delta^{1/3} T^{2/3}\\})$. This last bound is shown to be minimax optimal by providing a matching lower bound.

To overcome the requirement of knowing $S$, $\Delta$ and $P$, the authors use a Bandit over Bandit framework.

For general convex losses with known $S$, $\Delta$ and $P$, their algorithm is suboptimal, so they provide another algorithm with improved guarantees by using discretized action space. However, this algorithm is not polynomial-time computable.

**Questions:**

I think it would have been beneficial for the reader if the authors had picked, e.g. a single notion of regret and setting and written a more in-depth version of that in the main paper with more space dedicated to providing intuition to the reader. The other variants could be provided in the appendix. To be clear, I am not suggesting that the authors' work itself is of poor quality; rather, I think the organisation of the contributions could be arranged more helpfully. It currently reads like a full ablation was done over the range of scenarios, and all results are being included in large paragraphs, one chunk at a time. This makes for very dense passages with lots of internal context switching.

**Ethical Concerns:**

["NO or VERY MINOR ethics concerns only"]

**Final Justification:**

I think this paper is technically solid if poorly organised. I recommend acceptance, though I am not deeply familiar with the line of work the authors focus on.

**Limitations:**

Yes

**Paper Formatting Concerns:**

No concerns

**Quality:**

3

**Strengths And Weaknesses:**

*Strengths:*

- Comprehensive treatment of the non-stationary online convex bandit setting
- Extensive regret analysis with many considerations for different assumptions
- Extensive discussion of limitations of their contributions
- Clear progress from the first algorithm to the second algorithm to address shortcomings

*Weaknesses:*

- In general I did not find this paper to be pleasant to read, there were so many juggling defintions of regret, so many settings (e.g. general convex, strongly convex), and so many different assumptions on knowledge (e.g. $\Delta$ known and incorpated into tuning vs not) as well as two very different proposed algorithms that it was very challenging to construct a coherent picture of the full scope of contributions. It read like a summarized thesis instead of a focused study.

---

> ### Author Rebuttal · Authors · 2025-07-29
>
> ### **Response to remarks on organization and presentation**
>
> Thank you for your constructive feedback. We aimed to provide a comprehensive study because the field currently lacks a unfied treatment of non-stationary Convex Bandit Optimization. However, we agree that the presentation could be made more accessible.
>
> As we note in lines 105-114, existing works study different aspects of the problem in isolation: [7, 14] focus on dynamic regret $ R^{\text{dyn}}(T, \Delta)$, while [13, 33] address path-length regret $R^{\text{path}}(T, P)$. As we illustrate in Fig. 1 and formalize in Proposition 1, the dynamic regret and path-length regret can both be upper bounded using the switching regret $R^{\text{swi}}(T, S)$, but the reverse does not necessarily hold, suggesting that our results may be more fundamental. This work is the first to systematically unify and extend previously scattered results, establishing a complete picture of the state-of-the-art. We also prove lower bounds and derive parameter-free guarantees.
>
> In the revision, we will emphasize in the "Introduction" section before Section 1.1 that adaptive regret $R^{\text{ada}}(\textsf{B}, T)$ and switching regret $R^{\text{swi}}(T, S)$ are the primary focus of this work, and that dynamic regret and path-length regret can be bounded using $R^{\text{ada}}(\textsf{B}, T)$ and $R^{\text{swi}}(T, S)$. We will expand our discussion of Fig. 1 which visually clarifies the conversions between different regret notions. We will also expand Section 1.3 where we formally present the conversion results (Proposition 1) to provide clearer intuition before diving into technical details.
>
> **References:** Numbered references above correspond to the bibliography in the submitted manuscript.

---

> > ### Comment · Reviewer_TTmz · 2025-08-08
> >
> > Thank you for your response. I am still concerned about the overall focus and organisation of this paper, but I think it is a technically solid.

---

### Official Review · Reviewer_5yot · 2025-07-03

**Clarity:** 3
**Significance:** 3
**Originality:** 2
**Rating:** 6
**Confidence:** 4

**Summary:**

This paper studies minimizing dynamic regret for online bandit convex optimization. The authors provide new algorithms that can solve both convex and strongly convex functions, without knowing the strong convexity parameter. The regret bound for strongly convex functions is minimax optimal, whereas the regret for convex functions is suboptimal. The authors also provide an inefficient algorithm that has a better dependence on T, the total number of iterations.

**Questions:**

The questions are given above.

**Ethical Concerns:**

["NO or VERY MINOR ethics concerns only"]

**Final Justification:**

I have read the rebuttal as well as the other reviews. Overall, I believe this is a solid paper that makes valuable contributions to the BCO area. After discussing with the authors, the techniques also appear novel to me. I would like to maintain my score.

**Limitations:**

Yes. Please see the above section for more discussion.

**Quality:**

3

**Strengths And Weaknesses:**

Strengths:
---

Significance of the results:

This paper provides new and improved results for non-stationary bandit convex optimization:

For minimizing dynamic regret in the BCO setting, previous work mainly focuses on the setting for general convex functions and path-length regret, while this paper provides new results for strongly convex functions, and also regret with respect to other competitor-dependent terms, such as S, Delta.


The authors apply the Metagrad-style techniques to deal with strongly convex problems, and the main advantage is that the algorithm does not need to know the strong convexity parameter. This is also new to the non-stationary BCO setting. Moreover, the basic algorithm requires the exact value of P, Delta, and S as input, while the authors extend their algorithm with the BoB framework to deal with the setting where such parameters are unknown.


Finally, the authors also provide a highly inefficient algorithm, but with improved regret bounds with respect to T, even with out the assumption of strong convexity.


Presentation:

The paper is in general well-written and easy to follow.


Weakness:
---
My main concern is about the novelty of the algorithm and the connection to previous works. Specifically:

The proposed method in Algorithm 1 seems to be exactly exactly the algorithm in [23] for strongly convex functions, with OGD as the expert algorithm, the same surrogate loss function given in (11), and replacing the true gradient g_t with estimated gradient with one-point estimation, whose variance and bias is well studied in the bandit literature. It seems that the main thing one needs to do is to plug in the extra error term introduced by the one-point gradient estimation into the original proof given in [23] or the Metagrad algorithm [21]. Are there any extra significant challenges in the proof preventing us from doing so?

Other limitations:


The proposed TEWA-SE can deal with both general convex and strongly convex functions. However, the bound for general convex functions is sub-optimal as the dependence on T gets worse.


The cExO algorithm given in Section 3, which follows the algorithm given in [1], requires running exponential on the entire (discretized) decision space, which is already highly inefficient. Moreover, it requires solving the intractable problem in (18). Finally, it suffers a worse dependence on d.

---

> ### Author Rebuttal · Authors · 2025-07-29
>
> ### **1. Response to remarks on novelty of TEWA-SE and connections to prior work**
>
>  Thank you for this comment.
>
> **TEWA-SE:**
> Indeed, our proposed TEWA-SE algorithm represents rigorous adaptations of established techniques, as detailed in lines 115-127. We adapt sleeping experts and MetaGrad-type adaptivity techniques from Online Convex Optimization [21-23] to the bandit setting. Key modifications include replacing exact gradients with one-point bandit gradient estimates, and designing a quadratic surrogate function. These adaptations require non-trivial technical work, including careful bias-variance trade-offs and detailed analysis of tilted exponentially weighted averaging with noisy gradient estimates.
>
> To further clarify how TEWA-SE differs from its precursor [23], we would like to emphasize that our surrogate loss design differs from both [23] and the original MetaGrad paper [21]. We will describe this more explicitly in the revised manuscript. While [23] and [21] use $ \eta^2||\boldsymbol{g}_t||^2||\boldsymbol{x}_t-\boldsymbol{x}||^2$ and $\eta^2(\boldsymbol{g}_t^\top(\boldsymbol{x}_t-\boldsymbol{x}))^2$ as the quadratic term for the surrogate loss, we use $\eta^2G^2||\boldsymbol{x}_t-\boldsymbol{x}||^2$ in our   surrogate loss $\ell_t^\eta(\boldsymbol{x})=-\eta\boldsymbol{g}_t^\top(\boldsymbol{x}_t-\boldsymbol{x})+\eta^2G^2||\boldsymbol{x}_t-\boldsymbol{x}||^2$ in (11), where $G$ is a high probability upper bound on $||\boldsymbol{g}_t||$, as shown in (10).
>
> The surrogate loss from [23] or [21] necessitates an additional condition relating $\mathbb{E}[||\boldsymbol{g}_t\|]$ and $\mathbb{E}[||\boldsymbol{g}_t||^2]$ (or $\mathbb{E}[\boldsymbol{g}_t\boldsymbol{g}_t^\top]$) to be satisfied in the analysis, see e.g., [21, Theorem 2]. When working with tilted exponentially weighted average with noisy gradient estimates, directly using their quadratic term would complicate the proof. Our choice of the quadratic term, inspired by [22], simplifies this particular aspect of the proof.
>
> **Broader connections to prior work:**
> Attaining optimal performance for general non-stationary Convex Bandit Optimization is hard, and the field currently lacks a comprehensive, unified treatment. Existing works study different aspects of the problem in isolation: [7, 14] focus on dynamic regret $R^{\text{dyn}}(T, \Delta)$, while [13, 33] address path-length regret $R^{\text{path}}(T, P).$ As we illustrate in Fig. 1 and formalize in Proposition 1, the dynamic regret and path-length regret can both be upper bounded using the switching regret $R^{\text{swi}}(T, S)$, but the reverse does not necessarily hold, suggesting that our results may be more fundamental. This work is the first to systematically unify and extend previously scattered results, establishing a complete picture of the state-of-the-art. We also prove lower bounds and derive parameter-free guarantees by applying the Bandit-over-Bandit framework, which offer baselines for future research on this topic.
>
> Furthermore, to clarify how our work compares with prior work, we would like to emphasize the following points:
> - The reduction from $R^{\text{path}}(T, P)$ to $R^{\text{swi}}(T, S)$ in Proposition 1 is new and employs simple geometric arguments (see Lemma 2 and its proof from line 863). To better position our work relative to existing work, we will add a remark discussing this reduction technique and referencing [17], noting that while our unified approach through $R^{\text{swi}}(T, S)$ simplifies the analysis of $R^{\text{path}}(T, P)$, it may yield slightly looser bounds on $R^{\text{path}}(T, P)$ compared to direct analysis.
> - To our knowledge, the $d^{\frac{4}{5}}P^{\frac{2}{5}}T^{\frac{3}{5}}$ lower bound for path-length regret in Theorem 4 is new  and improves upon the only existing result, the $d\sqrt{PT}$ lower bound from [13, Theorem 5], w.r.t. $T$. This is attained by leveraging a different construction of a hard instance of loss functions.
>
> ### **2. Response to remark on limitations of cExO**
>
> The cExO algorithm builds on a long line of work, as detailed in lines 128-135 of our manuscript. Specifically, it adapts the exploration by optimization approach from [27] by adding a novel clipping step. Though not polynomial-time computable, cExO achieves the first minimax-optimal result for switching regret and dynamic regret w.r.t. $T, S$ and $\Delta$. Moreover, obtaining optimal regret guarantees in non-stationary environments is inherently more challenging than in the static setting. The suboptimal dependence on $d$ in our upper bound is unsurprising since even the best known static regret bounds of Fokkema et al. (2024) and Suggala et al. (2024) suffer from similar dimensional dependence.
>
> **References:**
> - Hidde Fokkema, Dirk van der Hoeven, Tor Lattimore, and Jack J Mayo. Online newton method for bandit convex optimisation. In *Conference on Learning Theory*, pages 1713–1714. PMLR, 2024.
> - Arun Suggala, Y Jennifer Sun, Praneeth Netrapalli, and Elad Hazan. Second order methods for bandit optimization and control. In *Conference on Learning Theory*, pages 4691–4763. PMLR, 2024.
> - (Numbered references above correspond to the bibliography in the submitted manuscript.)

---

> > ### Comment · Reviewer_5yot · 2025-08-05
> > **Response**
> >
> > Thank you for the detailed response, and I would like to apologize for the delay in reply.
> >
> > About surrogate loss: It seems that [22] also uses the same term G^2||x_t -x^*||^2. (in their equation (6)), so I am not sure why this is claimed to be a main novelty of the proposed methods. Could the authors add more discussion to this?

---

> > > ### Author Response · Authors · 2025-08-05
> > >
> > > We thank the reviewer for their thoughtful response. We agree that the TEWA-SE algorithm builds upon prior work, and we'd like to clarify our contributions.
> > >
> > > As mentioned in our original submission (line 195) and in our rebuttal, our surrogate function construction was inspired by [22]. However, [22] introduces this surrogate function in a different context, specifically for minimizing static regret with exact loss gradients available.
> > >
> > > Our contribution lies in integrating three distinct components: the surrogate function from [22], the sleeping experts framework, and noisy one-point gradient estimates. This combination enables us to extend [22]'s approach to establish adaptive and dynamic regret guarantees. Notably, in [22] (Section 2, last paragraph), the authors state that "In this paper we mainly focus on the minimization of regret, and it is an interesting question to explore whether our method can be extended to adaptive and dynamic regrets." Our work and [23] address this question in different settings.
> > >
> > > In our rebuttal, our emphasis on TEWA-SE's integration of various technical components was intended to directly address the reviewer's concern that TEWA-SE is "exactly the algorithm in [23] for strongly convex functions." While TEWA-SE may appear similar to [23], using the surrogate function from [23] or the original MetaGrad paper [21] would require additional restrictive conditions relating $\mathbb{E}[||\boldsymbol{g}_t||]$ and $\mathbb{E}[||\boldsymbol{g}_t||^2_2]$ (or $\mathbb{E}[\boldsymbol{g}_t\boldsymbol{g}_t^\top]$) to be satisfied and may yield suboptimal rates in dimension $d$ for strongly-convex losses, similar to [21]. Our deliberate choice of the surrogate function from [22] eliminates these limitations and allows us to achieve optimality w.r.t. $T, S$ and $\Delta$ in the strongly-convex case.
> > >
> > > Finally, as we detailed in our rebuttal, we wish to emphasize that TEWA-SE represents just one contribution of this manuscript. More broadly, we extend and unify several existing results while establishing new reduction results and tighter lower bounds. We also design cExO as the first algorithm achieving optimality w.r.t. $T, S$ and $\Delta$ for general convex loss functions using exploration by optimization with clipping, though without polynomial-time computability.

---

> > > > ### Comment · Reviewer_5yot · 2025-08-05
> > > > **Response**
> > > >
> > > > Thank you for the clarification!

---

### Decision · Program_Chairs · 2025-09-17

**Decision:**

Accept (poster)

**Comment:**

This paper studies several variants of Bandit Convex Optimization (BCO) under different non-stationarity assumptions. The reviewing team was overwhelmingly positive about the submission. Clear accept. Good job!

Please incorporate all the feedback received from the reviewers in the revised version.